# MMA-Asia: A Multilingual and Multimodal Alignment Framework for Culturally-Grounded Evaluation

## Abstract

Large language models (LLMs) are now used worldwide, yet their multimodal understanding and reasoning often degrade outside Western, high-resource settings. A critical, yet underexplored challenge is assessing whether these models possess genuine "cultural awareness" that is consistent across different modalities (text, vision, speech) and languages. We propose MMA-ASIA, a comprehensive framework to evaluate LLMs' cultural awareness with a focus on Asian contexts. MMA-ASIA centers on a human-curated, multilingual, and multimodally aligned multiple-choice benchmark covering 8 Asian countries and 10 languages, comprising 27,000 questions; over 79% require multi-step reasoning grounded in cultural context, moving beyond simple memorization. Crucially, this is the first dataset aligned at the input level across three modalities: text, image (visual question answering), and speech. This enables direct tests of cross-modal transfer. Building on this benchmark, we propose a five-dimensional evaluation protocol that measures – (i) cultural-awareness disparities across countries, (ii) cross-lingual consistency, (iii) cross-modal consistency, (iv) cultural knowledge generalization, and (v) grounding validity. To ensure rigorous assessment, a Cultural Awareness Grounding Validation Module detects "shortcut learning" by checking whether the requisite cultural knowledge supports correct answers. Finally, through comparative model analysis, attention tracing, and an innovative Vision-ablated Prefix Replay (VPR) method, we probe why models diverge across languages and modalities, offering actionable insights for building culturally reliable multimodal LLMs.

## 1 Introduction

Large language and vision–language models are being increasingly deployed across various cultures and languages. Yet, their behavior remains uneven; performance is strongest in high-resource, Western contexts and degrades in non-Western settings, particularly across Asia (Chiu et al., 2025; Romero et al., 2024; Vayani et al., 2025; Wang et al., 2024; Myung et al., 2025; Ng et al., 2025). As multimodal, multilingual models proliferate (Chen et al., 2024a; Bai et al., 2023; Jiang et al., 2023; OpenAI et al., 2024; Touvron et al., 2023; Romanou et al., 2024), a fundamental question arises: Does a model possess a coherent underlying understanding of a culture, or does it merely exhibit fragmented knowledge depending on the input modality and language? While recent works have probed cultural knowledge in text or images separately, they fail to capture the consistency of cultural awareness—defined here as the model's ability to provide stable, reasoned answers to semantically equivalent inputs regardless of whether the prompt is textual, visual, or spoken.

In this paper, we investigate (i) cultural awareness consistency, defined as the extent to which a model gives stable answers to semantically equivalent inputs when the representation (text, image+question, or spoken question) or the language changes; (ii) cultural awareness grounding, defined as whether correct answers rely on appropriate cultural signals rather than exploitable shortcuts; and (iii) cultural awareness generalization, defined as whether a model that has access to the relevant cultural knowledge can perform the required reasoning within those cultural contexts Balepur et al. (2024); Molfese et al. (2025); Zheng et al. (2023); Kreutzer et al. (2025). Nevertheless, conducting such evaluations presents significant challenges. Existing culture-centric datasets (e.g., Myung et al. (2025); Wang et al. (2024)) frequently suffer from two key limitations: (i) insufficient alignment of instances across modalities, (ii) inadequate representation of low-resource Asian languages. Furthermore, evaluation processes are easily hacked through memorization or elimina-

Table 1: **Comparison of existing culture-related benchmark datasets with MMA-Asia.** 📖, 🖼 and 🔊 denote the the text, image and speech modalities, respectively. "MLA" , "MDA" and "RI" denote "multilingual alignment", "multimodal alignment" and "Response interpretability", respectively.

| Benchmark | MLA | MDA | RI | # of Ctries | # of Langs | Modality | Multi-step reasoning | Question forms | Total samples | Total domains |
|---|---|---|---|---|---|---|---|---|---|---|
| SeaEval | ✗ | ✗ | ✗ | 4 | 1 | 📖 | ✓ | Diverse | 415 | - |
| CLIcK | ✗ | ✗ | ✗ | 1 | 2 | 📖 | - | Diverse | 1,995 | 11 |
| BLEnD | ✓ | ✗ | ✗ | 16 | 13 | 📖 | - | Fixed | 52,557 | 6 |
| Culturalbench | ✗ | ✗ | ✗ | 45 | 1 | 📖 | - | Fixed | 1,696 | 17 |
| CVQA | ✓ | ✗ | ✗ | 30 | 31 | 📖, 🖼 | - | Fixed | 10,000 | 10 |
| CulturalbenchVQA | ✗ | ✗ | ✗ | 11 | 1 | 📖, 🖼 | - | Diverse | 2,378 | 5 |
| ALM-bench | ✓ | ✗ | ✗ | 73 | 100 | 📖, 🖼 | - | Diverse | 22,763 | 19 |
| Md3 | ✗ | ✗ | ✗ | 3 | 1 | 🔊 | - | Diverse | 3,689 | 2 |
| MULTI-AUDIOJAIL | ✓ | ✗ | ✗ | 6 | 6 | 🔊 | - | Diverse | 102,720 | - |
| Ours | ✓ | ✓ | ✓ | 8 | 10 | 📖, 🖼, 🔊 | ✓ | Diverse | 27,000 | 9 |

tion in multiple choice questions (MCQs), which bypass the genuine reasoning capabilities (Wang et al., 2025; Hartmann et al., 2023; Liu et al., 2025; Yong et al., 2025). As a result, we still lack a principled way to separate actual cultural competence from artifacts.

To address this research gap, we introduce **MMA-ASIA**[1], an explainable evaluation framework for Asian cultural knowledge. MMA-ASIA aligns tri-modal items (textual question, image+question, and Text-to-Speech (TTS)-spoken question) with identical semantics and provides parallel local-language and English versions authored by native experts across 8 countries and 10 languages to make a comprehensive evaluation. The framework measures five axes: (1) cultural awareness disparity, (2) cross-modal consistency, (3) cross-lingual consistency, (4) cultural knowledge generalization under held-out regimes, and (5) grounding validation via targeted ablations and negative controls.

Using MMA-ASIA, we evaluate 15 multilingual and multimodal LLMs(e.g., GPT-4o, Qwen, Llama). We find that (i) accuracy drops markedly in low-resource Asian languages compared to English, (ii) cross-modal consistency lags text-only performance, indicating incomplete transfer from language to vision and speech, and (iii) grounding controls reduce a non-trivial fraction of apparent "wins," revealing shortcut use. We also analyze multi-step, culture-specific reasoning errors and where visual or linguistic cues fail to connect. We summarize our contributions as follows:

- **Aligned tri-modal, multilingual benchmark.** We release **MMA-ASIA** with 27,000 multilingual multimodal questions authored by in-country experts across 8 countries and 10 languages.
- **Five-axis evaluation protocol.** We formalize cultural awareness *consistency* (cross-modal/cross-lingual) and *grounding* with negative controls and ablations, plus generalization tests under held-out themes/countries. We provide reference implementations and CI-tested evaluation scripts.
- **Extensive baselines and analyses.** We report zero-shot baselines for 14 model families (multilingual LLMs and VLMs), including common-support subsets, and diagnose failure modes by modality, language, and reasoning step count.

## 2 RELATED WORK

**Cultural knowledge in text.** Recent benchmarks assess culture-specific knowledge via MCQs (Kim et al., 2024; Wang et al., 2024; Susanto et al., 2025; Myung et al., 2025; Chiu et al., 2025; Cho et al., 2025), consistently showing (i) performance gaps favoring English/high-resource settings and (ii) sensitivity to formatting. However, most lack aligned multimodal counterparts to test cross-modal cultural understanding.

**Cultural perception in images (VQA).** Multilingual VQA datasets use community-sourced images and questions (Romero et al., 2024; Nayak et al., 2024; Vayani et al., 2025), revealing vision-language gaps and language sensitivity. Yet they typically lack text/speech-parallel versions of identical items, making it hard to isolate whether failures stem from cultural knowledge, visual grounding, or language handling.

---

[1]We will release the data, splits, prompts, decoding settings, and per-item metadata (e.g., knowledge points, reasoning tags) for benchmarking, reproducing, and future extensions.

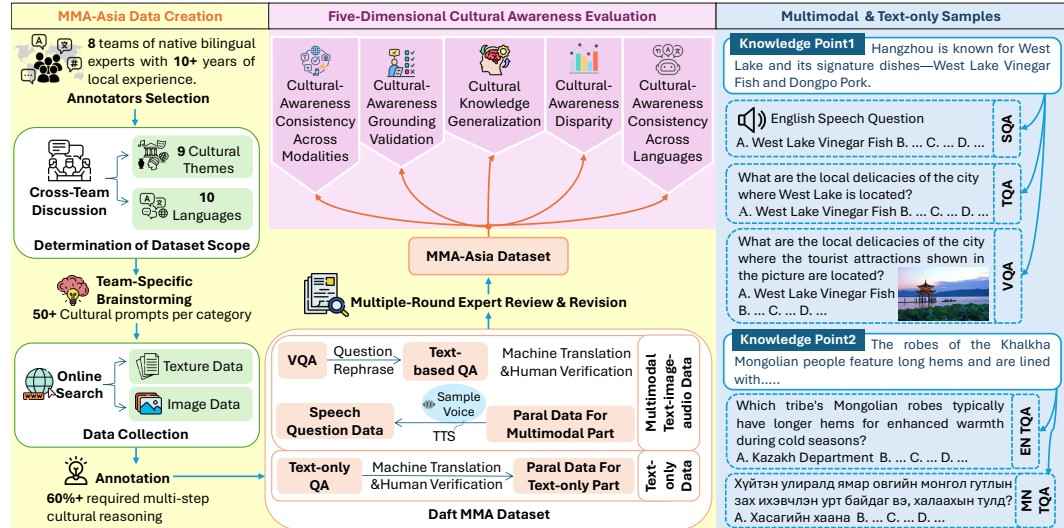

Figure 1: **An overview of the MMA-Asia evaluation framework:** data creation pipeline (yellow), representative dataset samples (pink), and evaluation dimensions (blue).

**Speech and accent robustness.** Speech datasets reveal substantial accent-related biases (Eisenstein et al., 2023; Roh et al., 2025), and perturbations can drastically change outcomes. However, they rarely evaluate cultural knowledge directly or analyze trimodal alignment and consistency.

**Consistency and grounding.** MCQ performance may reflect shortcuts rather than grounded reasoning (Balepur et al., 2024; Molfese et al., 2025; Zheng et al., 2023; Aakanksha et al., 2024; Kirk et al., 2024). Benchmarks seldom include negative controls or report cross-lingual/cross-modal consistency—essential for distinguishing cultural competence from artifacts.

**How is MMA-ASIA different from others?** Existing datasets evaluate cultural understanding within a single modality or language at a time, without tightly *aligning* instances across modalities and languages, and no built-in *grounding controls*. MMA-ASIA addresses these gaps by: (i) providing semantically aligned tri-modal items (text, image+question, speech) in parallel local-language and English versions; (ii) adopting *cross-modal* and *cross-lingual* consistency as prime metrics; and (iii) integrating targeted ablations and negative controls to test whether answers rely on the intended cultural signal rather than shortcuts. This design enables clearer attribution of failure modes: knowledge vs. language vs. modality, and more reliable measurement of cultural awareness in multimodal, multilingual models. We summarize key differences among representative datasets in Table 1.

## 3 BENCHMARK CONSTRUCTION

MMA-ASIA was collaboratively constructed by research teams from eight countries: China, Singapore, Japan, South Korea, Mongolia, Vietnam, Indonesia, and India. The pipeline comprised five stages: (i) annotator selection, (ii) selection of representative cultural themes and languages, (iii) collection of text and image materials, (iv) question authoring and annotation by country sub-teams, and (v) human review and revision for quality and cultural representativeness. For the definition of cultural themes, we followed the framework proposed by Adilazuarda et al. (2024).

### 3.1 ANNOTATOR SELECTION

An in-country expert team curated each national subset. All team members were native speakers of the local language and proficient in English. Annotators had lived in the respective country for more than ten years, ensuring deep familiarity with local cultural contexts. Detailed annotator information is in Appendix A.1. Before annotation began, we held project-wide briefings to explain the scope and requirements. We also distributed a detailed English annotation guideline (see Appendix A.3).

### 3.2 CULTURAL THEMES AND LANGUAGES

Based on established taxonomies from related work Romero et al. (2024); Myung et al. (2025), we constructed an initial set of cultural categories. Through rigorous selection criteria, we retained only highly representative categories, excluding those with insufficient cultural specificity (e.g., "Science and Technology") or negligible cross-national variation (e.g., "Plants and Ani-

mals"). Through collaborative discussions, we finalized 9 cultural themes, *Daily life habits/Culture, Food/Cuisine, Transportation, Buildings, History, Geographical location and climate, Education, Fashion/Clothing and Language/Race*. We balanced the number of questions across themes as much as possible. Each national subset includes the country's official language(s) and English. For India, we selected Hindi as the representative language due to its large speaker base among the 22 official languages. For Singapore, we included all four official languages: English, Chinese, Malay, and Tamil. In total, MMA-ASIA covers ten languages; full details are available in Appendix A.2.

### 3.3 TEXT AND IMAGE DATA COLLECTION

For both the text-only and multimodal tracks, teams generated at least 56 keywords or short phrases per category as *cultural prompts*. If a category could not supply enough prompts, the shortfall was filled using prompts from other categories. These prompts were designed to capture both diversity and geographic breadth, reducing the risk of homogenizing Asian cultures. Using these prompts, team members retrieved relevant texts and images from the web and extracted short passages to serve as the basis for question authoring. When source content was ambiguous, we cross-checked with multiple references to ensure authenticity. All images were obtained from Creative Commons (CC)-licensed resources (details in Appendix A.2).

### 3.4 QUESTION CREATION AND ANNOTATION

Teams used collected materials to create multi-choice QA data, requiring at least 60% of questions to involve multi-step cultural reasoning. Multi-step reasoning questions require sequential derivation and/or synthesis from multiple independent knowledge components, not just single-fact recall or paraphrase (detailed examples in Appendix A.2). Question templates are not fixed for preserving variety in question styles. Each national subset has two components: a *multimodal* and a *text-only* component. All QA data were authored in official local languages and translated to English using Claude 4 for Tamil and GPT-4o for other languages. All translations underwent manual verification; mistranslations were corrected, and for terms lacking standard English equivalents, we applied phonetic transliteration or adopted the locally prevalent rendering.

**Multimodal Component.** Annotators created VQA items where the correct answer requires visual understanding. Each VQA question was rephrased into a semantically-equivalent text-only MCQ. The answer options and the correct answer were kept unchanged. We provided both the original and rephrased items in English and the local language. We also generated speech inputs by converting the text to audio using high-quality TTS systems (Appendix A.4 provides TTS toolkit and speech data building details). For Spoken QA, we considered two configurations: (i) converting only the question stem to speech while keeping textual options, and (ii) converting both the stem and the options to speech. To preserve comparability with VQA under controlled variables and to reduce ambiguity introduced by fully spoken options, our main experiments adopt the "spoken stem + textual options" configuration across five evaluation dimensions. Results for the fully spoken setting (spoken stem and options) on the test set are reported in Appendix A.10 for reference. To reflect accent effects, we produced English speech in both accent-neutral and locally accented versions.

**Text-only Component.** This component contains questions that were not suitable for pairing with an image or are inherently text-based. We applied the same requirement that at least 60% of questions involve multi-step reasoning. All questions were created in multiple languages.

For each question, annotators additionally identified the requisite *knowledge points*. As shown in Fig. 1, these denote the minimal information necessary to arrive at the correct answer, usually summarized in a few concise sentences. These knowledge points are included with the dataset and support the evaluation of whether model explanations reflect culturally authentic reasoning.

For each question, annotators identified the requisite knowledge points, the minimal information needed for the correct answer, typically summarized in a few sentences (shown in Fig. 1). These knowledge points are used for model's cultural awareness grounding validation (Section 4.3).

### 3.5 HUMAN REVIEW AND REVISION

After each country team completed a draft, in-country linguists conducted quality reviews. The review covered: ambiguity in wording, accuracy of English translations, clarity and fluency of the speech data, completeness of knowledge points, and appropriateness of answer options. Teams revised their subsets based on this feedback, yielding the final high-quality release.

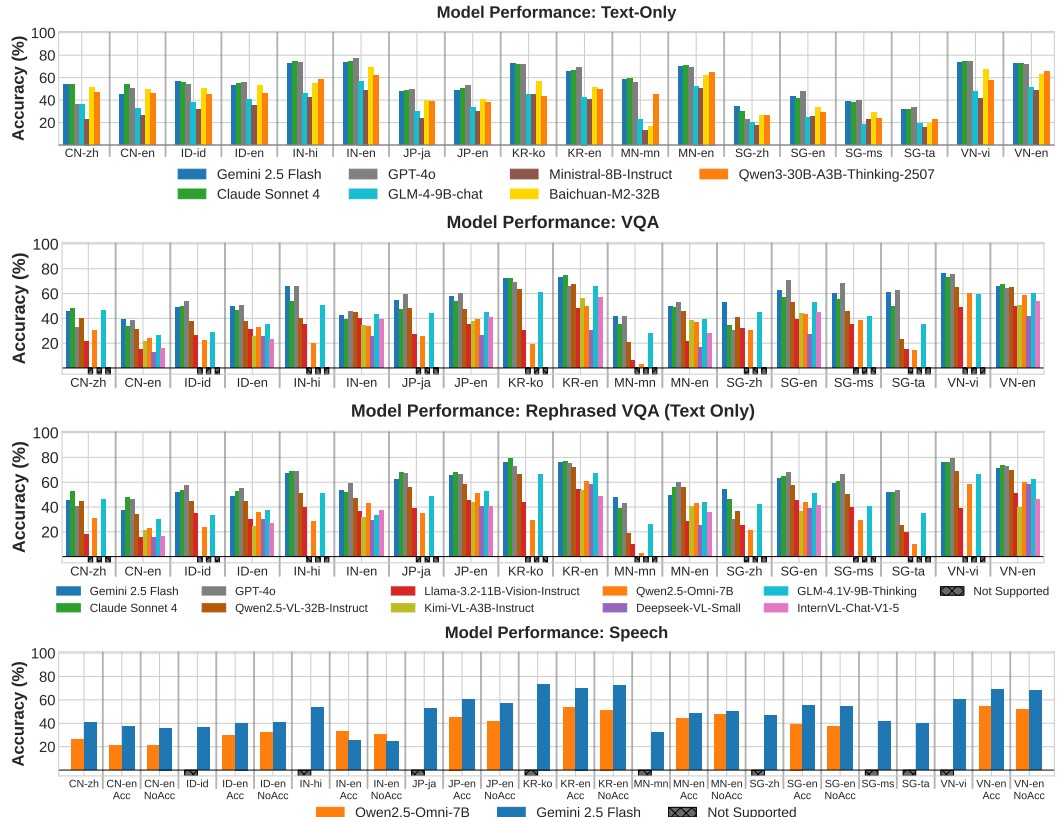

Figure 2: **Performance of LLMs on MMA-Asia across modalities.** Exact values are provided in Appendix A.8. For each country and modality, the dataset contains 500 questions presented in multiple languages. The vertical axis reports Accuracy (%), defined as the number of items where the model's chosen option exactly matches the correct option, divided by 500. The x-axis label {*Country*}–{*Language*} denotes the cultural dataset for {*Country*}, presented in {*Language*}.

## 4 EXPERIMENTS

We evaluate existing LLMs on the MMA-ASIA benchmark. Unless stated otherwise, all runs are *zero-shot* with a unified prompt template whose language matches the question language (prompts and experimental settings are in Appendix A.6). We report results along five dimensions and, for each, analyze the factors that drive performance. We access three closed-source models: GPT-4o (OpenAI et al., 2024), Claude-Sonnet-4, and Gemini 2.5 Pro (Team, 2025a), and eleven open-source multilingual or multimodal models, including the Qwen (Bai et al., 2025; Team, 2025b), LLaMA (Touvron et al., 2023), and GLM (GLM et al., 2024; Team et al., 2025c) families. Models and tasks are detailed in Appendix A.5. For models without multilingual support, we report English-only scores for comparability. For speech evaluation, we include only models that accept *speech tokens* directly; models that require intermediate automatic speech recognition (ASR) are excluded.

### 4.1 MAIN RESULTS

Figure 2 summarizes accuracy on MMA-ASIA. Nearly all state-of-the-art models score below 80% and most below 50%, highlighting the benchmark's difficulty. Closed-source models outperform open-source models; even the strongest open-source family (Qwen) trails the closed-source average by more than 10 percentage points. Performance varies with (i) the resource level of the language and (ii) the evaluation modality. The following subsections analyze: (1) cultural-awareness disparities across countries and languages, (2) cross-lingual and cross-modal consistency, (3) cultural-awareness grounding, and (4) cultural knowledge generalization.

### 4.2 CULTURAL AWARENESS DISPARITY

**Across countries (language factor controlled).** To isolate country effects, we compare each model's scores across countries within a fixed modality and, for each country, retain the model's

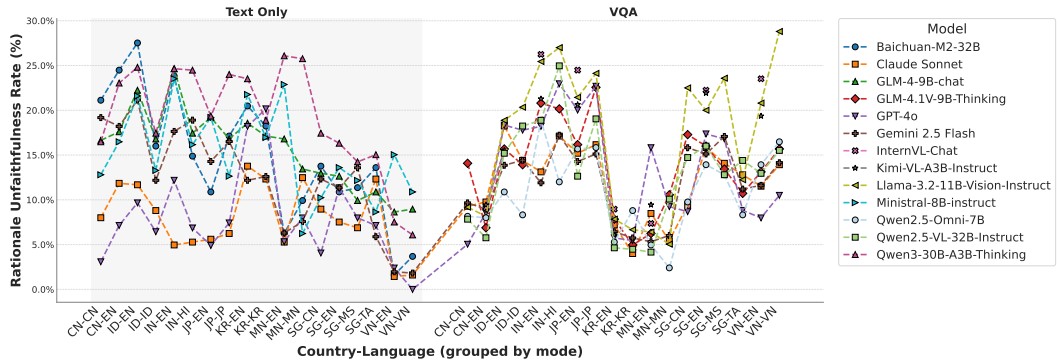

Figure 3: **Rationale Unfaithfulness Rates of LLMs across text-only and VQA.** Similar trends are observed for Rephrase VQA and Spoken QA; detailed results are provided in Appendix A.14.

*best* score over its available languages (Figure 2, Tables 13, 14, 15, 16). For example, in the text-only setting, GPT-4o scores 71.4% in Korean and 68.8% in English; we use the precision in English to represent the awareness of Korean culture in GPT-4o, to avoid confounding cultural competence with language proficiency. Models show higher awareness for Korean culture on average (63.98% across 4 modalities), plausibly due to global diffusion and richer data availability Jang et al. (2024); Dal Yong (2018). Remarkably, Vietnamese culture (62.96%) is on par with Korean, likely reflecting Vietnam's high social-media penetration (79.8% of the population) and thus large volumes of user-generated content (DataReportal et al., 2025). In contrast, China and India exhibit larger gaps, consistent with multilayered cultural forms and greater regional heterogeneity. Mongolia trails further, consistent with low-resource language settings and sparser training corpora.

**Across languages (within culture).** English prompts often outperform low-resource languages, reflecting the breadth of English corpora and limited cross-lingual transfer (Hu et al., 2025; Zheng et al., 2025b;a). This advantage diminishes or reverses for medium/high-resource local languages (e.g., Chinese, Japanese), where culture-specific terms and proper names are well represented locally but rare in English corpora, hurting retrieval and grounding. For example, "乌护 (Wuhu)," a lineage among Uyghur ancestral groups, lacks a standard English equivalent; transliteration is rare and ambiguous in English data. Thus, when the model is competent in the relevant local language, using that language can yield better cultural grounding than English.

**Across modalities (holding language fixed).** We observe a consistent ordering: *text-only > VQA > spoken QA*. Data availability follows the same order (text ≫ image–text ≫ raw speech). Speech adds uncertainty (noise, homophony), and many architectures encode modalities separately and then fuse downstream, introducing alignment/compression losses that widen gaps. Interestingly, in Speech, Qwen and Gemini outperform their standard English baselines in 6 and 5 country-specific cultural settings, respectively (Figure 2). Accents appear to serve as a prior cue for specific cultures, enhancing the models' accuracy on corresponding tasks. We attribute this to the co-occurrence of accents and their related cultural content within the data (see Appendix A.12 for detailed analysis). From the foregoing analysis, it is evident that LLMs in Asian cultural contexts also display cultural and modality biases shaped by data distributions; furthermore, given the limited effectiveness of cross-lingual cultural knowledge transfer, English cannot be assumed to perform reliably better on culture-related tasks. In contrast, in the speech modality, accents, often treated as noise, paradoxically serve as effective cultural cues that activate relevant context and improve performance. In speech, accents, typically considered noise, actually serve as effective cultural cues that activate relevant context and improve performance.
Meanwhile, further analysis of how different cultural categories affect model performance, as well as the relationship between cultural categories and language are provided in Sec A.7.

### 4.3 CULTURAL AWARENESS GROUNDING VALIDATION

MCQs are convenient but can be solved via shortcuts (e.g., option elimination) rather than grounded knowledge (Myung et al., 2025; Romero et al., 2024; Wang et al., 2024; 2025; Hartmann et al., 2023). We adopt two measures: (i) **Retained background knowledge:** Each item accompanied by its *knowledge points* (supporting evidence). (ii) **Explainable responses:** During testing, models must provide a textual rationale for their choice.

Table 2: Evaluated under the MCQ+Explanation metric, Text-only modality performance is reported as Accuracy (%), defined as the number of items where the model's choice exactly matches the correct option, divided by 500.

| Model | CN-en | ID-en | IN-en | JP-en | KR-en | MN-en | SG-en | VN-en |
|---|---|---|---|---|---|---|---|---|
| Gemini 2.5 Flash | 27.2 | 32.1 | 56.0 | 33.9 | 53.0 | 63.7 | 32.1 | 71.0 |
| Claude Sonnet 4 | 41.8 | 43.1 | 69.2 | 45.0 | 52.2 | 65.7 | 34.3 | 70.8 |
| GPT-4o | 43.1 | 46.1 | 65.0 | 48.3 | 50.6 | 64.1 | 36.3 | 69.4 |
| GLM-4-9B-chat | 15.0 | 27.2 | 32.4 | 14.4 | 24.2 | 35.8 | 12.2 | 42.4 |
| Mistral-8B-Instruct | 9.5 | 14.0 | 25.5 | 11.0 | 19.0 | 27.9 | 11.6 | 33.6 |
| Baichuan-M2-32B | 25.3 | 25.7 | 45.2 | 29.9 | 31.1 | 55.4 | 22.7 | 61.2 |
| Qwen3-30B-A3B | 22.8 | 21.4 | 37.6 | 18.6 | 26.3 | 38.9 | 12.9 | 58.3 |

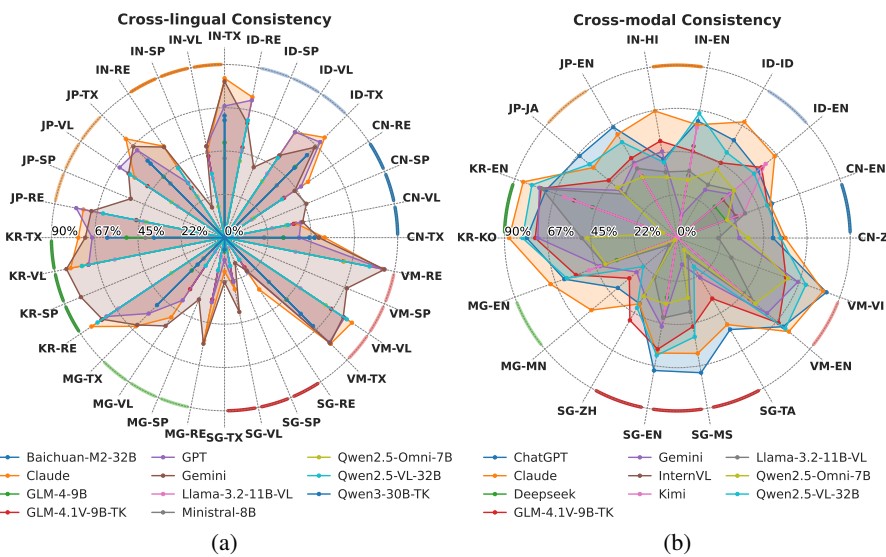

(a)                    (b)

Figure 4: **(a) Cross-lingual consistency with fixed country and modality and (b) cross-modal consistency with fixed language and country.** TX/VL/RE/SP represent text QA, visual QA, rephrase QA, and speech QA.

We use a LLM-as-Judge approach to verify whether, *given a correct answer*, the model's explanation matches the item's knowledge points. To reduce variability across judges, we require explanations in English. The human consistency checks, the multimodel consistency evaluation, and all LLM-as-Judge parameter settings are described in Appendix A.13. Figure 3 reports the *Rationale Unfaithfulness Rate* (RUR) for text-only and VQA items, defined as the proportion of correct answers whose explanations contradict or omit the required knowledge. Despite strong overall accuracy, proprietary models (Claude, GPT, and Gemini) still show RUR values between 5% and 20%. The issue is more pronounced for open-source models: Llama-3.2-11B-Vision-Instruct and Qwen3-30B-A3B-Thinking have the highest RUR, with Llama's rate particularly elevated on non-English inputs. Qwen3-30B-A3B-Thinking often produces long explanations that contain hallucinations and sometimes derives the correct option from premises that contradict the ground truth, indicating reliance on generic heuristics rather than culturally grounded reasoning.

These results suggest that MCQs alone do not reliably measure cultural understanding (especially for open-source models) and should be augmented with culture-grounded verification, including knowledge-point checks and rationale assessment. Table 2 reports the Text-Only results of the MCQ+Explanation evaluation, where model performance is assessed jointly on the selected options and the generated explanations. This explanation-based adjudication removes spurious correct answers and yields a more faithful estimate of the models' cultural awareness. The results for the other modalities are provided in Sec. A.9.

## 4.4 CULTURAL-AWARENESS CONSISTENCY ACROSS LANGUAGES

Cross-lingual cultural-awareness consistency is defined as the degree to which a model gives consistent outputs to semantically equivalent prompts posed in different languages, irrespective of answer correctness, which can be expressed as Eq. 1 (Wang et al., 2024).

Figure 5: **Attention heatmap visualization over image regions during incorrect model answers.** Color scale from ■ blue (low) to ■ red (high) indicates increasing model attention.

$$\text{Consistency}_s = \frac{1}{N} \sum_{i=1}^{N} \frac{1}{\binom{m_i}{s}} \sum_{S \subseteq L_i, |S| = s} \mathbf{1}\left(\left|\{a_i^\ell : \ell \in S\}\right| = 1\right). \tag{1}$$

where $L_i$ denotes the set of languages available for question $i$, and $m_i = |L_i|$ is its cardinality; $S \subseteq L_i$ with $|S| = s$ denotes any size-$s$ language subset; $a_i^\ell$ is the model's answer to question $i$ when prompted in language $\ell$; $\mathbf{1}(\cdot)$ is the indicator function; and $\left|\{a_i^\ell : \ell \in S\}\right| = 1$ asserts that all answers within $S$ are identical. If $m_i < s$, there are no valid subsets and item $i$ contributes zero.

As shown in Figure 4a, contemporary multilingual LLMs exhibit weak cross-lingual consistency on culturally grounded tasks in both text-only and VQA settings. The weakness is most evident for language pairs with large resource gaps. For Mongolian culture, the disparity between Mongolian and English yields only 65.2% consistency for Claude, while all open-source models remain below 50%. By contrast, Korean culture shows higher consistency, plausibly reflecting the global diffusion of contemporary Korean media and the resulting multilingual exposure to related knowledge (Jang et al., 2024; Dal Yong, 2018). Consistency also declines sharply as the number of evaluated languages increases. For Singapore-related items, when Chinese, English, Tamil, and Malay are assessed jointly, the maximum consistency does not exceed 45% (Gemini on VQA), despite relatively high pairwise values of 60.60% (EN–TA), 64.20% (EN–MS), and 55.20% (EN–ZH). In some culturally challenging cases, visual context can partially bridge languages: for Indian culture, GLM-4.1 achieves 44.20% cross-lingual consistency on Hindi VQA, which is 13.2 points higher than its rephrased text-only counterpart, although both remain low.

From these observations, we observe that the consistency between languages depends on the data resources and cultural exposure. Resource asymmetry degrades consistency, whereas cultural prominence helps. Consistency decays nonlinearly as more languages are considered, and strong pairwise agreement does not guarantee multi-language coherence. Visual cues can narrow gaps in certain settings, but are insufficient to overcome the structural limitations of low-resource languages.

### 4.5 Cultural-Awareness Consistency Across Modalities

Cross-modal cultural-awareness consistency evaluates whether a model gives the same output for semantically equivalent queries presented in different modalities. Figure 4b shows that, across the eight Asian countries, the pattern largely matches the cross-lingual case: averaged over models, cross-lingual consistency is 48%, and cross-modal consistency rarely exceeds 67%. This gap indicates asymmetric transfer of cultural knowledge across modalities. Under low-resource language settings, almost all models struggle to maintain stable cross-modal answers. Within the same national context, medium- to high-resource local languages typically yield higher cross-modal consistency than English. To examine the observed ordering *text-only > VQA > spoken QA* (see Section 4.2), we conduct a detailed error analysis with Qwen2.5-VL-32B-Instruct. Because spoken QA adds additional complexities (noise, accents, intonation), our analysis in this section focuses on VQA versus text-only performance; we leave a fuller study of speech to future work.

We specifically isolate instances where the model succeeded with pure text input but failed in the VQA context. Apart from the most common errors arising from a lack of understanding of culture-related image contexts, our analysis reveals two additional predominant categories of errors:

**The pitfall of prompt-guided selective attention.** Models often tend to focus predominantly on explicitly mentioned objects in prompts, whereas cultural VQA requires a more nuanced ability to identify culture-specific visual cues within images. To validate whether the models' focuses are truly culture-specific, we extract visual evidence using answer-conditioned multi-layer Grad-CAM (Selvaraju et al., 2019), interpolating and mapping the resulting heatmaps back to the original image for visualization. Specifically, given an image $v$, a textual prompt $x$ and an answer $y_{a:b} = (y_a, \ldots, y_b)$ autoregressively produced by a model with parameters $\theta$, we define an answer-conditioned objective on the log-likelihood of the answer tokens as Eq. 2. Here we use the token-sum objective. For mem-

**Question:** In the official history, which of the following generals was killed by the person in the picture? A. Hua Xiong B. Yan Liang C. Wen Chou D. Yu Jin

**Baseline Output:** The image depicts a statue of Guan Yu, one of the most famous and revered generals from the Three Kingdoms period in Chinese history...Hua Xiong: This is a key figure in the story of the "Battle of Changban" (or "Battle of Changping"). According to historical records and the novel Romance of the Three Kingdoms, Guan Yu famously defeated Hua Xiong in battle... but this happened during the Battle of Xiapi, which came after the Battle of Changban. D. Yu Jin: Yu Jin was a general of Cao Wei who surrendered to Guan Yu during the Battle of Fancheng. He was not killed by Guan Yu. Final Answer: The person in the picture (Guan Yu) is most famously known for killing Hua Xiong. Therefore, the correct answer is: **A.**

**Vision-ablated Prefix Replay:** [Replay Prefix]: The image depicts a statue of Guan Yu ---- Full result ---- The image depicts a statue of Guan Yu...Hua Xiong (华雄):Hua Xiong was a general serving under Dong Zhuo during the late Eastern Han dynasty. He was killed by Guan Yu in the Battle of Hulao Gate, as described in the Romance of the Three Kingdoms...Yan Liang (颜良):Yan Liang was a general serving under Yuan Shao. He was killed by Guan Yu during the Battle of Baima (白马之战) in 200 CE... Wen Chou (文丑):Wen Chou was another general serving under Yuan Shao. He was also killed by Guan Yu shortly after the Battle of Baima, during the same campaign against Yuan Shao. Yu Jin (于禁)... He was not killed by Guan Yu... Correct Answer: Based on the historical context and the question, the general who was killed by Guan Yu is Yan Liang. Final Answer: **B.**

Figure 6: **Comparison of baseline vs. "VPR" outputs, with hallucinations highlighted in green.**

ory efficiency, only the visual tower is set to require gradients. We denote the forward activation of the $\ell$-th visual block as $A(\ell) \in \mathbb{R}^{C \times H \times W}$, the gradient of the block is $\mathbf{G}^{(\ell)}$. A per-layer Grad-CAM is built via channelwise inner product followed by $\mathrm{ReLU}$ (Eq. 3).

$$J_{\text{sum}} = -\sum\nolimits_{t=a}^{b} \log p_\theta(y_t \mid v, x, y_{<t}) \quad (2) \qquad \mathbf{M}^{(\ell)} = \mathrm{ReLU}\Big(\sum\nolimits_{c=1}^{C} \mathbf{G}_c^{(\ell)} \odot \mathbf{A}_c^{(\ell)}\Big) \in \mathbb{R}^{H \times W} \quad (3)$$

After lightweight smoothing and normalization of $\mathbf{M}(\ell)$, the resulting $\hat{M}^{(\ell)}$ then undergoes cross-layer aggregation and bilinear interpolation upsampling on the $k$ blocks at the end of the visual tower, as defined by Eqs. 4 and 5, to the original image resolution. For more robust results, we specifically choose the last 3 blocks and do cross-layer aggregation by averaging them.

$$\mathbf{M} = \mathrm{Agg}_{\ell \in \mathcal{L}}\big(\hat{\mathbf{M}}^{(\ell)}\big) \quad (4) \qquad \tilde{\mathbf{M}} = \mathrm{Bilinear}(\mathbf{M}, H_{\text{img}}, W_{\text{img}}) \in [0, 1]^{H_{\text{img}} \times W_{\text{img}}}. \quad (5)$$

This phenomenon is clearly revealed by the heatmaps in Figure 5, showing how the model focuses on subjects explicitly mentioned in the prompt when answering questions. However, this selective attention can unfortunately lead the model to overlook other critically important local details within the image, consequently resulting in erroneous inferences. In Case 1 (Fig. 5), when the model is presented with the question "Which of the following figures is the master of the person shown in the picture?" in Chinese, its attention is predominantly drawn to the figure on the left side of the image, who is explicitly referenced in the prompt. However, the model overlooks the Howling Celestial Dog in the upper-right corner of the image, a crucial clue for identifying Erlang Shen. Additional attention-visualization examples and analyses are given in Appendix A.15.

**Image tokens contribute to reasoning hallucinations.** We find that visual content increases reasoning hallucinations in models compared to text-only QA, despite accurate image recognition capabilities. In Figure 6, while the baseline model correctly identifies "Guan Yu", it still produces multiple reasoning hallucinations (highlighted in green). However, text-only Rephrase VQA queries show no such hallucinations (Appendix A.11), indicating that reasoning errors likely originate from image tokens and suggest modality-specific bias in multimodal reasoning. To test this hypothesis, we propose Vision-ablated Prefix Replay (VPR), which generates image descriptions then removes visual conditions while maintaining fixed prefixes for subsequent reasoning (details in Appendix A.11). In Figure 6, VPR conditions the model's reasoning on the generated caption 'The image depicts a statue of Guan Yu' while discarding visual tokens, eliminating hallucinations and producing the correct answer. Furthermore, we select 50 questions where VQA answers contained hallucinations but corresponding text-only queries were correct. VPR eliminates hallucinations and produced correct answers for 19 of these cases (38%), supporting our hypothesis.

Multimodal models demonstrate significant inconsistency in cultural awareness across modalities, indicating flawed cultural knowledge transfer. In VQA, this deficiency stems from two core issues: "*selective attention pitfall*" where models over-focus on text-prompted subjects while missing key visual cues, and visual token-induced "*reasoning hallucinations*".

### 4.6 Cultural Knowledge Generalization

Prior work (Balepur et al., 2024; Molfese et al., 2025; Zheng et al., 2023) suggests that scaling increases LLMs' factual memory but not genuine logical generalization. To distinguish whether cultural multi-step reasoning errors stem from knowledge gaps or generalization deficits, we conduct a deconstruction study. We decompose each question into atomic sub-questions testing single knowledge points, which models answer first (see Table 23 for an example). We then evaluate the original question under two conditions: (i) with in-context "sub-question $\rightarrow$ model answer" pairs, and (ii) from scratch. If models solve all subquestions but fail the original question from scratch, they possess the knowledge but cannot transfer it, indicating generalization deficits. If they err on sub-questions, failure likely reflects missing culture-specific knowledge.

Table 3: Cultural knowledge generalization performance of different models under different language settings. We define Successful Correction as correct sub-questions and final answers; Integration Failure as incorrect final synthesis despite correct sub-questions; and Sub-question Failure as errors in the initial reasoning stage.

| Language | Model | Successful Correction (%) | Integration Failure (%) | Sub-question Failure (%) |
|---|---|---|---|---|
| **English** | Claude | 35.3 | 2.6 | 62.1 |
| | Qwen | 18.5 | 10.7 | 70.8 |
| | Mistral | 18.9 | 1.4 | 80.7 |
| | GLM | 30.9 | 2.0 | 67.1 |
| **Chinese** | Claude | 41.1 | 3.5 | 55.4 |
| | Qwen | 35.5 | 2.6 | 61.9 |
| | Mistral | 25.2 | 1.0 | 73.8 |
| | GLM | 40.5 | 2.8 | 56.7 |
| **Indonesian** | Claude | 20.2 | 9.2 | 70.6 |
| | Qwen | 13.4 | 3.7 | 82.9 |
| | Mistral | 4.4 | 1.1 | 94.5 |
| | GLM | 6.5 | 5.3 | 88.2 |

We investigate the failure modes of Claude Sonnet 4, GLM-4-9B-chat, Mistral-8B-Instruct, and Qwen3-30B-A3B-Thinking on multi-step reasoning tasks across English, Chinese, and Indonesian. Table 10 summarizes the number of failure cases for each model in each language. Our experiment utilizes problems where each model initially failed, decomposing them into 3–8 atomic sub-questions with ground-truth answers. An LLM (GPT-4o) judges the correctness of each step. The results in Table 3 reveal distinct failure patterns influenced significantly by language resource availability and model architecture.

**Knowledge Availability vs. Integration:** In high-resource languages, errors primarily stem from integration failures rather than knowledge deficits. Post-decomposition, Claude and GLM achieve successful correction rates of 41.1% and 40.5% in Chinese, demonstrating adequate cultural knowledge but impaired single-step activation. Conversely, Indonesian performance is bottlenecked by fundamental knowledge gaps, with sub-question failure rates reaching 94.5% (Mistral) and 88.2% (GLM), confirming that decomposition cannot remedy training data deficiencies.

**The "Lost in Integration" Phenomenon:** Beyond knowledge gaps, integration failures, where models answer sub-questions correctly but fail final synthesis, constitute a significant error source. This is pronounced in Qwen's English performance: despite reasonable sub-question accuracy, integration failure reaches 10.7% (versus ∼2% for GLM and Mistral), showing reasoning misalignment where isolated fact retrieval succeeds but cross-lingual synthesis fails. Claude exhibits similar patterns in Indonesian (9.2% integration failure), indicating that logical synthesis degrades faster than fact recall in lower-resource languages. Cross-lingual cultural knowledge transfer remains challenging regardless of model scale. In high-resource settings, closed-source models (Claude) demonstrate broad knowledge coverage but limited cultural generalization, while open-source models face knowledge gaps and poor transferability. In low-resource languages, closed-source models show greater degradation in cultural generalization, whereas knowledge deficits account for over 88.5% of open-source model failures, underscoring their inadequate low-resource language training.

## 5 CONCLUSION

We introduced MMA-ASIA, a tri-modal (text, image, speech), multilingual benchmark and framework for evaluating cultural awareness in LLMs across 8 Asian countries and 10 languages. Our contributions include an aligned, human-curated dataset with substantial multi-step reasoning, a five-dimensional protocol that measures accuracy, cross-lingual and cross-modal consistency, cultural knowledge generalization, and grounding validity, and analysis tools that reveal shortcut use. Results show persistent data-driven cultural bias, uneven cross-lingual transfer, and fragile multi-modal reasoning (selective visual attention and image-induced hallucinations). At the same time, accented speech can act as a useful cultural cue. Models also struggle to integrate known facts into multi-step reasoning, indicating a generalization bottleneck. We argue for consistency- and grounding-aware evaluation, as well as methods that strengthen cross-modal alignment and broaden high-quality coverage in low-resource languages. MMA-ASIA provides data, protocols, and baselines to track progress toward culturally reliable multimodal LLMs.

## ICLR ETHICS STATEMENT

### CONTRIBUTE TO SOCIETY AND TO HUMAN WELL-BEING

- We acknowledge that people worldwide are stakeholders in computing, and we will use our skills for the benefit of society, its members, and the natural environment.

- We strive to minimise negative consequences, including threats to health, safety, personal security, and privacy. Our dataset is created solely from cultural facts and excludes any negative or false information.

- When the interests of multiple groups conflict, we give increased attention and priority to the needs of those who are less advantaged.

- In creating the dataset, we deliberately account for cultural diversity and base all content strictly on verifiable cultural facts.

### UPHOLD HIGH STANDARDS OF SCIENTIFIC EXCELLENCE

- We are committed to open enquiry, intellectual rigour, integrity, and collaboration.

- We report accurately and honestly. We do not make false or misleading claims, fabricate or falsify data, or misrepresent results. Methods and results are presented in a transparent and reproducible manner.

- All content is developed from publicly available data; no personal private information is collected or disclosed. This work involves no interventions on human subjects and poses no risk of harm; accordingly, ethical approval from an institutional review board was not required.

- We acknowledge all contributions to the research and comply with agreements related to intellectual property, publication, and authorship.

### AVOID HARM

- Our questions are created from publicly available, verifiable cultural facts. The dataset contains no pornographic or violent content, nor any racist or discriminatory material.

### BE HONEST, TRUSTWORTHY, AND TRANSPARENT

- We describe the professional backgrounds of contributors in Section A, and all researchers adhere to principles of honesty, trustworthiness, and transparency.

### BE FAIR AND TAKE ACTION NOT TO DISCRIMINATE

- Dataset creators are required to avoid content that is racist or contains hate speech. Each contributor's work is assessed fairly.

### RESPECT THE WORK REQUIRED TO PRODUCE NEW IDEAS AND ARTEFACTS

- We properly cite all prior research, tools, models, and commercial software that we use or build upon. We treat all project members with respect and equity.

- All images are used under appropriate Creative Commons licences and solely for research purposes; any subsequent release will strictly follow the relevant licence terms.

- Except for the image subset, all dataset content is authored by our team, and no infringement is involved.

### RESPECT PRIVACY

- The dataset content is derived from publicly available and verifiable cultural facts.

- We apply post-processing to images to protect any potential personal privacy.

- We do not disclose any private information of team members or contributors.

### HONOUR CONFIDENTIALITY

- This work is not subject to any non-disclosure agreements. Upon paper acceptance, we will openly release the dataset.

## REPRODUCIBILITY STATEMENT

DID YOU REPORT THE NUMBER OF PARAMETERS IN THE MODELS USED, THE TOTAL COMPUTATIONAL BUDGET (E.G., GPU HOURS), AND COMPUTING INFRASTRUCTURE USED?

Yes, all this information can be found in Appendix A.5, A.6, A.4.

DID YOU DISCUSS THE EXPERIMENTAL SETUP, INCLUDING HYPERPARAMETER SEARCH AND BEST-FOUND HYPERPARAMETER VALUES?

Yes, all this information can be found in Appendix A.6.

DID YOU REPORT DESCRIPTIVE STATISTICS ABOUT YOUR RESULTS (E.G., ERROR BARS AROUND RESULTS, SUMMARY STATISTICS FROM SETS OF EXPERIMENTS), AND IS IT TRANSPARENT WHETHER YOU ARE REPORTING THE MAX, MEAN, ETC. OR JUST A SINGLE RUN?

Yes, it can be found in Section 4 and Appendix A.6, A.8, A.11, A.15, A.10 in the paper.

IF YOU USED EXISTING PACKAGES (E.G., FOR PREPROCESSING, FOR NORMALIZATION, OR FOR EVALUATION, SUCH AS NLTK, SPACY, ROUGE, ETC.), DID YOU REPORT THE IMPLEMENTATION, MODEL, AND PARAMETER SETTINGS USED?

Yes, you can find it in the Appendix A.4, A.6.

## DID YOU USE HUMAN ANNOTATORS (E.G., CROWDWORKERS) OR RESEARCH WITH HUMAN PARTICIPANTS?

Yes.

DID YOU REPORT THE FULL TEXT OF INSTRUCTIONS GIVEN TO PARTICIPANTS, INCLUDING E.G., SCREENSHOTS, DISCLAIMERS OF ANY RISKS TO PARTICIPANTS OR ANNOTATORS, ETC.?

Yes, you can find it in the Appendix A.3.

DID YOU REPORT INFORMATION ABOUT HOW YOU RECRUITED (E.G., CROWDSOURCING PLATFORM, STUDENTS) AND PAID PARTICIPANTS, AND DISCUSS IF SUCH PAYMENT IS ADEQUATE GIVEN THE PARTICIPANTS' DEMOGRAPHIC (E.G., COUNTRY OF RESIDENCE)?

Yes. you can find it in the Appendix A.1.

DID YOU DISCUSS WHETHER AND HOW CONSENT WAS OBTAINED FROM PEOPLE WHOSE DATA YOU'RE USING/CURATING? FOR EXAMPLE, IF YOU COLLECTED DATA VIA CROWDSOURCING, DID YOUR INSTRUCTIONS TO CROWDWORKERS EXPLAIN HOW THE DATA WOULD BE USED?

Yes. The purpose of the dataset for research was clearly defined at the beginning of the paper.

DID YOU REPORT THE BASIC DEMOGRAPHIC AND GEOGRAPHIC CHARACTERISTICS OF THE ANNOTATOR POPULATION THAT IS THE SOURCE OF THE DATA?

Yes, you can find it in Appendix A.1.

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

# A APPENDIX

## A.1 ANNOTATOR DEMOGRAPHIC

Our annotation team comprises members from eight different countries. All team members are native speakers of their local languages and proficient in English, with professional backgrounds in natural language processing or speech processing. Each annotator has lived in the respective country for more than ten years. Table 4 presents the annotators' details and professional backgrounds; to protect privacy, we replace personal names with numeric identifiers within each country. After each team completes the first round of annotations, in-country linguistic experts conduct a data review. Table 5 lists the language experts' information; likewise, we anonymize personal names.

## A.2 DATA STATISTICS

Under the MMA-ASIA framework, the dataset covers 8 countries and 10 languages, with each country's split presented in both English and its local language, totaling 27,000 questions. Table 6 lists the countries included in our dataset and their corresponding local language(s). Over 79% of all items are multi-step cultural reasoning questions. We define a multi-step cultural reasoning item as one whose solution requires sequential derivation and/or synthesis from at least two independent knowledge components, rather than mere recall or paraphrase of a single cultural fact. Table 7 presents an example multi-step question with its analysis. The proportion of multi-step items by country and modality is shown in Figure 8. The dataset spans nine categories—*Daily Life/Culture, Food/Cuisine, Transportation, Buildings, History, Geographical Location & Climate, Education, Fashion/Clothing, and Language/Ethnicity*—with per-country category distributions summarized in Figure 7.

The following outlines the potential contents included in each category.

- **Daily Life/Culture**: Covers everyday etiquette, customs, and values—greeting practices, daily routines and social decorum, family and community interactions, as well as festivals and folk practices. It may also include consumption and leisure preferences, common life scenarios, and behavioral norms.

- **Food/Cuisine**: Regional cuisines, representative dishes and ingredients, cooking techniques, dietary taboos and table manners, utensils, and dining settings. Also includes festive foods, street snacks, and regional taste differences.

Table 4: Data Annotator Demographics and Skills

| Country | ID | Gender | Age | Education | English Prof. | Local Lang. Prof. | Professional Background |
|---|---|---|---|---|---|---|---|
| China | 1 | Male | 28 | Ph.D. | Proficient | Proficient | NLP |
| | 2 | Male | 25 | Master Deg. | Proficient | Proficient | NLP |
| | 3 | Male | 30 | Ph.D. | Proficient | Proficient | NLP |
| | 4 | Male | 25 | Master Deg. | Familiar | Fluent | NLP |
| | 5 | Male | 24 | Master Deg. | Familiar | Proficient | NLP |
| Japan | 1 | Male | 23 | Master Deg. | Proficient | Proficient | SP |
| | 2 | Male | 35 | Ph.D. | Fluent | Proficient | SP |
| Mongolia | 1 | Male | 26 | Ph.D. | Proficient | Proficient | SP |
| | 2 | Male | 25 | Master Deg. | Proficient | Proficient | SP |
| Korea | 1 | Male | 31 | Ph.D. | Proficient | Proficient | NLP |
| | 2 | Male | 26 | Master Deg. | Proficient | Proficient | NLP |
| | 3 | Female | 25 | Master Deg. | Proficient | Proficient | NLP |
| India | 1 | Male | 25 | Bachelor's Deg. | Proficient | Proficient | NLP |
| | 2 | Male | 31 | Ph.D. | Proficient | Proficient | NLP |
| Vietnam | 1 | Male | 21 | Bachelor's Deg. | Proficient | Proficient | SP |
| | 2 | Male | 20 | Bachelor's Deg. | Proficient | Proficient | SP |
| | 3 | Male | 21 | Bachelor's Deg. | Proficient | Proficient | SP |
| | 4 | Male | 21 | Bachelor's Deg. | Proficient | Proficient | SP |
| | 5 | Male | 29 | Master Deg. | Proficient | Proficient | SP |
| Indonesia | 1 | Female | 21 | Bachelor's Deg. | Proficient | Proficient | NLP |
| | 2 | Female | 26 | Ph.D. | Proficient | Proficient | NLP |
| Singapore | 1 | Female | 26 | Ph.D. | Proficient | Proficient | NLP |
| | 2 | Female | 18 | Bachelor's Deg. | Proficient | Proficient | NLP |
| | 3 | Male | 18 | Bachelor's Deg. | Proficient | Proficient | NLP |
| | 4 | Male | 21 | Bachelor's Deg. | Proficient | Proficient | NLP |
| | 5 | Female | 28 | Ph.D. | Proficient | Proficient | SP |

*Note: NLP: Natural Language Processing; SP: Speech Processing.*

Table 5: Data Reviewer Demographics and Skills

| Country | ID | Gender | Age | Education | English Prof. | Local Lang. Prof. |
|---|---|---|---|---|---|---|
| China | 1 | Female | 30 | Master Deg. | Proficient | Proficient |
| Japan | 1 | Female | 32 | Master Deg. | Proficient | Proficient |
| Mongolia | 1 | Male | 25 | Master Deg. | Proficient | Proficient |
| Korea | 1 | Male | 54 | Bachelor's Deg. | Proficient | Proficient |
| India | 1 | Male | 28 | Master Deg. | Proficient | Proficient |
| Vietnam | 1 | Male | 27 | Master Deg. | Proficient | Proficient |
| Indonesia | 1 | Female | 28 | Bachelor's Deg. | Proficient | Proficient |
| Singapore | 1 | Female | 26 | Bachelor's Deg. | Proficient | Proficient |
| | 2 | Female | 22 | Bachelor's Deg. | Proficient | Proficient |
| | 3 | Male | 30 | Ph.D. | Proficient | Proficient |
| | 4 | Male | 26 | Ph.D. | Proficient | Proficient |

Table 6: Local Languages by Country

| Country | Local Language |
| --- | --- |
| China | Chinese |
| Singapore | English, Chinese, Malay, Tamil |
| Japan | Japanese |
| Korea | Korean |
| India | Hindi |
| Indonesia | Indonesian |
| Vietnam | Vietnamese |
| Mongolia | Mongolian |

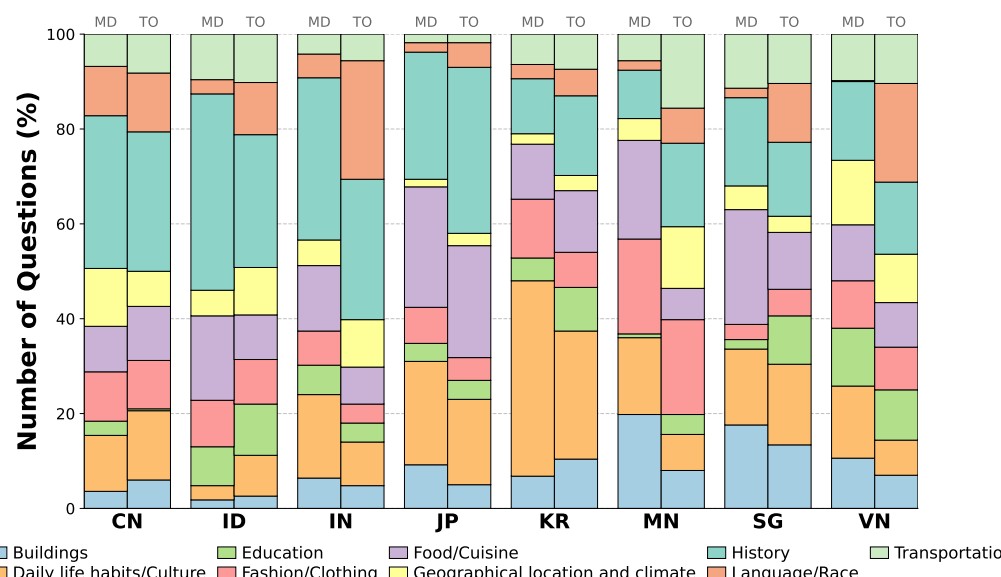

Figure 7: Distribution of question categories across countries and modalities

- **Transportation**: Transportation across historical periods and the evolution of vehicles, along with regional differences in modes of transport. May also cover landmark transit systems and commuting culture.

- **Buildings**: Traditional and modern architectural styles; religious and public buildings; housing forms and materials; city skylines and the preservation of historic districts. Can also address symbolic meanings in architecture and region-specific structural features.

- **History**: Major historical periods and events, notable figures and heritage sites, and how historical memory shapes contemporary society and culture. May also include colonial/independence histories and cultural change driven by migration and war.

- **Geographical Location & Climate**: Landforms and terrain, climate zones and seasonal variation, natural resources, and ecosystems. Extends to lifestyle, clothing, and dietary adaptations shaped by geography and climate.

- **Education**: Renowned national works of literature, art, or music, as well as the structure of the education system and pathways to advancement, including stories surrounding prestigious institutions.

- **Fashion/Clothing**: Traditional attire and its ceremonial contexts; modern dress styles and aesthetic trends; accessories and color preferences; occupational/school uniforms and seasonal clothing. May also discuss cultural symbolism embedded in garments.

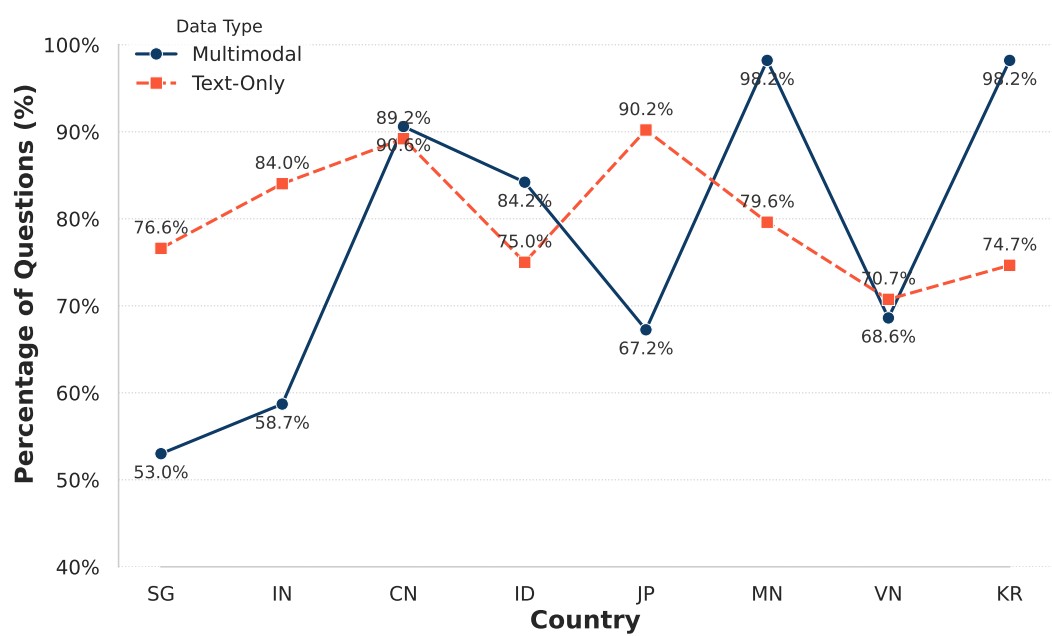

Figure 8: Multi-step reasoning question proportions across countries

- **Language/Ethnicity**: Code-switching (where applicable); official and commonly used languages; dialects and accent features; writing systems and naming conventions; multi-ethnic compositions and cultural practices. Also includes politeness strategies in language and norms of cross-group communication.

Table 7: An example of multi-step reasoning question

| Question | An ancient tower became famous due to a poem by the Tang Dynasty poet Cui Hao. In which period was it proposed to be constructed with iron materials? *(A) 16th year of Guangxu, Qing Dynasty    (B) 1st year of Jiaqing, Qing Dynasty    (C) 7th year of Tongzhi, Qing Dynasty    (D) 8th year of the Republic of China* |
|---|---|
| Reasoning Decomposition | **Step 1: Identify the tower.** The poem is Cui Hao's "Yellow Crane Tower," so the tower is *Yellow Crane Tower*. **Step 2: Recall historical events.** In the 10th year of Guangxu (1884) the tower was destroyed by fire. In the 16th year of Guangxu (1890), Zhang Zhidong (Governor-General of Hubei and Hunan) first proposed rebuilding the tower using iron materials. **Step 3: Match with the options.** A (1890): **Correct**, matches the historical fact. B (1796): Incorrect, too early. C (1868): Incorrect, before the fire. D (1919): Incorrect, after the Qing Dynasty and not the first proposal. **Final Answer: A. 16th year of Guangxu, Qing Dynasty**. |

Figure 9 and Figure 10 show some samples of our dataset.

A.3 ANNOTATION GUIDELINE

The content of the guideline distributed to annotators is shown in Figure 11. To minimize heuristic cues arising from non-cultural knowledge, we add a consistency constraint on distractors: they must belong to the same category as the correct option and closely resemble it in observable attributes and semantic representation. We also encourage each team to uncover cultural elements unique to their own country, rather than focusing only on widely known aspects. For the Language category, if code-switching is prevalent in the annotators' country, we strongly encourage including such language-assessment examples in the Text-Only portion of the dataset. All content involving racism or hate speech is prohibited from inclusion in our dataset.

| Knowledge Point | Source Question | English Question |
|---|---|---|
| The Mogao Caves, also known as the Thousand Buddha Grottoes, feature the "Transformation Tableau of the Medicine Buddha Sutra" in Cave 220, which depicts scenes from the High Tang period, and the "Mural of Zhang Yichao's Army in Procession" in Cave 156, which showcases scenes from the Late Tang period. | 某石窟有千佛洞之称，其在唐代有以下哪些壁画作品？
A.《药师经变画》
B.《炽盛光佛图》
C.《张议潮统军出行图》
D.《五台山图》 | Which mural paintings from the Tang Dynasty are found in a certain grotto known as the Thousand Buddha Caves?",
A. Transformation Tableau of the Medicine Buddha Sutra
B. Painting of Tejaprabhā Buddha
C. Mural of Zhang Yichao's Army in Procession
D. Mount Wutai |
| Gyeongbokgung Palace was built with Bukaksan as its main mountain, strategically placing its buildings on a spacious site with Gwanghwamun as its main gate, opening onto a wide boulevard that formed the center of Hanyang, the capital of the Joseon Dynasty. Gyeongbokgung was destroyed during the Imjin War in 1592, the 25th year of King Seonjo's reign. | 백악산을 주산으로 넓은 지형에 건물을 배치하고 정문 앞으로 넓은 육조거리가 펼쳐진 한양의 중심이 1592년 소실된 계기는 무엇인가?
A. 6.25 전쟁
B.화재 발생
C.임진왜란
D.을미사변 | What was the cause of the destruction in 1592 of the central area of Hanyang, where buildings were laid out across a wide terrain with Bugaksan as the main mountain and a broad Yukjo Street stretching out in front of the main gate?
A. Korean War
B.Fire outbreak
C. Imjin War
D. Eulmi Incident |

| Knowledge Point | Source Question | English Question |
|---|---|---|
| Char Kway Teow: This stir-fried flat noodle dish often contains prawns. It is typically served with lime. Fried Hokkien Mee: a stir-fried dish of yellow noodles and rice vermicelli cooked in a rich stock made from pork bones and prawn heads. It is almost always served with a wedge of calamansi lime on the side. | பின்வரும் பொதுவான சிங்கப்பூர் நூடுல்ஸ் உணவுகளில் எது அடிக்கடி இறால் மற்றும் எலுமிச்சை சேர்க்கப்பட்டிருக்கும்?
A. சார்க்வே டியோ
B. லெக்சா
C. வறுத்த ஹொாக்கியன் மீ
D.கிரே∴பிஷ் நூடுல்ஸ் | Which of the following common Singaporean noodle dishes often include prawns and lime?
A. Char kway teow
B. Leksa
C. Fried Hokkien Mee
D. Crayfish Noodles |
| The word "Gamcha" is Bengali/Assamese word which comes from two very simple and commonly used Bengali/Assamese words, (গা) ga which means "Body", nd (মুছা) mucha which means "wipe". It is often used as traditional full sized handkerchief. And it looks like a towel and acts as one too in case of necessity, it acts as a mask which means dust and pollution stays away from you, it also acts as sun protection so that you don't get much tanned or skin burn. | रवि दक्षिण एशिया में उष्णकटिबंधीय जलवायु के लिए अनुकूलित पारंपरिक परिधानों का अध्ययन कर रहे हैं। उन्होंने पश्चिम बंगाल में ग्रामीण पुरुषों के एक समूह को पसीना पोंछने, धूप में सिर ढकने और यहां तक कि अस्थायी मास्क के रूप में हल्के सूती कपड़े का उपयोग करते हुए देखा। वह संभवतः किस पारंपरिक वस्तु का उल्लेख कर रहे हैं?
A. कुर्ता
B. धोती
C. गमछा
D. शेरवानी | Ravi is studying traditional garments in South Asia adapted for tropical climates. He notices a group of rural men in West Bengal using a light cotton cloth for wiping sweat, covering their heads in the sun, and even as a makeshift mask. Which traditional item is he likely referring to?
A. Kurta
B. Dhoti
C. Gamcha
D. Sherwani |

Figure 9: Text-Only Question Examples

## A.4 DETAILS OF THE TTS TOOL AND PROCEDURE FOR BUILDING THE SPEECH DATA

Before generating the English audio, we standardized the input text through normalization of numbers and symbols, handling of abbreviations and special terms, and sentence segmentation.

To ensure high-quality speech synthesis, we employed CosyVoice(Du et al., 2025) for English audio generation. This tool supports voice cloning from sampled speakers, producing speech that preserves the timbre and accent of the reference voice. We collected representative recordings from native speakers across eight countries to capture diverse accents for speech synthesis. For standard English, we adopted CosyVoice's built-in default English voice (English female voice). For non-English languages, CosyVoice was also used to generate Chinese, Japanese, and Korean audio. In addition, we employed in-house high-quality TTS systems built by different speech processing teams for Vietnamese, Tamil, Mongolian, and Malay, while the Coqui-ai TTS toolkit (Coqui.ai, 2025) was used for Indonesian and Hindi.

Each generated audio sample was individually verified. When errors occurred—such as inappropriate pauses, missing segments, or mispronunciations—we first adjusted the input text and resynthesized the audio, as TTS systems are often highly sensitive to textual variations. If repeated corrections still failed, we resorted to manual re-recording. Unlike in other language tasks, our requirement here was not fluency or naturalness, but rather clear articulation of the questions and answer options.

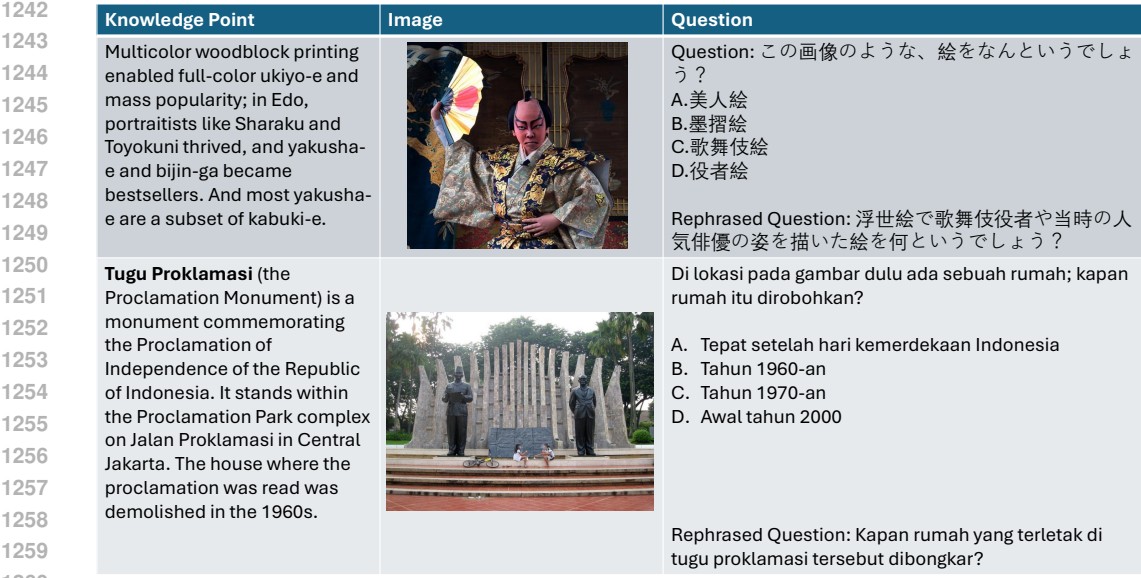

| Knowledge Point | Image | Question |
|---|---|---|
| Multicolor woodblock printing enabled full-color ukiyo-e and mass popularity; in Edo, portraitists like Sharaku and Toyokuni thrived, and yakusha-e and bijin-ga became bestsellers. And most yakusha-e are a subset of kabuki-e. | | Question: この画像のような、絵をなんというでしょう？
A.美人絵
B.墨摺絵
C.歌舞伎絵
D.役者絵

Rephrased Question: 浮世絵で歌舞伎役者や当時の人気俳優の姿を描いた絵を何というでしょう？ |
| **Tugu Proklamasi** (the Proclamation Monument) is a monument commemorating the Proclamation of Independence of the Republic of Indonesia. It stands within the Proclamation Park complex on Jalan Proklamasi in Central Jakarta. The house where the proclamation was read was demolished in the 1960s. | | Di lokasi pada gambar dulu ada sebuah rumah; kapan rumah itu dirobohkan?

A.  Tepat setelah hari kemerdekaan Indonesia
B.  Tahun 1960-an
C.  Tahun 1970-an
D.  Awal tahun 2000


Rephrased Question: Kapan rumah yang terletak di tugu proklamasi tersebut dibongkar? |

Figure 10: VQA/Rephrase VQA Question Examples

All audio files were standardized in WAV format with a 16 kHz sampling rate. And our speech generation uses two input types.

1. Only the question stem from the *Rephrase VQA (text-only)* item;

2. The entire *Rephrase VQA (text-only)* item, including the question and its answer options.

Table 8 summarizes the models used for speech generation across different languages.

Table 8: Models used for speech data generation

| Language | Model |
|---|---|
| English with accent, Korean, Japanese, Chinese | CosyVoice2-0.5B (Du et al., 2025) |
| English without accent | CosyVoice-300M (Du et al., 2024) |
| Vietnamese, Tamil, Mongolian, Malay | Internally developed TTS systems |
| Indonesian | Indonesian TTS (Kim et al., 2021) |
| Hindi | AI4Bharat Indic-TTS (Sankar et al., 2024) |

## A.5   MODEL SELECTION

**Open-source multilingual text-only LLMs.**   We evaluate Qwen3-30B-A3B-Thinking-2507 Team (2025b), Baichuan-M2-32B Team et al. (2025b), GLM-4-9B-ChatGLM et al. (2024), and Ministral-8B-Instruct Jiang et al. (2023) for their multilingual capabilities.

**Open-source multimodal LLMs.**   We evaluate nine vision–language (and omni) models on image and text: Qwen2.5-VL-32B-Instruct (Bai et al., 2025), Llama-3.2-11B-Vision-Instruct (Touvron et al., 2023), Kimi-VL-A3B-Instruct (Team et al., 2025a), DeepSeek-VL-Small (Wu et al., 2024), GLM-4.1V-9B-Thinking (Team et al., 2025c), InternVL-Chat-V1-5 (Chen et al., 2024b), and Qwen2.5-Omni-7B (Xu et al., 2025). Kimi-VL-A3B-Instruct, DeepSeek-VL-Small, and InternVL-Chat-V1-5 are evaluated in English only; Qwen2.5-VL-32B-Instruct, Llama-3.2-11B-Vision-Instruct, and GLM-4.1V-9B-Thinking are evaluated in multiple languages. Qwen2.5-Omni-7B is evaluated across image, text, and speech in a multilingual setting. We summarize the information of the LLMs evaluated in our experiments in Table 9.

## A.6   PROMPT TEMPLATES AND EXPERIMENTAL SETTINGS

**Prompt templates for evaluation tasks across modalities.**   Table 11 presents the English prompts used in our evaluations across different modalities. When the query switches to another language, the

**User Guideline for Cultural Dataset Creation**

**Target categories:**

Daily Life/Culture, Food/Cuisine, Transportation, Buildings, History, Geographical Location and Climate, Education, Fashion/Clothing, and Language/Ethnicity.

Please follow the steps below to create the cultural dataset.

**1) Cultural Prompt Creation**

Hold a group discussion and brainstorm prompts. For **both tracks—Text-Only** and **Multi-Modal**—propose **at least 56** cultural keywords or short phrases **for each category** above. Ensure diversity and broad regional coverage across the country. Prefer culturally distinctive, locally specific items, not only well-known ones.

**2) Data Collection**

- Using the prompts, search the web for relevant **texts and images**, and selectively excerpt passages to serve as the **content base** for question construction.
- Guidelines:
- **Verify uncertain content** via multiple sources; do not use unverifiable information.
- **Images must be under a CC license** and permitted for research use. If no usable image can be found for a prompt, assign it to the **Text-Only** track.
- For **Language/Ethnicity** in the Text-Only track, you may include **code-switching** scenarios or **dialects** where locally relevant.

**3) Question Construction**

- Build questions from the collected cultural materials. Ensure **over 60%** of your questions require **multi-step cultural reasoning**. Start drafting in the **local language**. Keep in mind:
- Wording must be **clear and grammatical**, with correct spelling.
- If a referent is used, its **coreference must be unambiguous**.
- For **image-based** questions, the answer **must depend on the image**.
- Ensure that image-based questions are **answerable from visual evidence** in the image.
- Provide **four options** per question. Include **at least one correct answer**; distractors must be in the **same category** and **reasonably similar** to prevent trivial elimination. **No blank options**.
- Record the **knowledge points** used to create each question; these will be used to assess answer faithfulness. Provide a **2–4 sentence** summary of the **minimal knowledge** required to answer.
- Avoid repeating the **same query style** (e.g., repeatedly asking about "taste"). Vary both **knowledge points** and **question angles**.
- The **question stem** must not exceed **300 words**. Each **option** should be ≤ **50 words** and as concise as possible.
- **Randomize** the position of the correct option; do not always use the same letter.

**4) Translation**

Translate all questions into **English** using a **closed-source LLM**, then **manually check** each translation for accuracy and correct any errors. For terms without official English renderings, use **transliteration** or a **widely accepted** translation.

Please fill in the questions and options according to the **template provided in the email attachment**.

Figure 11: Annotator Guideline

corresponding translated version of the prompt will be used to ensure input-language consistency. Table 12 presents the prompts used to invoke closed-source model APIs for translation, answer-consistency evaluation, and answer extraction.

**Experiments setting.** All evaluations in this work are conducted in the zero-shot setting, using single-turn inference for each model on an NVIDIA H100 80G. For image inputs that exceed a model's maximum allowable resolution, we proportionally downscale the image until it is under 1 megapixel before testing. We decode with greedy search (no sampling; `do_sample=false`, `num_beams=1`), so temperature/top-$p$/top-$k$ are not used; the maximum output length is set to 2048 tokens to ensure reproducibility. GPT-4o and Gemini 2.5 Flash are accessed via Open-

Table 9: Overview of selected LLMs

| Model | Type | Language Coverage | Modalities |
|---|---|---|---|
| Qwen2.5-VL-32B-Instruct | Vision–Language (VL) | Multilingual | Image + Text |
| Llama-3.2-11B-Vision-Instruct | Vision–Language (VL) | Multilingual | Image + Text |
| Kimi-VL-A3B-Instruct | Vision–Language (VL) | English only | Image + Text |
| DeepSeek-VL2-Small | Vision–Language (VL) | English only | Image + Text |
| GLM-4.1V-9B-Thinking | Vision–Language (VL) | Multilingual | Image + Text |
| InternVL-Chat-V1-5 | Vision–Language (VL) | English only | Image + Text |
| Qwen2.5-Omni-7B | Omni-modal | Multilingual | Image + Text + Speech |
| Qwen3-30B-A3B-Thinking-2507 | Text-only LLM | Multilingual | Text |
| Baichuan-M2-32B | Text-only LLM | Multilingual | Text |
| GLM-4-9B-Chat | Text-only LLM | Multilingual | Text |
| Ministral 8B-Instruct | Text-only LLM | Multilingual | Text |
| GPT-4o | Vision–Language (VL) | Multilingual | Image + Text |
| Claude-Sonnet-4 | Vision–Language (VL) | Multilingual | Image + Text |
| Gemini 2.5 Flash | Omni-modal | Multilingual | Image + Text + Speech |

Table 10: Number of Incorrect Cases Across Different Models and Languages.

| Model | Number of Incorrect Cases | | |
|---|---|---|---|
| | English | Chinese | Indonesian |
| GLM-4-9B-chat | 337 | 320 | 312 |
| Mistral-8B-Instruct | 370 | 384 | 341 |
| Claude Sonnet 4 | 232 | 231 | 222 |
| Qwen3-30B-A3B | 271 | 265 | 276 |

Router API platform, and Claude via the Anthropic API. For all closed-source models, we set `temperature=0` to minimize randomness and improve reproducibility. Our speech inputs are no longer than 30 seconds and sampled at 16 kHz—well below Gemini 2.5 Flash's maximum speech-input duration and Qwen2.5-Omni-7B's maximum input token limit. So we do not perform any input-length processing.

A.7 PERFORMANCE VARIATION ACROSS DATA CATEGORIES

Table 11: English prompts for different modals.

| Modal | Prompt Template |
|---|---|
| Text-Only/Rephrase | Please answer the following culture-related question.\n{question}\n{options}\nThis is a multiple-choice question. Please first return all possible option letters, then explain your choice in English. |
| VQA | Based on the image, please answer the following culture-related question.\n{question}\n{options}\nThis is a multiple-choice question. Please first return all possible option letters, then explain your choice in English. |
| Speech question & text options | This is a culture-related question.\n Based on the question mentioned in this audio, please choose the correct answers from the following provided options. {options}\nThis is a multiple-choice question. Please first return all possible option letters, then explain your choice in English. |
| Speech question & options | This is a culture-related question. Based on the question and options mentioned in this audio, please choose the correct options. This is a multiple-choice question. Please first return all possible option letters, then explain your choice in English. |

Table 12: Prompts Used with Closed-Source APIs for Translation, Answer Extraction, and Answer–Reference Consistency Checking

| Task | Prompt |
|------|--------|
| Translation | Translate the following sentence into English. \n{Input sentence}\nThis is a multiple-choice question in the cultural domain of {Country}. Pay particular attention to the terms in the input and use their official translations; if no official translation exists, you may use transliteration. Ensure accuracy, faithfulness, and fluency. Return only the translation; do not include any additional hints or analysis. |
| LLM-as-Judge | You are an expert evaluator. Your task is to determine if the 'Model Answer' correctly and completely incorporates the information from the 'Knowledge Point'. Knowledge Point:\n{knowledge_point}\nModel Answer\n{model_answer}\n\n. If the Model Answer correctly and completely includes the information from the Knowledge Point, or if the model's response aligns with the Knowledge Point's content and viewpoint. And the Model Answer doesn't contain any factual error, Answer 'yes'. Otherwise Answer 'no'. Please only return 'yes' or 'no'. |
| Answer Extraction | This is a model's response to a multiple-choice question. First, understand the text, then extract the model's chosen options, returning only the option letters (e.g., A, B, C). Do not include the option content. Output the result in a format like [A, B]. If the response does not contain any final choice, return NA. |

To investigate how different cultural categories affect model performance, we evaluate models by category under each modality, taking the average accuracy across all languages within a modality as the category's accuracy. As shown in Figures 12, 13, and 14, under the Text-Only and VQA modalities, models consistently struggle more with the "Fashion/Clothing" and "Transportation" categories compared to others, while performing strongly on "Daily life habits/Culture" and "Education." In the speech modality, models show slightly weaker performance on "Food/Cuisine" and "History," but still maintain strong results on "Daily life habits/Culture" and "Education".

These patterns likely reflect both data and task asymmetries across categories. Daily life habits/culture and education rely more on high-frequency, broadly documented facts and relatively coarse-grained reasoning, making them easier for models trained on abundant, well-aligned multilingual data. By contrast, fashion/clothing and transportation encode highly time-sensitive and region-specific concepts (e.g., changing trends, local garment names, route and line names), while food/cuisine and history also depend on rare, culturally bound proper nouns; in the speech modality, models must directly map variable acoustic realizations of these low-frequency terms to the correct concepts, further depressing performance in these categories.

We further explore the relationship between cultural types and languages based on Figure 16. When Mongolian is employed as the input language, the model exhibits markedly superior performance in the education category relative to other categories. Conversely, when Hindi, Vietnamese, and Tamil are utilized as input languages, the model demonstrates suboptimal performance in the Fashion/Clothing, Geographical location and climate, and education categories, respectively.

We posit that the underlying cause of this phenomenon may be attributed to the fact that for low-resource languages such as Mongolian, casual text from the internet (e.g., social media, forums) is relatively scarce. High-quality corpora for such languages predominantly originate from government documents, digitized textbooks, academic papers, or Wikipedia. This compels the model to "absorb" a substantial volume of formal, educational texts during the training phase. Consequently, the model exhibits a form of "overfitting" with exceptionally high performance when processing the education category. In contrast, Hindi internet data may be dominated by news, politics, religion, or literature. Contemporary fashion and clothing-related content on the Indian internet likely exists primarily in English (or Hinglish, a Hindi-English code-mixed variety). If the model is trained exclusively on pure Hindi, it consequently lacks the vocabulary and contextual framework necessary to describe "fashion." Similarly, if Vietnamese training corpora lack specific technical descriptions of geography and climate, the model will exhibit domain-specific knowledge gaps. Tamil, being widely used

across diverse regions including Sri Lanka, India, and Singapore, encounters substantial educational disparities across these countries, and such regional conflicts may impede the model's ability to adequately fit educational domain knowledge.

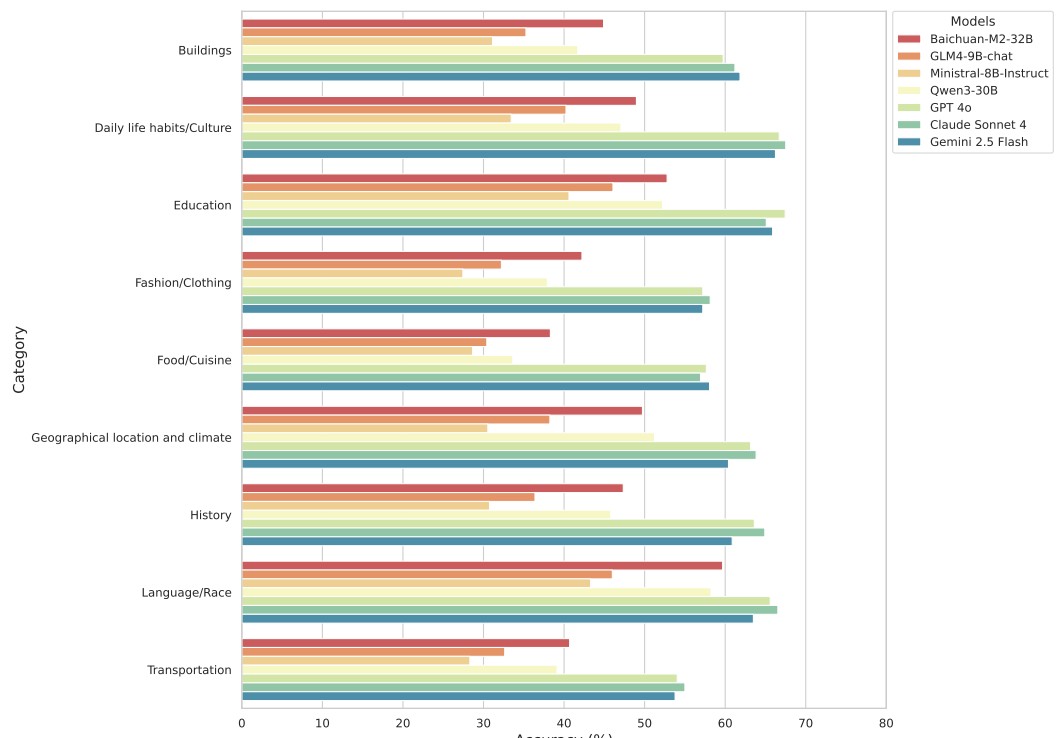

Figure 12: Performance of different models on different categories in the text-only dataset.

### A.8 PERFORMANCE OF LLMS ON MMA-ASIA ACROSS MODALITIES

The exact data corresponding to the bar chart in Section 4.2, Figure 2 are presented in Tables 13, 14, 15, 16.

Table 13: Text-only modality performance with exact numbers, measured by Accuracy (%): the number of items where the model's choice exactly matches the correct option, divided by 500. "-" means "not support". The better-performing result among different languages within the same country is **bolded**.

| Model | CN-zh | CN-en | ID-id | ID-en | IN-hi | IN-en | JP-ja | JP-en | KR-ko |
|---|---|---|---|---|---|---|---|---|---|
| Gemini 2.5 Flash | **53.6** | 45.4 | **56.2** | 53.2 | 73.0 | **73.6** | 47.4 | **48.2** | **72.2** |
| Claude Sonnet 4 | **53.8** | 53.6 | **55.6** | 54.8 | **74.4** | 74.2 | 48.8 | **50.6** | **71.4** |
| GPT-4o | 36.0 | **50.2** | 53.6 | **55.8** | 73.2 | **77.2** | 49.8 | **53.2** | **71.4** |
| GLM-4-9B-chat | **36.0** | 32.6 | 37.6 | **49.4** | 45.6 | **57.0** | 30.2 | **33.6** | **45.4** |
| Mistral-8B-Instruct | 23.2 | **26.0** | 31.8 | **35.6** | 42.2 | **49.0** | 23.6 | **30.2** | **44.8** |
| Baichuan-M2-32B | **51.2** | 49.8 | 50.6 | **53.2** | 54.6 | **69.0** | 39.6 | **40.8** | **56.4** |
| Qwen3-30B-A3B | **47.0** | 45.8 | 44.8 | **46.2** | 58.6 | **62.2** | **38.4** | 38.0 | 43.4 |
| Model | KR-en | MN-mn | MN-en | SG-zh | SG-en | SG-ms | SG-ta | VN-vi | VN-en |
| Gemini 2.5 Flash | 65.2 | 58.2 | **70.0** | 34.0 | **43.4** | 39.0 | 31.4 | **73.6** | 73.0 |
| Claude Sonnet 4 | 66.0 | 59.4 | **71.0** | 29.8 | **41.8** | 37.8 | 31.6 | **74.4** | 72.2 |
| GPT-4o | 68.8 | 55.4 | **69.4** | 22.4 | **47.6** | 39.4 | 33.2 | **74.8** | 71.8 |
| GLM-4-9B-chat | 42.8 | 23.0 | **52.6** | 20.2 | **24.8** | 18.4 | 19.4 | 47.6 | **51.0** |
| Mistral-8B-Instruct | 40.8 | 13.0 | **50.8** | 17.2 | **25.2** | 22.4 | 15.4 | 41.8 | **48.6** |
| Baichuan-M2-32B | 51.6 | 16.8 | **61.6** | 26.6 | **33.6** | 28.8 | 19.4 | **67.6** | 62.8 |
| Qwen3-30B-A3B | **49.8** | 44.6 | **65.0** | 26.4 | **29.2** | 23.6 | 22.8 | 57.8 | **65.8** |

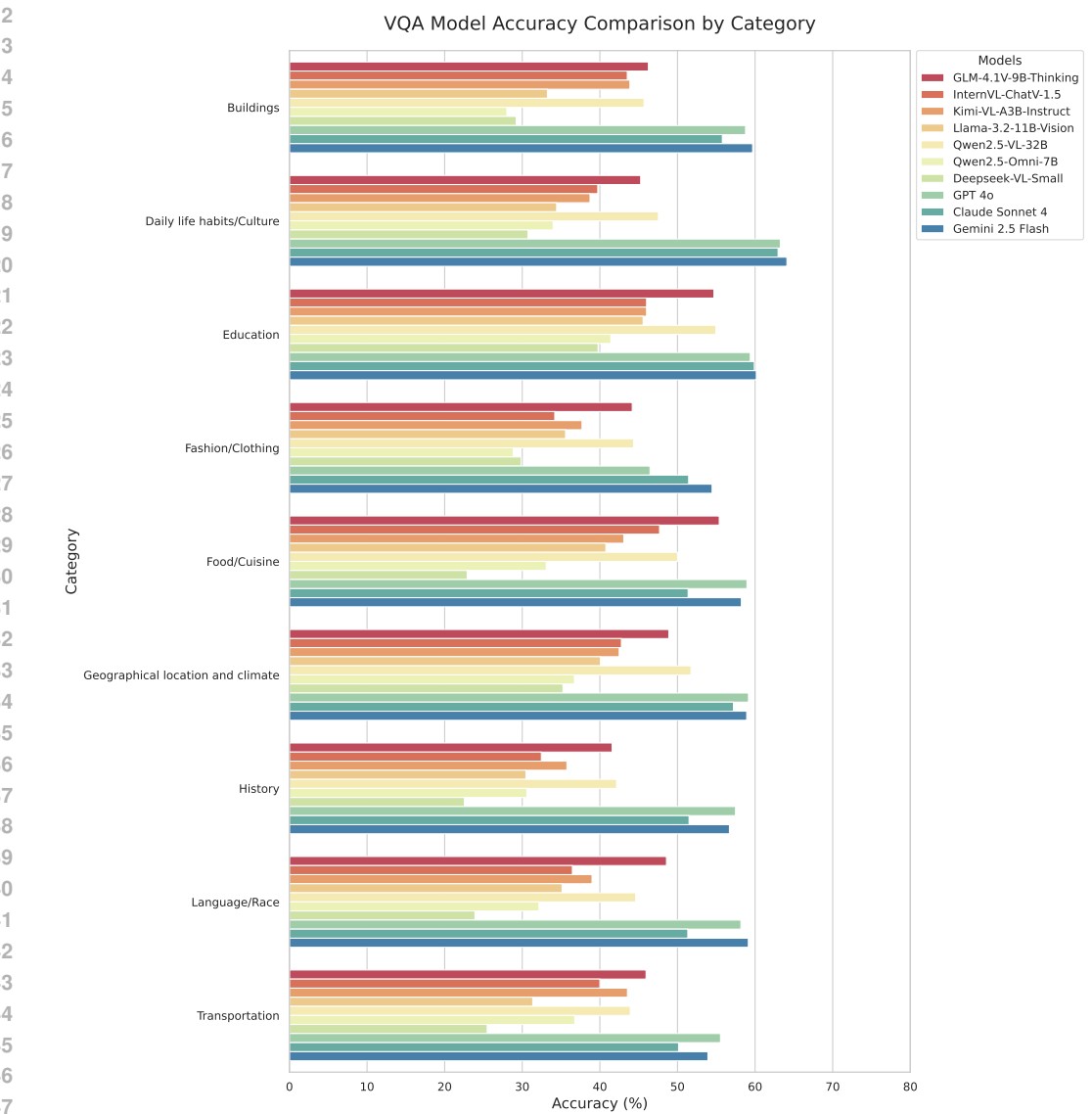

Figure 13: Performance of different models on different categories in the VQA dataset.

### A.9 Joint MCQ Answering and Explanation Performance of LLMs on MMA-ASIA Across Modalities

The model performance based on both the MCQ choices and the model-generated explanations are provided in Tables 2, 17, 18, 19.

### A.10 Results for fully spoken question and answering

We considered two configurations when constructing the TTS-Spoken QA dataset: (i) converting only the question stem to speech while keeping the answer options as text, and (ii) converting both the stem and the options to speech. To preserve comparability with VQA under controlled variables and to minimize ambiguity introduced by fully spoken options, our main experiments adopt the "spoken stem + textual options" setting across five evaluation dimensions. Results for the fully spoken setting (spoken stem and spoken options) on the test set are provided in Figure 17 and Table 20 for reference. We find that converting both the question and options to speech leads to a significant performance drop compared with the "spoken question + textual options" configuration, indicating that spoken options introduce greater uncertainty than the spoken question itself. This warrants further investigation in future work.

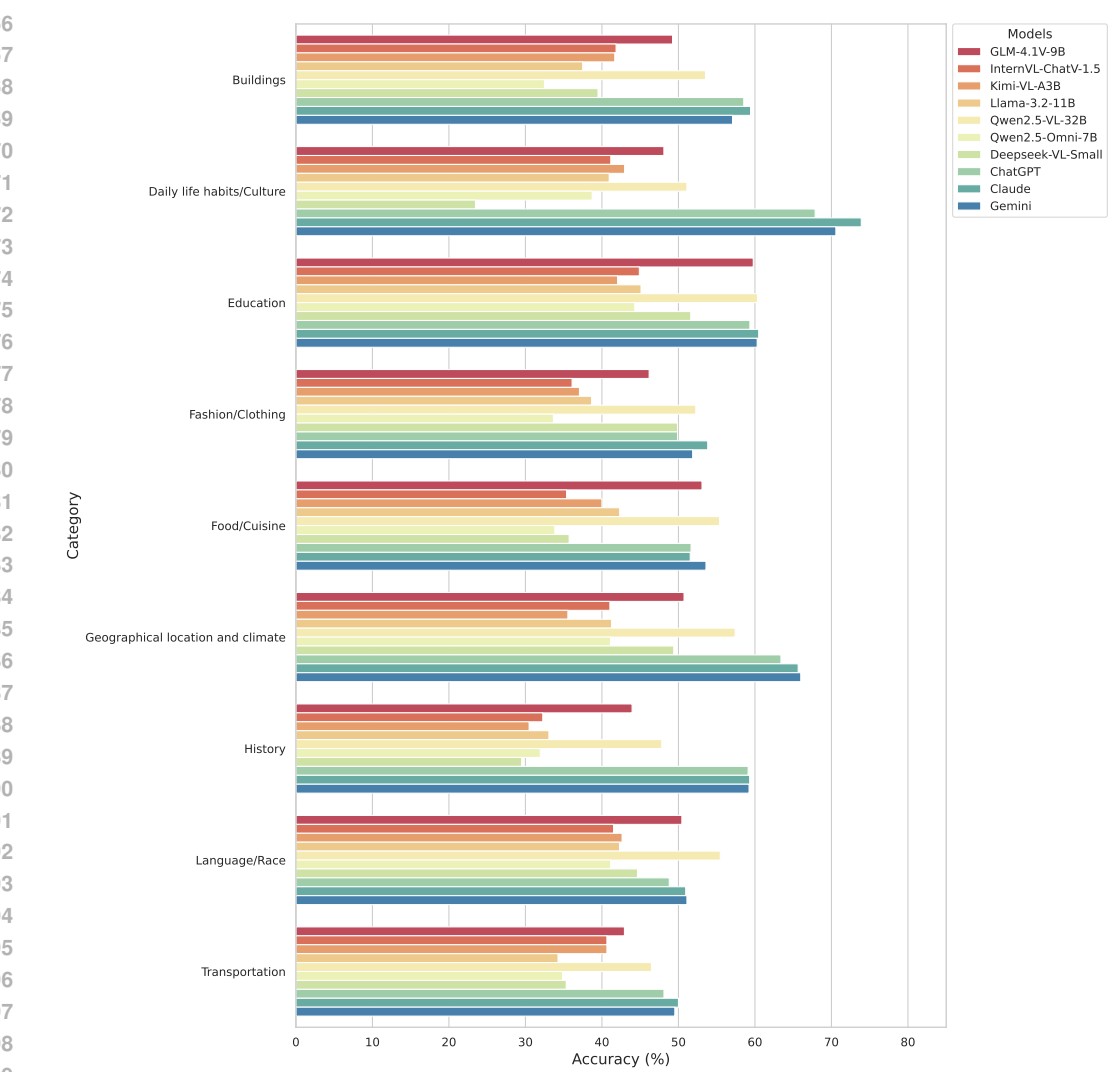

Figure 14: Performance of different models on different categories in the rephrased VQA dataset.

## A.11 VISION-ABLATED PREFIX REPLAY

We found that visual content increases reasoning hallucinations in models compared to text-only QA, despite accurate image recognition capabilities. To validate our hypothesis, we propose a "Vision-ablated Prefix Replay" (VPR) method. This method enables a model with parameters $\theta$ to first describe the image contents based on image $\mathbf{x}^{\text{img}}$ and text prompt $\mathbf{x}^{\text{text}}$. After this initial description, we structurally ablate the visual condition and fix the prefix $\hat{\mathbf{S}}_{1:n}$ for subsequent reasoning generation. This evaluates the marginal contribution of visual conditions to reasoning. Specifically, assuming the model completes image description within the first $n$ tokens, we remove visual conditions starting from the $(n+1)$-th token and use only the text prompt and generated tokens as prefix. The joint probability distribution of the subsequent sequence $\mathbf{S}_{>n} = (s_{n+1}, \ldots, s_T)$ can be expressed as:

$$p_\theta\big(\mathbf{S}_{>n} \mid \mathbf{x}_{\text{text}},\ \varnothing,\ \hat{\mathbf{S}}_{1:n}\big) = \prod_{t=n+1}^{T} p_\theta\big(s_t \mid \mathbf{x}_{\text{text}},\ \varnothing,\ \hat{\mathbf{S}}_{1:n},\ \mathbf{s}_{n+1:t-1}\big). \tag{6}$$

Previous work has explored related ideas. For example, Zhang et al. (2025) attempt to suppress visual leakage by blocking the attention paths to image-token positions during decoding. However, this cannot fully eliminate the influence of visual content: in a causal language model, the information of earlier image tokens is encoded into subsequent question tokens, so residual visual

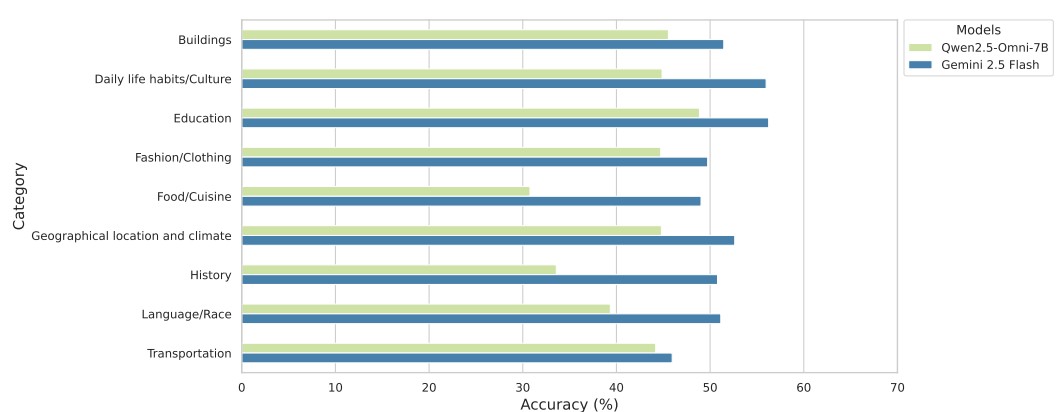

Figure 15: Performance of different models on different categories in the Speech dataset.

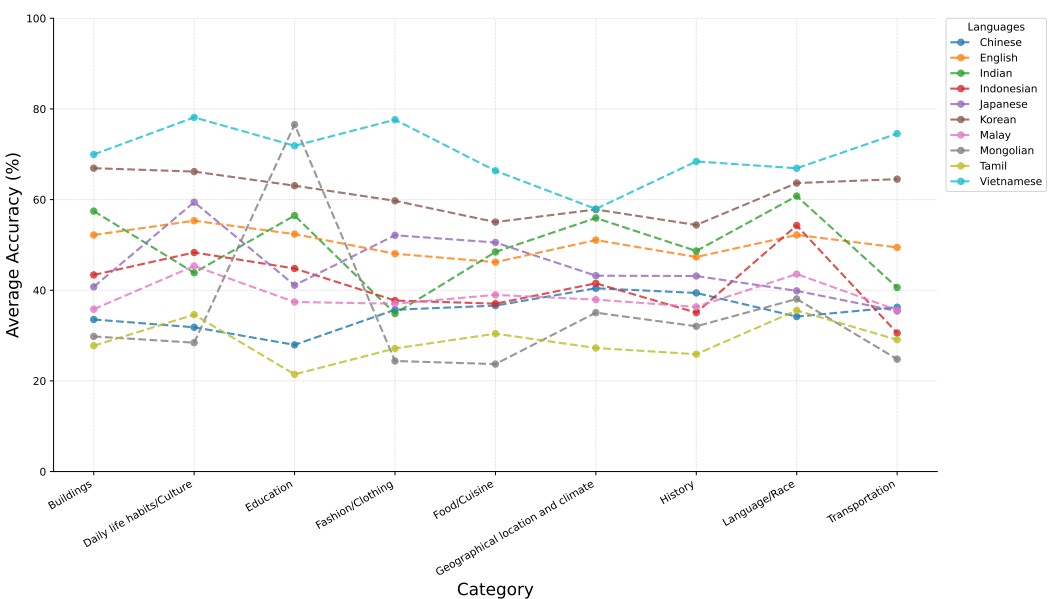

Figure 16: Average performance of the models on different question categories under different language settings.

information remains even when attention to image tokens is blocked. In contrast, our method first elicits a textual description of the image, then removes the visual input and recomputes the representations of the prefix tokens, thereby purging visual information and enabling a more precise assessment of the image content's contribution to the model's reasoning process.

For the question in Figure 6, the completed responses under different generation modes are provided in Table 21.

## A.12 ANALYSIS OF SPEECH AS A CULTURAL PRIOR

Compared to images and text, speech input introduces greater uncertainty through environmental noise, homophony, and accents—with accents closely tied to cultural context. Our research reveals that accents function beyond mere noise. Testing synthetic speech in standard English versus multiple national accents, we found Qwen and Gemini outperformed their standard English baselines in 6 and 5 country-specific cultural settings (Figure 2), respectively. Notably, Qwen achieved 2.8% and 3.6% accuracy gains for Indonesian and Japanese accents (Table 16). We attribute this to systematic co-occurrence of accented English with country-specific entities and contexts in training corpora, enabling accents to serve as cultural and lexical priors during inference. Our findings demonstrate

Table 14: VQA modality performance with exact numbers, measured by Accuracy (%): the number of items where the model's choice exactly matches the correct option, divided by 500. "-" means "not support". The better-performing result among different languages within the same country is bolded.

| Model | CN-zh | CN-en | ID-id | ID-en | IN-hi | IN-en | JP-ja | JP-en | KR-ko |
|---|---|---|---|---|---|---|---|---|---|
| Gemini 2.5 Flash | **45.6** | 38.8 | 48.6 | **49.8** | **65.4** | 42.4 | 54.2 | **58.0** | 72.0 |
| Claude Sonnet 4 | **47.6** | 33.6 | **49.8** | 46.4 | **54.0** | 39.0 | 47.4 | **53.6** | 71.8 |
| GPT-4o | 32.8 | **38.4** | 53.6 | 50.2 | **65.8** | 45.6 | 59.4 | **60.0** | 69.2 |
| Qwen2.5-VL-32B | **40.2** | 30.8 | 37.6 | **37.8** | 40.2 | **45.0** | **47.6** | 46.8 | 63.2 |
| Llama-3.2-11B-Vision | 21.6 | 15.2 | 26.2 | **31.2** | 35.4 | **40.2** | 27.2 | **35.2** | 30.0 |
| Kimi-VL-A3B-instruct | - | **21.2** | - | **25.4** | - | **34.4** | - | **37.2** | - |
| Qwen2.5-Omni-7B | **30.6** | 23.6 | 21.8 | **32.4** | 19.4 | **33.6** | 25.6 | **39.2** | 18.6 |
| Deepseek-VL-Small | - | **12.2** | - | **25.4** | - | **25.0** | - | **26.0** | - |
| GLM-4.1V-9B-Thinking | **46.4** | 26.2 | 28.6 | **34.8** | **50.8** | 43.4 | 44.0 | **44.8** | 61.0 |
| InternVL-Chat-V1-5 | - | **15.4** | - | **23.0** | - | **39.2** | - | **40.6** | - |

| Model | KR-en | MN-mn | MN-en | SG-zh | SG-en | SG-ms | SG-ta | VN-vi | VN-en |
|---|---|---|---|---|---|---|---|---|---|
| Gemini 2.5 Flash | **72.8** | 41.2 | **49.2** | 53.0 | **62.2** | 59.8 | 60.8 | **76.6** | 65.4 |
| Claude Sonnet 4 | **74.4** | 35.0 | **49.0** | 34.6 | **57.2** | 55.2 | 49.8 | **73.2** | 67.4 |
| GPT-4o | 65.8 | 41.4 | **53.0** | 30.2 | **70.6** | 68.4 | 62.4 | **75.8** | 63.8 |
| Qwen2.5-VL-32B | **67.0** | 20.6 | **45.2** | 40.8 | **52.8** | 45.6 | 22.6 | **65.0** | 64.6 |
| Llama-3.2-11B-Vision | **47.8** | 5.8 | **21.6** | 32.2 | **39.4** | 35.0 | 15.0 | 48.6 | **49.6** |
| Kimi-VL-A3B-instruct | **56.0** | - | - | 38.6 | **43.6** | - | - | - | **50.0** |
| Qwen2.5-Omni-7B | **49.2** | 3.2 | **36.6** | 30.0 | **43.4** | 38.6 | 14.2 | **59.8** | 58.2 |
| Deepseek-VL-Small | **30.0** | - | - | 16.4 | **27.2** | - | - | - | **41.6** |
| GLM-4.1V-9B-Thinking | **65.4** | 28.0 | **39.4** | 44.8 | **52.8** | 41.8 | 34.8 | 59.4 | **59.8** |
| InternVL-Chat-V1-5 | **56.6** | - | - | 28.0 | **44.8** | - | - | - | **54.0** |

Table 15: Rephrase VQA (Text-Only) modality performance with exact numbers, measured by Accuracy (%): the number of items where the model's choice exactly matches the correct option, divided by 500. "-" means "not support". The better-performing result among different languages within the same country is **bolded**.

| Model | CN-zh | CN-en | ID-id | ID-en | IN-hi | IN-en | JP-ja | JP-en | KR-ko |
|---|---|---|---|---|---|---|---|---|---|
| Gemini 2.5 Flash | **45.2** | 37.2 | **52.0** | 48.8 | **67.0** | 53.8 | 62.4 | **65.6** | **76.4** |
| Claude Sonnet 4 | **52.8** | 48.0 | **53.8** | 52.6 | **68.6** | 51.8 | 67.8 | **68.2** | **79.6** |
| GPT-4o | 41.0 | **46.2** | 57.8 | 55.2 | **69.0** | 59.6 | **67.0** | 66.4 | 73.2 |
| Qwen2.5-VL-32B | **45.0** | 34.4 | **45.0** | 44.4 | **51.1** | 47.0 | 56.0 | **58.4** | 66.8 |
| Llama-3.2-11B-Vision | 18.4 | 15.6 | **35.2** | 30.4 | **39.8** | 36.6 | 39.2 | **45.6** | 43.8 |
| Kimi-VL-A3B-Instruct | - | **21.6** | - | **24.6** | - | **32.2** | - | **43.8** | - |
| Qwen2.5-Omni-7B | 31.4 | 22.8 | 23.4 | **35.8** | 28.4 | **42.8** | 35.0 | **51.4** | 29.6 |
| Deepseek-VL-Small | - | **15.6** | - | **30.6** | - | **29.2** | - | **40.4** | - |
| GLM-4.1V-9B-Thinking | **46.4** | 30.6 | **37.6** | 33.6 | **51.4** | 33.8 | 49.0 | **53.0** | 66.6 |
| InternVL-Chat-V1.5 | - | **16.8** | - | **27.0** | - | **37.4** | - | **40.9** | - |

| Model | KR-en | MN-mn | MN-en | SG-zh | SG-en | SG-ms | SG-ta | VN-vi | VN-en |
|---|---|---|---|---|---|---|---|---|---|
| Gemini 2.5 Flash | 76.4 | 47.8 | **49.2** | 54.6 | **63.6** | 59.6 | 52.4 | **76.0** | 71.6 |
| Claude Sonnet 4 | 77.0 | 38.8 | **56.0** | 46.0 | **65.2** | 61.0 | 51.8 | **76.0** | 73.6 |
| GPT-4o | **75.2** | 43.2 | **59.8** | 30.2 | **68.2** | 66.4 | 53.6 | **79.4** | 72.8 |
| Qwen2.5-VL-32B | **72.0** | 19.0 | **56.2** | 36.4 | **57.6** | 50.8 | 25.4 | 69.2 | **70.0** |
| Llama-3.2-11B-Vision | **54.6** | 10.2 | **29.0** | 25.4 | **45.2** | 40.2 | 20.0 | 39.4 | **51.2** |
| Kimi-VL-A3B-Instruct | **53.8** | - | - | 40.4 | 36.6 | - | - | - | **40.0** |
| Qwen2.5-Omni-7B | **60.6** | 2.6 | **43.4** | 21.4 | **43.6** | 29.4 | 9.8 | 58.4 | **60.0** |
| Deepseek-VL-Small | **58.8** | - | - | 25.2 | **39.2** | - | - | - | **58.4** |
| GLM-4.1V-9B-Thinking | **67.0** | 26.6 | **43.8** | 42.0 | **51.0** | 40.4 | 34.8 | **66.2** | 62.8 |
| InternVL-Chat-V1.5 | **48.6** | - | - | 36.0 | **41.6** | - | - | - | **46.0** |

that accents can function as valuable cultural cues rather than simply noise sources for model exploitation.

Table 16: Speech modality (speech question & text options) performance with exact numbers, measured by Accuracy (%): the number of items where the model's choice exactly matches the correct option, divided by 500. "-" means "not support". The better-performing result among different languages within the same country is **bolded**.

| Model | CN-zh | CN-en | CN-en | ID-id | ID-en | ID-en | IN-hi | IN-en | IN-en |
|---|---|---|---|---|---|---|---|---|---|
| | - | Acc | NoAcc | - | Acc | NoAcc | - | Acc | NoAcc |
| Qwen2.5-Omni-7B | **26.2** | 21.2 | 21.0 | 29.4 | 29.4 | **32.0** | 33.2 | **33.2** | 30.4 |
| Gemini 2.5 Flash | **40.6** | 37.6 | 36.0 | 36.6 | 40.2 | **40.4** | **53.4** | 25.4 | 24.2 |

| Model | JP-ja | JP-en | JP-en | KR-ko | KR-en | KR-en | MN-mn | MN-en | MN-en |
|---|---|---|---|---|---|---|---|---|---|
| | - | Acc | NoAcc | - | Acc | NoAcc | - | Acc | NoAcc |
| Qwen2.5-Omni-7B | **45.4** | 41.8 | - | **53.6** | 51.4 | - | 43.8 | **47.8** | - |
| Gemini 2.5 Flash | 52.4 | **60.2** | 57.0 | **73.6** | 70.2 | 72.0 | 32.0 | 48.0 | **50.4** |

| Model | SG-zh | SG-en | SG-en | SG-ms | SG-ta | VN-vi | VN-en | VN-en |
|---|---|---|---|---|---|---|---|---|
| | - | Acc | NoAcc | - | - | - | Acc | NoAcc |
| Qwen2.5-Omni-7B | **39.4** | 39.4 | 37.0 | - | - | - | **54.6** | 52.0 |
| Gemini 2.5 Flash | 47.0 | **55.2** | 54.4 | 41.4 | 39.6 | 60.6 | **68.6** | 67.8 |

Table 17: Evaluated under the MCQ+Explanation metric, VQA modality performance with exact numbers, measured by Accuracy (%): the number of items where the model's choice exactly matches the correct option, divided by 500.

| Model | CN-en | ID-en | IN-en | JP-en | KR-en | MN-en | SG-en | VN-en |
|---|---|---|---|---|---|---|---|---|
| Gemini 2.5 Flash | 29.8 | 35.9 | 30.5 | 43.7 | 66.4 | 43.9 | 47.1 | 53.8 |
| Claude Sonnet 4 | 23.8 | 28.2 | 25.9 | 38.4 | 67.2 | 40.5 | 41.4 | 55.9 |
| GPT-4o | 29.9 | 31.9 | 27.4 | 40.0 | 60.1 | 37.2 | 53.2 | 55.8 |
| Qwen2.5-VL-32B | 25.0 | 22.6 | 26.1 | 34.2 | 62.4 | 41.0 | 36.8 | 51.6 |
| Llama-3.2-11B-Vision | 7.0 | 12.3 | 14.8 | 13.8 | 40.0 | 15.2 | 19.4 | 28.8 |
| Kimi-VL-A3B-instruct | 11.8 | 9.9 | 13.1 | 16.6 | 49.9 | 29.2 | 21.7 | 30.6 |
| Qwen2.5-Omni-7B | 15.6 | 21.5 | 14.7 | 23.5 | 43.9 | 31.6 | 29.5 | 44.3 |
| GLM-4.1V-9B-Thinking | 19.3 | 19.1 | 22.6 | 28.6 | 57.6 | 33.2 | 36.8 | 46.7 |
| InternVL-Chat-V1-5 | 6.3 | 7.6 | 13.0 | 16.1 | 47.6 | 20.6 | 22.6 | 30.5 |

Table 18: Evaluated under the MCQ+Explanation metric, Rephrase VQA (Text-Only) modality performance with exact numbers, measured by Accuracy (%): the number of items where the model's choice exactly matches the correct option, divided by 500.

| Model | CN-en | ID-en | IN-en | JP-en | KR-en | MN-en | SG-en | VN-en |
|---|---|---|---|---|---|---|---|---|
| Gemini 2.5 Flash | 30.2 | 27.1 | 27.9 | 58.3 | 75.1 | 45.0 | 53.7 | 62.1 |
| Claude Sonnet 4 | 39.5 | 37.2 | 43.3 | 59.7 | 75.1 | 51.7 | 55.0 | 63.0 |
| GPT-4o | 39.8 | 35.7 | 48.4 | 61.1 | 68.8 | 53.5 | 62.6 | 66.4 |
| Qwen2.5-VL-32B | 28.0 | 27.1 | 30.0 | 47.5 | 70.1 | 51.1 | 44.3 | 57.8 |
| Llama-3.2-11B-Vision | 7.6 | 13.8 | 17.6 | 24.5 | 52.6 | 21.6 | 27.1 | 35.5 |
| Kimi-VL-A3B-Instruct | 12.5 | 12.3 | 15.2 | 26.5 | 51.7 | 32.2 | 23.5 | 28.5 |
| Qwen2.5-Omni-7B | 18.0 | 25.6 | 25.5 | 39.4 | 59.5 | 39.6 | 35.3 | 49.9 |
| GLM-4.1V-9B-Thinking | 22.9 | 19.8 | 20.8 | 42.6 | 64.6 | 36.3 | 39.3 | 51.8 |
| InternVL-Chat-V1.5 | 9.3 | 11.0 | 16.9 | 21.2 | 45.4 | 28.0 | 25.1 | 31.3 |

Table 19: Evaluated under the MCQ+Explanation metric, Speech modality (speech question & text options) performance with exact numbers, measured by Accuracy (%): the number of items where the model's choice exactly matches the correct option, divided by 500.

| Model | CN-en | ID-en | IN-en | JP-en | KR-en | MN-en | SG-en | VN-en |
|---|---|---|---|---|---|---|---|---|
| Qwen2.5-Omni-7B | 21.0 | 32.0 | 30.4 | 41.8 | 51.4 | 47.8 | 37.0 | 52.0 |
| Gemini 2.5 Flash | 36.0 | 40.4 | 24.2 | 57.0 | 72.0 | 50.4 | 54.4 | 67.8 |

## A.13 CONSISTENCY ANALYSIS AND HYPERPARAMETERS SETTING FOR LLM-AS-JUDGE

Before adopting the LLM-as-judge paradigm, we conducted a small-scale study to assess human–model agreement and inter-model agreement. In addition, we further investigated the accuracy of using the LLM-as-Judge method in English versus non-English languages.

Table 20: Speech modality (speech question & options) performance with exact numbers, measured by Accuracy (%): the number of items where the model's choice exactly matches the correct option, divided by 500. "-" means "not support". The better-performing result among different languages within the same country is **bolded**.

| Model | CN-zh | CN-en | CN-en | ID-id | ID-en | ID-en | IN-hi | IN-en | IN-en |
|---|---|---|---|---|---|---|---|---|---|
| | - | Acc | NoAcc | - | Acc | NoAcc | - | Acc | NoAcc |
| Qwen2.5-Omni-7B | **28.4** | 14.2 | 15.2 | - | 23.8 | **27.0** | - | **22.2** | 20.8 |
| Gemini 2.5 Flash | **27.4** | 13.8 | 15.2 | 25.0 | **31.0** | 27.0 | **34.6** | 34.2 | 29.4 |

| Model | JP-ja | JP-en | JP-en | KR-ko | KR-en | KR-en | MN-mn | MN-en | MN-en |
|---|---|---|---|---|---|---|---|---|---|
| | - | Acc | NoAcc | - | Acc | NoAcc | - | Acc | NoAcc |
| Qwen2.5-Omni-7B | - | **35.2** | 32.8 | - | 34.0 | **35.0** | - | 23.6 | **38.4** |
| Gemini 2.5 Flash | 35.0 | **42.8** | 36.2 | 38.6 | 20.8 | **49.4** | 10.6 | 18.2 | **28.6** |

| Model | SG-zh | SG-en | SG-en | SG-ms | SG-ta | VN-vi | VN-en | VN-en |
|---|---|---|---|---|---|---|---|---|
| | - | Acc | NoAcc | - | - | - | Acc | NoAcc |
| Qwen2.5-Omni-7B | **27.0** | 26.0 | 23.6 | - | - | - | **39.2** | 37.0 |
| Gemini 2.5 Flash | 28.8 | **42.2** | 37.6 | 16.0 | 15.8 | **52.0** | 46.2 | 45.8 |

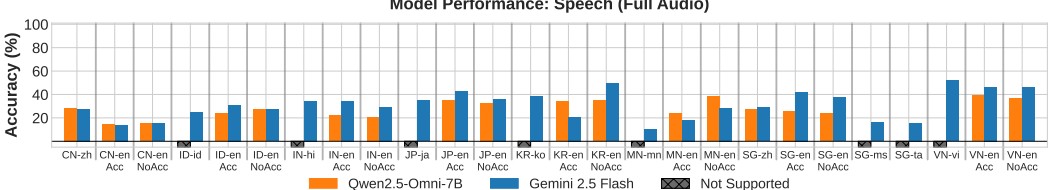

Figure 17: Speech modality (speech question & options) performance with exact numbers, measured by Accuracy (%): the number of items where the model's choice exactly matches the correct option, divided by 500.

**Human–model agreement.** We sampled 50 items from the dataset, each comprising a multi-step reasoning question and its decomposed sub-questions. Three annotators independently judged whether the model's answer was semantically consistent with the gold answer for each (binary: *yes/no*). For each sub-question, the human judgment was determined by majority vote. We then queried **Claude Sonnet 4**, **GPT-4o**, and **Gemini 2.5 Flash** via API to obtain their judgments on the same items. An item was counted as *consistent* for a model only if the model's judgments for all sub-questions and the final question matched the human judgments. Results showed human–model agreement of **98%** for GPT-4o, **98%** for Claude, and **96%** for Gemini. Considering cost, we selected **GPT-4o** as the primary judge.

**Inter-model agreement.** Given the binary nature of the task and the observed human–model agreement rates (98%, 98%, 96%), the conservative lower-bound on inter-model agreement is **96%**. We therefore conclude that a single, top-performing judge model suffices for our setting, and cross-model adjudication is unnecessary.

**Human–model agreement rates under different language settings.** Prior work has extensively evaluated the performance of different LLMs on the same tasks across different languages, revealing significant gaps between English and low-resource language settingsGupta et al. (2025); Romanou et al. (2024); Xuan et al. (2025). (Xuan et al., 2025) utilize the MMLU-ProX benchmark to demonstrate that leading models, including GPT-4o and Gemini, suffer a massive accuracy drop, up to 30%, when transitioning from English to low-resource languages. Complementing this, (Romanou et al., 2024) highlight a cultural gap in the INCLUDE benchmark, where a lack of regional knowledge accounts for nearly 40% of model failures in non-English contexts. Furthermore, in the educational domain, (Gupta et al., 2025) report that GPT-4o and Gemini exhibit significant performance biases, struggling with complex pedagogical tasks such as feedback generation in languages like Telugu and Farsi compared to their English performance. We have also conducted experiments on using LLMs as evaluators in non-English scenarios. We compared the accuracy of using LLMs to evaluate outputs in English, Malay, and Tamil, with this evaluation performed on 50 semantically identical questions across different languages. We used human judgment as ground truth to calculate each model's Human-Model consistency across different languages. As shown in Table 22, model evaluation accuracy in non-English scenarios is significantly lower than in English scenarios, espe-

cially in low-resource languages. Therefore, our evaluation experiments for model's explaination are currently conducted only in English settings to ensure accurate reflection of model capabilities.

## A.14 RATIONALE UNFAITHFULNESS RATES ACROSS REPHRASE VQA AND SPOKEN QA

The results for LLMs' Rationale Unfaithfulness Rates (RUR) across Rephrase VQA and Speech are shown in Fig. 18. We observe patterns consistent with the Text-Only and VQA modalities: closed-source models generally have lower RURs than open-source models, though they still fall within the 5%–20% range. Among open-source models, Llama shows a markedly higher RUR on non-Spanish languages than on Spanish, which we attribute to linguistic bias stemming from the disproportionately large share of Spanish in Llama's training data relative to other languages.

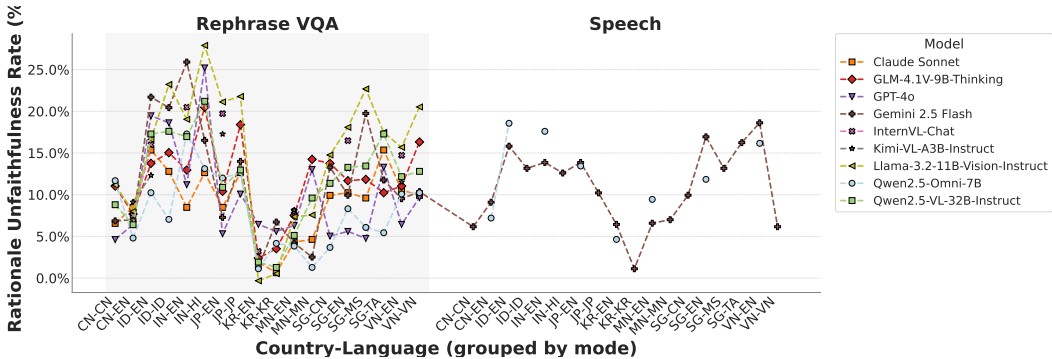

Figure 18: Rationale Unfaithfulness Rates of LLMs across Rephrase VQA and Speech.

## A.15 ATTENTION HEATMAPS FOR INCORRECT MODEL PREDICTIONS

Figures 19 shows attention heatmaps over image regions when the model produces incorrect answers. In Case 2, the model concentrates on the subject's clothing and surrounding scene but overlooks the footwear beside the person and the key text in the upper-left corner that identifies the subject. Similarly, in Figure 3 the model focuses excessively on attire and playing posture while ignoring the crucial detail that the instrument's body is wrapped in snakeskin. All of these support our finding: the model's excessive focus on items mentioned in the prompt causes it to overlook critical cultural details.

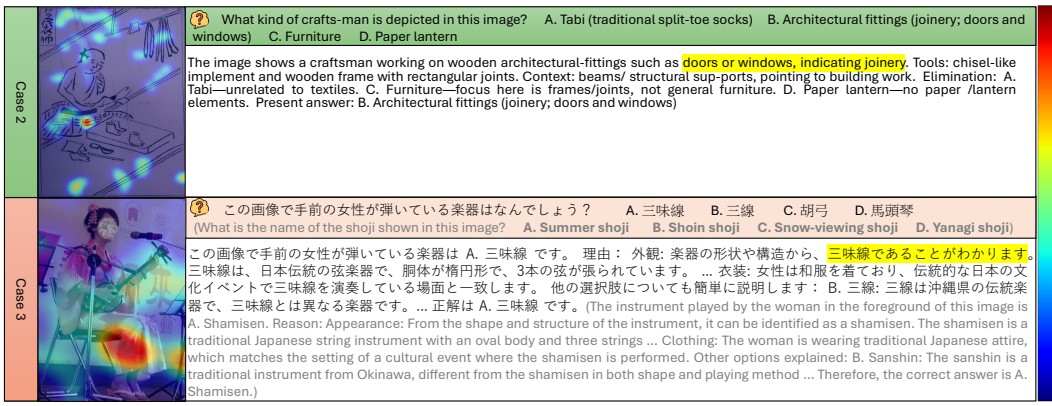

Figure 19: Attention heatmap visualization over image regions during incorrect model answers. Color scale from ■ blue (low) to ■ red (high) indicates increasing model attention.

## A.16 ERROR CASE STUDY

As illustrated in Figure 20, when queried with "What is the mountain in the picture called?", the model exhibits hallucinations under both English and Chinese prompts; however, the Chinese reasoning more closely approximates the ground truth. When the VQA query is reformulated into a

corresponding text-only question, errors under the English prompt are primarily attributed to cross-lingual alignment failures. Specifically, the model erroneously maps "the world's number one mountain" to the Chinese phrase "天下第一奇山", resulting in an incorrect prediction. Conversely, the Chinese prompt elicits the correct answer and explanation. This observation substantiates our hypothesis that image tokens play a more significant role in inducing hallucinations during reasoning. Furthermore, for non-English languages with adequate representation, employing the native language for culture-specific queries consistently enhances performance.

Figure 21 depicts the divergence in model responses across Text-Only, VQA, and Speech modalities for the same inquiry. VQA results indicate that the model lacks specific visual grounding for "Chongqing Guoyuan Port," rendering it unable to recognize the location. Consequently, it relies on generic scene-level inference, which leads to erroneous outputs. In the Speech modality, failures stem from token misinterpretation or transcription errors—for instance, the phonetic misrecognition of "Chongqing Guoyuan" as "Chongqing Guoyan", which similarly yields an incorrect response. In other words, the lack of relevant image training data and the disturbances introduced by speech input both make the model more prone to errors in the VQA and speech modalities than in the text-only modality.

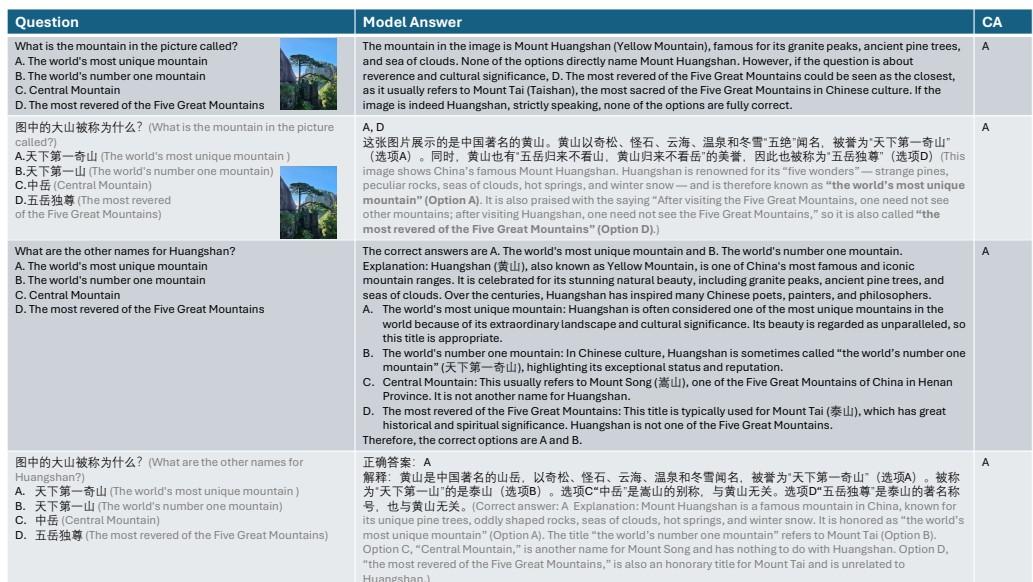

Figure 20: A comparison of the model's responses to the question "What are the other names for Huangshan?" across languages and modalities. The first two examples are VQA questions in English and Chinese, and the last two are text-only questions in English and Chinese.

## A.17 SUB-QUESTION DECOMPOSITION

Table 23 illustrates examples of deconstructing multi-step reasoning questions into single-fact verification subquestions. We enumerate all single-fact subquestions embedded in each multi-step item to assess the model's generalization.

## A.18 THE USE OF LARGE LANGUAGE MODELS

In this work, LLMs are used for three purposes: (1) we use GPT-4o and Claude Sonnet 4 to translate created datasets to the English version; (2) we use GPT-4o to evaluate whether model responses match the reference answers in Section 4.3 and Section 4.6; and (3) we use GPT-4o to polish the manuscript.

## A.19 DATA SENSITIVITY AND PRIVACY DISCUSSION

In this section, we discuss data sensitivity and privacy for three modalities.

**Text.** We collect culture-related textual facts from the public internet and filter out any content containing racism or hate speech. Our "knowledge points" are a few sentences manually summarized by annotators, and all questions are authored from scratch, so no infringement issues are involved.

| Question | Model Answer | CA |
|---|---|---|
| What multimodal transport functions are demonstrated by Chongqing Guoyuan Port?
A. Water-to-water transfer
B. Rail-water intermodal transport
C. Road-water intermodal transport
D. Air-water intermodal transport | The possible options are A, B, C.
Chongqing Guoyuan Port can demonstrate multimodal transport functions including water - to - water transfer, rail - water intermodal transport, and road - water intermodal transport. It doesn't have air - water intermodal transport function. So the answer is A, B, and C.
If you have any other questions about Chinese culture or anything else, feel free to let me know. | A, B, C |
| What multimodal transport functions are reflected in the transportation hub shown in the picture?
A. Water-to-water transfer
B. Rail-water intermodal transport
C. Road-water intermodal transport
D. Air-water intermodal transport | The possible options are B, C.
The image shows a large number of containers, which are typically used for rail and road transport.
There's also a waterway nearby, indicating rail-water intermodal transport and road-water intermodal transport. So, the correct answer is B and C.
What do you think about this? Do you have any other questions related to this? | A, B, C |
| Audio Content: (What multimodal transport functions are demonstrated by Chongqing Guoyuan Port?)
A. Water-to-water transfer
B. Rail-water intermodal transport
C. Road-water intermodal transport
D. Air-water intermodal transport | The correct options are B, C, and D.
Chongqing **Guoyan** Port demonstrates rail - water intermodal transport, road - water intermodal transport, and air - water intermodal transport.
Well, that's the answer. If you have any other questions or want to discuss more about multimodal transport, feel free to let me know. | A, B, C |

Figure 21: Comparison of model responses across text-only, VQA, and speech modalities to the English question "What multimodal transport functions are demonstrated by Chongqing Guoyuan Port?". The first, second, and third examples correspond to text-only, VQA, and speech, respectively, with red highlights marking misinterpretations of the speech input.

**Images.** We use only images under Creative Commons licenses and strictly for research purposes. We apply an automated face-blurring tool (Ren et al., 2023) to protect privacy, followed by manual review to catch any misses or false positives. Please note that faces of historical figures or cartoon characters are not masked. Our images do not contain pornography, violence, or other harmful content.

**Speech.** We obtained voice samples from individuals outside the data team for speech synthesis, without disclosing any personal information. Consent was obtained prior to recording, and the audio is used solely for research.

### A.20    ANALYSIS OF CULTURAL AND LINGUISTIC BIASES

Building upon the phenomena observed in our experiments, we conducted a systematic and in-depth analysis of model biases. Given the intrinsic inextricability of culture and language, we structured our analysis along two dimensions: **Language Bias** and **Culture Bias**, while also exploring the underlying causes of these phenomena.

**Language Bias**    Experimental results reveal a discernible bias across different models regarding linguistic scripts. Specifically, the Llama-3.2-11B model tends to exhibit superior performance for the same cultural context when prompts are formulated in Latin-based languages (e.g., English, Indonesian, and Malay), whereas performance degrades noticeably in non-Latin languages.

As illustrated in Fig.22, when presented with the prompt *"In the city where Qian Xuesen's ancestral home is located, you hear two people having a conversation: A: Qin ah zong ah peng! B: Qin ah zong ah peng! What are they doing?"* in English and Chinese respectively, the model correctly identifies the answer and provides an explanation in the English context. However, in the Chinese context, the model ignores the critical premise regarding *"the city where Qian Xuesen's ancestral home is located,"* resulting in severe hallucinations.

Similarly, Qwen3-30B-A3B-Thinking-2507 demonstrates robust performance on non–low-resource languages but experiences a sharp performance decline in Mongolian and Tamil. Specifically, under identical cultural contexts, performance drops by 20.4% and 6.4% respectively compared to English prompts, highlighting a bias in the distribution of training languages.

Concurrently, we observed that while errors occur in non–low-resource languages, they primarily manifest as inference errors regarding adherence to problem premises. Conversely, in low-resource languages, model outputs often show weak correlation with the question, frequently devolving into mere elaborations of the options. This indicates a marked deficiency in contextual integration for low-resource languages, limiting the models' capacity for cultural comprehension and analysis.

These phenomena not only uncover biases in language distribution under identical cultural contexts but also expose the limitations of cross-lingual knowledge transfer within multilingual Large Language Models (LLMs). Since the next-token prediction task serves as the primary pre-training objective, the majority of data relies on predicting the subsequent token within a monolingual sequence, lacking explicit objectives for cross-lingual semantic space alignment. Consequently, knowledge tends to remain siloed within single languages, restricting cross-lingual transfer and contextual grounding in low-resource languages. Future research should explore methods to leverage

knowledge inherent in high-resource languages to aid understanding and generation in low-resource languages via semantic alignment.

**Culture Bias** Prior literature has established that most models, due to training data distribution, exhibit stronger capabilities in mainstream Western cultures while underperforming in Asian contexts. In our study, we observed similar cultural biases across eight Asian nations. Our analysis reveals that models are most attuned to Korean culture (63.98%), supported by global diffusion and data richness Jang et al. (2024); Dal Yong (2018). Vietnamese culture (62.96%) shows similarly high performance, hypothetically driven by the abundance of digital content resulting from the country's 79.8% social media penetration rate (DataReportal et al., 2025). However, distinct challenges emerge for China and India, where larger performance gaps consistent with cultural and regional heterogeneity are observed. The results for Mongolia fall further behind, corroborating the impact of data scarcity in low-resource language domains.

In terms of data distribution, cultural popularity correlates positively with the quality and quantity of training data (e.g., Korea, Vietnam). However, cultural diversity and complexity serve as significant confounding factors. Although Chinese culture possesses a vast volume of data, its high complexity and regional diversity make it difficult for models to fit; for instance, the distinction between Northern "Tangyuan" and Southern "Yuanxiao" is subtle yet non-negligible. Similarly, India encompasses 22 official languages, with distinct regional customs for the same festivals. Conversely, Mongolia exhibits a significant "long-tail effect": it is not only a typical low-resource language domain but also suffers from data sparsity exacerbated by the concurrent use of Cyrillic and Traditional Mongolian scripts.

From a model training perspective, the local languages of Indonesia, Vietnam, and Malaysia share the Latin alphabet with English, facilitating the cross-lingual transfer of English knowledge during training. Furthermore, shared tokens reduce the compression rate of these languages in the vocabulary, thereby enhancing model performance.

These findings highlight that cultural bias in current models is jointly driven by uneven data distribution and cultural–linguistic complexity (e.g., China, India, Mongolia), beyond simple "data volume" effects seen in Korea and Vietnam. Going forward, a key direction is to curate higher-quality, regionally diverse data and develop training strategies that better support cross-script transfer and fine-grained, intra-cultural distinctions, especially for low-resource and long-tail cultures.

| Question | Category | Language | Model Answer |
|---|---|---|---|
| In the city where Qian Xuesen's ancestral home is located, you hear two people having a conversation:
A: Qin ah zong ah peng!
B: Qin ah zong ah peng!
What are they doing?
**A.** Sawing wood  **B.**  Tug of war
**C.** Fluffing cotton **D.** Rock-paper-scissors | Language/Race | English | In the conversation, the two people are saying \"Qin ah zong ah peng\" which is a Wu Chinese phrase used as a signal to start a game of rock-paper-scissors. |
| 在钱学森祖籍所在的城市，你听到两个人正在对话：
甲：琴啊棕啊棚！
乙：琴啊棕啊棚！
请问，他们这是在做什么呢?
**A.**锯木头 **B.**拔河 **C.**弹棉花 **D.**石头剪刀布 | Language/Race | Chinese | 他们这是在弹棉花。"琴啊棕啊棚"是一句传统的中国谐音谜语，谐音为"弹棉花"。 |
| What characteristics of Mongolian people's production and daily life do traditional saddles reflect?
**A.** Agriculture-based, horse tack primarily used for farm work.
**B.** Nomadic lifestyle and military requirements are equally emphasized.
**C.** Urban trade is frequent, and saddlery primarily focuses on aesthetics.
**D.** Relying on donkey and mule transport, saddle culture had limited influence. | Transportation | English | Possible Option Letters: B
Explanation:
Traditional Mongolian saddles mainly reflect two key aspects of Mongolian life:
**Nomadic lifestyle**
Mongolians have long lived as nomads on the grasslands and relied heavily on horses for moving, herding animals, and daily travel. Their saddles are strong, stable, and suitable for long-distance riding across rough terrain.
**Military needs**
The Mongols were famous warriors. Horses were essential in battle, so the saddles were designed to help riders stay steady and fight while riding, especially for shooting arrows on horseback.
Why the other options are wrong:
Option A: The saddle reflects a mobile, horse-based lifestyle, not mainly agriculture.
Option C: It is about practicality and function, not urban trade or decoration.
Option D: Mongolians mainly relied on horses, not donkeys or mules. So Option B is the most accurate choice. |
| Уламжлалт эмээл нь Монгол ард түмний үйлдвэрлэл, өдөр тутмын амьдралын ямар шинж чанаруудыг тусгасан байдаг вэ?
**A.**Хөдөө аж ахуй нь голлон, морин тоног төхөөрөмж нь ихэвчлэн талбайн ажилд ашиглагддаг.
**B.** Нүүдэлчин амьдрал ба цэргийн хэрэгцээ тэнцүү эрхэмлэгддэг.
**C.** Хотын худалдаа ихтэй, эмээл гоё сайхны эрхэмлэгдэг.
**D.** Илжиг, луусанд тулгуурласан тээвэр, эмээлний соёлын нөлөө хязгаарлагдмал. | Transportation | Mongolian | Буцах боломжит сонголтууд: A, B, D
Тайлбарлал:
A. Хөдөө аж ахуй нь голлон, морин тоног төхөөрөмж нь ихэвчлэн талбайн ажилд ашиглагддаг.
* Уламжлалт эмээл нь Монгол ард түмний үйлдвэрлэлийн үндсэн шинж чанаруудыг тусгасан байдаг. Монгол ард түмний үйлдвэрлэл нь хөдөө аж ахуй эрхэмлэдэг байсан бөгөөд энэ нь талбайн ажил болон мориныг ашигласан хөделмөрөөр илэрдэг. Морины тоног төхөөрөмжийг талбайн ажилд голчлон ашигладаг. Зарим тохиолдолд морийг нүүдэл, өдөр тутмын хөдөлгөөн, аж ахуйн бусад үйл ажиллагаатай холбоотой ажилд ашигладаг.
B. Нүүдэлчин амьдрал ба цэргийн хэрэгцээ тэнцүү эрхэмлэгддэг
* Монгол ард түмний үйлдвэрлэл нь нүүдэлчин амьдралд тулгуурладаг. Нүүдэлчин амьдрал нь тал нутгаар нүүх, мал маллах, байршлаа байнга өөрчлөх зэрэгтэй холбоотой. Үүнтэй зэрэгцэн, Монголчууд цэргийн хэрэгцээг маш өндөрт тавьдаг байсан. Цэргийн хэрэгцээ нь нүүдэлчин амьдралтай салшгүй холбоотой бөгөөд энэ нь Монгол ард түмний үйлдвэрлэлийн үндсэн шинж чанаруудын нэг юм.
D. Илжиг, луусанд тулгуурласан тээвэр, эмээлний соёлын нөлөө хязгаарлагдмал
* Монгол ард түмний үйлдвэрлэлд илжиг, луусанд тулгуурласан тээвэр байсан ч энэ нь үндсэн гол хэсэг нь биш юм. Илжиг, луусны тээвэр нь тодорхой нөхцөлд ашиглагддаг ч морьтой харьцуулахад хязгаарлагдмал нөлөөтэй. Энэ нь эмээлний соёл Монголын нүүдэлчин, морин соёлд илүү гүн шингэснийг харуулдаг.
Сонголт: A, B, D |

Figure 22: **Language and Cultural Bias Cases.** The first and second examples are the English and Chinese versions of the same question; the third and fourth examples are the English and Mongolian versions of the same question.

Table 21: Model outputs across different modes for the same question.

| Mode | Answer |
| --- | --- |
| **Rephrase VQA baseline** | Answer: B. Yan Liang. Explanation: In historical records and the classic Chinese novel *Romance of the Three Kingdoms*, Guan Yu is known for his legendary bravery and martial prowess. One of his most famous feats occurred during the Battle of Hulao Gate (虎牢关之战) in 194 CE, which took place during the late Eastern Han dynasty. Here's the context: **Hua Xiong**: During the Battle of Hulao Gate, Hua Xiong was a general serving under Dong Zhuo, a powerful warlord who controlled the Han imperial court. Hua Xiong challenged the coalition forces led by Yuan Shu, Yuan Shao, and others. Guan Yu volunteered to fight Hua Xiong and defeated him in battle. This event is well-known as one of Guan Yu's early displays of martial skill. **Yan Liang and Wen Chou**: After defeating Hua Xiong, Guan Yu continued to distinguish himself in battle. He later encountered two other generals, Yan Liang and Wen Chou, who were also serving under Dong Zhuo. In the same battle, Guan Yu fought and killed both Yan Liang and Wen Chou. This event further solidified his reputation as a formidable warrior. **Yu Jin**: Yu Jin was a general who served under Cao Cao, a rival of Guan Yu's lord, Liu Bei. While Yu Jin and Guan Yu did face each other in battle, Yu Jin was not killed by Guan Yu. Instead, Yu Jin surrendered to Guan Yu during the Battle of Fancheng (樊城之战) in 219 CE, after being trapped by water flooding caused by Guan Yu. Why the Correct Answer is B (Yan Liang): Yan Liang was explicitly mentioned in historical records and the *Romance of the Three Kingdoms* as one of the generals killed by Guan Yu during the Battle of Hulao Gate. The other options do not align with the historical context: Hua Xiong was defeated but not killed by Guan Yu in the same battle. Wen Chou was also killed by Guan Yu, but the question asks for a single correct answer, and Yan Liang is more commonly associated with this specific event. Yu Jin was not killed by Guan Yu; he surrendered instead. Thus, the correct answer is B. Yan Liang. |
| **VQA baseline** | The image depicts a statue of Guan Yu, one of the most famous and revered generals from the Three Kingdoms period in Chinese history. Guan Yu is known for his loyalty, bravery, and righteousness, and he is often associated with the Shu Han kingdom. Key Points to Consider: Guan Yu's Achievements: Guan Yu is celebrated for his military prowess and is known for several significant victories during the Three Kingdoms era. Hua Xiong (华雄): This is a key figure in the story of the "Battle of Changban" (or "Battle of Changping"). According to historical records and the novel Romance of the Three Kingdoms, Guan Yu famously defeated Hua Xiong in battle. Hua Xiong was a general serving under Dong Zhuo, and his defeat marked a turning point in the coalition against Dong Zhuo. Analysis of the Options: A. Hua Xiong: This is correct. Hua Xiong was indeed killed by Guan Yu during the Battle of Changban. This event is well-documented in both historical records and the Romance of the Three Kingdoms. B. Yan Liang (颜良): While Guan Yu did kill Yan Liang, this occurred later in his career, during the Battle of Baima (White Horse). However, the question specifically asks about the person in the picture, and the most iconic victory associated with Guan Yu is his defeat of Hua Xiong. C. Wen Chou (文丑): Wen Chou was another general defeated by Guan Yu, but this happened during the battle of Xiapi, which came after the Battle of Changban. D. Yu Jin (于禁): Yu Jin was a general of Cao Wei who surrendered to Guan Yu during the Battle of Fancheng. He was not killed by Guan Yu. Final Answer: The person in the picture (Guan Yu) is most famously known for killing Hua Xiong. Therefore, the correct answer is: A. |
| **Vision-ablated Prefix Replay** | The image depicts a statue of Guan Yu, one of the most famous and revered generals from the Three Kingdoms period in Chinese history. Guan Yu is known for his loyalty, bravery, and martial prowess. Analysis of the question: We need to determine which general was killed by Guan Yu based on historical records. **Hua Xiong** (华雄): Hua Xiong was a general serving under Dong Zhuo during the late Eastern Han dynasty. He was killed by Guan Yu in the Battle of Hulao Gate, as described in the Romance of the Three Kingdoms. This event is one of the most iconic moments in Guan Yu's early career. **Yan Liang** (颜良): Yan Liang was a general serving under Yuan Shao. He was killed by Guan Yu during the Battle of Baima (白马之战) in 200 CE. This battle was part of the larger conflict between Cao Cao and Yuan Shao. **Wen Chou** (文丑): Wen Chou was another general serving under Yuan Shao. He was also killed by Guan Yu shortly after the Battle of Baima, during the same campaign against Yuan Shao. **Yu Jin** (于禁): Yu Jin was a general serving under Cao Cao. He was not killed by Guan Yu but rather surrendered to Guan Yu during the Battle of Fancheng in 219 CE when Guan Yu's forces surrounded him. Yu Jin later returned to Cao Cao's service after Guan Yu's defeat. Correct Answer: Based on the historical context and the question, the general who was killed by Guan Yu is Yan Liang. Final Answer: B. |

Table 22: Human–model agreement rates for different models used as LLM-as-judge under different language settings.

| Language | GPT-4o | Claude Sonnet 4 | Gemini 2.5 Flash |
|---------|--------|-----------------|------------------|
| English | **98%** | **98%** | **96%** |
| Malay | 90% | 88% | 92% |
| Tamil | 76% | 80% | 72% |

Table 23: Sub-question decomposition of multi-step reasoning questions.

| Multi-Step Question | Sub-Question | Answer for Sub-Question |
|---------------------|--------------|-------------------------|
| Which of the following buildings belongs to the same ethnic characteristic architecture as the stilted building? A. Tulou B. Moxiaolou C. Diaofang D. Yaodong | What ethnic group is associated with the stilted building? | The Tujia ethnic group. |
| | What ethnic group is associated with the Tulou? | Hakka |
| | What ethnic group is associated with the Moxiaolou? | The Tujia ethnic group. |
| | . . . | . . . |

