# OpenReview forum: "MMA-ASIA: A Multilingual and Multimodal Alignment Framework for Culturally Grounded Evaluation"
_ICLR.cc/2026/Conference — ICLR 2026 Conference Withdrawn Submission_

### Official Review · Reviewer_Lf39 · 2025-10-28

**Soundness:** 3
**Presentation:** 2
**Contribution:** 2
**Rating:** 4
**Confidence:** 4

**Summary:**

The paper introduces MMA-ASIA, a comprehensive benchmark designed to evaluate the cultural awareness and multimodal reasoning capabilities of large language models (LLMs) in Asian contexts. It includes 27,000 human-curated, multilingual, tri-modal (text, image, speech) questions across 8 countries and 10 languages, with most requiring multi-step, culturally grounded reasoning. The authors propose a five-dimensional evaluation protocol assessing cross-lingual, cross-modal, and cultural consistency, along with grounding validity. They also introduce a Cultural Awareness Grounding Validation Module to detect shortcut learning and a Vision-ablated Prefix Replay (VPR) method to analyze model divergence. Extensive baseline evaluations across 14 model families provide insights into current models’ strengths and weaknesses in culturally aligned multimodal understanding.

I find this work timely and relevant, especially as multimodal LLMs expand globally but remain under-evaluated in non-Western contexts. The benchmark’s tri-modal alignment and five-dimensional evaluation are impressive contributions, though I would have appreciated broader linguistic coverage, detailed study of literature review and deeper analyses of cultural-topic interactions.

**Strengths:**

1. Introduced the first tri-modal, multilingual benchmark explicitly designed for Asian cultural contexts, covering multiple modalities (text, image, speech) aligned at the input level for direct cross-modal evaluation.
2. Proposed a comprehensive five-dimensional evaluation protocol capturing diverse aspects of cultural awareness, which incorporates a Cultural Awareness Grounding Validation Module to detect shortcut learning and ensure true reasoning.
3. Provided extensive baseline comparisons across some model families.
4. Provided detailed analysis and ablation studies.

**Weaknesses:**

1. The benchmark covers *only 8 Asian countries and 10 languages*, which is quite limited considering that *Asia has around 50 countries*. This represents roughly *15–20%* of the region’s linguistic diversity. Moreover, the *country selection appears heavily skewed toward Southeast Asia*, with most *Central and Southwest Asian* countries missing. Notably, *Arabic*, one of the most widely spoken Asian languages, is also absent.
2. Lines 46–47 state that the benchmark aims to address *(ii) inadequate representation of low-resource Asian languages)*. However, given the limited geographical and linguistic scope mentioned above, this issue *still persists* and undermines the benchmark’s stated objective.
3. The related works section overlooks several key studies relevant to cross-cultural and multilingual benchmarks. The following papers should be reviewed and cited appropriately: 10.18653/v1/2024.emnlp-main.671, openreview.net/forum?id=DFr5hteojx, openreview.net/forum?id=k3gCieTXeY, 10.48550/arXiv.2505.24119, 10.1145/3701716.371546, openreview.net/forum?id=yxzVanFoij, 10.18653/v1/2025.acl-long.919, 10.48550/arXiv.2412.04261, 10.18653/v1/2025.findings-naacl.341, openreview.net/forum?id=N38Sny0XJi, 10.48550/arXiv.2504.05747
4. Lines 51–52 mention *tri-modal items (textual question, image+question, and Text-to-Speech (TTS)-spoken question)*, but *no information is provided on the distribution of samples across modalities*. Additionally, *there are no evaluation results reported per modality*, which makes it difficult to assess whether multimodal balance was achieved or if one modality dominates.
5. The benchmark explores only *multiple-choice question (MCQ)* formats. Incorporating *open-ended or generative evaluation* tasks could provide a more comprehensive understanding of model reasoning and cultural grounding.
6. In Figure 2, *performance values appear nearly identical* and are *difficult to interpret*. It would be clearer and more informative to present this data in *tabular format* instead of using the current visualization.
7. Claims to provides extensive baseline comparisons across 14 major multilingual and multimodal model families, but very unclear from curent state of the paper.

**Questions:**

1. Why were only 8 Asian countries and 10 languages chosen, given Asia’s vast linguistic and cultural diversity?
2. How does the benchmark justify calling itself “Asian” when Central and Southwest Asian regions are largely excluded? Was Arabic intentionally omitted, and if so, what was the rationale?
3. How does the benchmark address its stated goal of improving representation for low-resource Asian languages despite this limited coverage?
4. Could you clarify the distribution of samples across the three modalities (text, image+text, TTS)?
5. Are there separate evaluation results for each modality to assess cross-modal consistency?
6. Why did you focus solely on multiple-choice questions instead of including open-ended or generative evaluation tasks?
7. Are there plans to extend the benchmark to more countries, languages, or modalities in future work?
8. Could you provide a more detailed analysis of cultural topic effects, how performance varies across themes like religion, tradition, or social norms?
9. Can you include language-level analyses, showing how performance differs across languages and model-language familiarity?
10. Could you add a language–topicperformance analysis to examine whether certain cultural topics are harder for specific languages?
11. Please improve Figure 2 by presenting the results in a tabular format with clear numerical values.
12. Will you update the related work section to include missing key studies (e.g., EMNLP 2024, ACL 2025, and OpenReview papers listed)?
13. How do the benchmark results compare when evaluating cross-cultural generalization versus within-culture accuracy?
14. Did you observe any bias patterns in model outputs linked to specific cultural or linguistic groups, and if so, how were they handled?
15. Check if model size has any impact of the performance and bias?
16. Add detailed results on 14 model fmailies as claimed. Currently, it is very unclear.


Minor: L44: Fix citation formatting → “Balepur et al. (2024); Molfese et al. (2025); Zheng et al. (2023).”
and some more such error across the paper.

---

> ### Author Response · Authors · 2025-11-23
> **To Reviewer Lf39 (Part 1)**
>
> We sincerely thank Reviewer Lf39 for the time and effort dedicated to reviewing our paper. We appreciate the constructive comments and valuable insights, particularly regarding the depth of our cultural analysis. These suggestions have helped us significantly improve the quality and clarity of our work. Below, we provide detailed responses to each question.
>
> ---
>
> ### Q1, Q2 & Q3
>
> **Questions:**
> * Why were only 8 Asian countries and 10 languages chosen, given Asia’s vast linguistic and cultural diversity?
> * How does the benchmark justify calling itself “Asian” when Central and Southwest Asian regions are largely excluded? Was Arabic intentionally omitted, and if so, what was the rationale?
> * How does the benchmark address its stated goal of improving representation for low-resource Asian languages despite this limited coverage?
>
> **Response:**
> Our dataset differs substantially from prior work such as CVQA and ALM-Bench. We aim to construct a **tri-modal alignment dataset** in which over **79%** of the questions require multi-step reasoning, and **we additionally record the specific knowledge point associated with each question** to support our five new evaluation dimensions. This makes data construction significantly more challenging than in CVQA and ALM-Bench: every stage **requires extensive manual effort**. Moreover, to ensure data quality, we did not rely on large language models or crowdsourcing (which would allow us to scale the dataset more easily), but instead involved people who are native to the evaluated culture and language.
>
> **Another major contribution of our paper lies in the dataset construction methodology.** We present the dataset creation pipeline in detail, and this procedure is scalable to cultural settings of any country.
>
> We initially attempted to collaborate with teams from multiple countries, including Arabic-speaking regions, but unfortunately were not able to establish partnerships with teams there. **The current dataset is a starting point rather than an endpoint**: we proposed five important evaluation aspects and evaluated them on selected Asian cultures. In parallel, we plan to collaborate with more researchers from different countries and continuously expand the cultural and linguistic coverage of our dataset, following the construction methodology described in the paper, including adding Arabic. We are looking for collaborations with universities in Thailand and Abu Dhabi, and will include Thai and Arabic data in future versions, while further scaling up our dataset.
>
> In our dataset, **Tamil, Mongolian, and Indonesian** are included to capture the performance of low-resource languages across our five proposed evaluation dimensions. We will incorporate additional low-resource languages in future work.
>
> ***
>
> ### Q4
>
> **Question:**
> Could you clarify the distribution of samples across the three modalities (text, image+text, TTS)?
>
> **Response:**
> For each country, the data are divided into two parts: **multimodal** and **text-only**. As shown in the table below, in the text-only part, there are 500 text-only questions for each country. In the multimodal part, each modality also contains 500 questions, and the corresponding questions are semantically aligned.
>
> | Dataset Part      | Modality                 | Number of questions |
> | :---------------- | :---------------------- | :------------------ |
> | Text-Only Part    | Text                    | 500                 |
> | Multi-modal Part  | Image+Text              | 500                 |
> | Multi-modal Part  | Rephrase Question (Text)| 500                 |
> | Multi-modal Part  | Speech (TTS)            | 500                 |
> ***
>
> ### Q5
>
> **Question:**
> Are there separate evaluation results for each modality to assess cross-modal consistency?
>
> **Response:**
> Yes, Tables 9, 10, 11, and 12 in the appendix report the accuracy for each modality separately.

---

> ### Author Response · Authors · 2025-11-23
> **To Reviewer Lf39 (Part 2)**
>
> ### Q6
>
> **Question:**
> Why did you focus solely on multiple-choice questions instead of including open-ended or generative evaluation tasks?
>
> **Response:**
> Our task is not merely a traditional MCQ task, but an **MCQ + Explanation** task. That is, it involves both multiple-choice selection and open-ended generation. In Section 4.3, we show the gap between using the combined decision based on these two metrics and using only MCQ accuracy. **We have updated the accuracy after combining these two metrics in Appendix A.9.**
>
> ***
>
> ### Q7
>
> **Question:**
> Are there plans to extend the benchmark to more countries, languages, or modalities in future work?
>
> **Response:**
> Yes, we are working on it. We are establishing collaborations with partners in **Thailand and Saudi Arabia** and are currently reaching out to collaborators in additional countries.
>
> ***
>
> ### Q8 & Q9
>
> **Questions:**
> * Could you provide a more detailed analysis of cultural topic effects, how performance varies across themes like religion, tradition, or social norms?
> * Could you add a language–topic performance analysis to examine whether certain cultural topics are harder for specific languages?
>
> **Response:**
> We added an analysis of the *cultural themes* in **Appendix A7**. Specifically, we report model performance across different cultural themes under each modality (Figures 12, 13, and 14), and further investigate the relationship between cultural themes and language to examine whether certain cultural topics are harder for specific languages.
>
> Our analysis is as follows:
>
> > “To investigate how different cultural categories affect model performance, we evaluate models by category under each modality, taking the average accuracy across all languages within a modality as the category’s accuracy. As shown in Figures 12, 13, and 14, under the Text-Only and VQA modalities, models consistently struggle more with the “Fashion/Clothing” and “Transportation” categories compared to others, while performing strongly on “Daily life habits/Culture” and “Education.” In the speech modality, models show slightly weaker performance on “Food/Cuisine” and “History,” but still maintain strong results on “Daily life habits/Culture” and “Education”.
> >
> > These patterns likely reflect both data and task asymmetries across categories. Daily life habits/culture and education rely more on high-frequency, broadly documented facts and relatively coarse-grained reasoning, making them easier for models trained on abundant, well-aligned multilingual data. By contrast, fashion/clothing and transportation encode highly time-sensitive and region-specific concepts (e.g., changing trends, local garment names, route and line names), while food/cuisine and history also depend on rare, culturally bound proper nouns; in the speech modality, models must directly map variable acoustic realizations of these low-frequency terms to the correct concepts, further depressing performance in these categories.
> >
> > We further explore the relationship between cultural types and languages based on Figure 16. When Mongolian is employed as the input language, the model exhibits markedly superior performance in the education category relative to other categories. Conversely, when Hindi, Vietnamese, and Tamil are utilized as input languages, the model demonstrates suboptimal performance in the Fashion/Clothing, Geographical location and climate, and education categories, respectively.
> >
> > We posit that the underlying cause of this phenomenon may be attributed to the fact that for low-resource languages such as Mongolian, casual text from the internet (e.g., social media, forums) is relatively scarce. High-quality corpora for such languages predominantly originate from government documents, digitized textbooks, academic papers, or Wikipedia. This compels the model to ”absorb” a substantial volume of formal, educational texts during the training phase. Consequently, the model exhibits a form of ”overfitting” with exceptionally high performance when processing the education category. In contrast, Hindi internet data may be dominated by news, politics, religion, or literature. Contemporary fashion and clothing-related content on the Indian internet likely exists primarily in English (or Hinglish, a Hindi-English code-mixed variety). If the model is trained exclusively on pure Hindi, it consequently lacks the vocabulary and contextual framework necessary to describe ”fashion.” Similarly, if Vietnamese training corpora lack specific technical descriptions of geography and climate, the model will exhibit domain-specific knowledge gaps. Tamil, being widely used across diverse regions including Sri Lanka, India, and Singapore, encounters substantial educational disparities across these countries, and such regional conflicts may impede the model’s ability to adequately fit educational domain knowledge.”
>
> We have also updated the above content in the paper.

---

> ### Author Response · Authors · 2025-11-23
> **To Reviewer Lf39 (Part 3)**
>
> ### Q10
>
> **Question:**
> Can you include language-level analyses, showing how performance differs across languages and model-language familiarity?
>
> **Response:**
> We had analyzed the role of language in the first paragraph of **Section 4.2 *CULTURAL AWARENESS DISPARITY*.** We argue that a country’s culture is always closely tied to its local language(s) (for example, Chinese culture is mainly analyzed in English and Chinese), and that evaluating a country’s cultural awareness in a non-local language is conceptually questionable—for instance, measuring a model’s knowledge of Japanese culture when prompted in Hindi.
>
> Our main conclusions regarding the relationship between language and model performance are as follows:
>
> 1.  **Overall, cultural performance is strongly shaped by language resources and representation forms.** Low-resource languages systematically hurt cultural performance, as sparse corpora limit models’ ability to learn those cultures (e.g., Mongolian).
> 2.  **English vs. local languages:**
>     * For low-resource languages, English prompts usually work better because English has richer corpora and cross-lingual transfer is limited.
>     * For medium/high-resource languages like Chinese/Japanese, local-language prompts can match or outperform English, especially for culture-specific terms and proper nouns that are poorly represented or inconsistently translated in English.
> 3.  Once a model has strong competence in a local language, prompting in that language gives better cultural alignment and knowledge activation than English.
> 4.  In speech, accents act as a joint cue of language and culture: models like Qwen and Gemini can outperform their standard English baselines because consistent co-occurrence of a given accent with specific cultural content helps them trigger the right cultural knowledge.
>
> **Please refer to Fig. 2 and Section 4.2 *CULTURAL AWARENESS DISPARITY* for more details.**
>
> ***
>
> ### Q11
>
> **Question:**
> Please improve Figure 2 by presenting the results in a tabular format with clear numerical values.
>
> **Response:**
> We had provided all the detailed numerical values underlying Figure 2 in **Tables 9, 10, 11, and 12 of the Appendix**. However, due to space limitations in the main text, we can only place these detailed results in the Appendix.
>
> ***
>
> ### Q12
>
> **Question:**
> Will you update the related work section to include missing key studies (e.g., EMNLP 2024, ACL 2025, and OpenReview papers listed)?
>
> **Response:**
> We had added citations at appropriate places in the paper, except for the following two works:
> * *Aya Expanse: Combining Research Breakthroughs for a New Multilingual Frontier* and *The State of Multilingual LLM Safety Research: From Measuring the Language Gap to Mitigating It* (these are not relevant to our paper).
> * For `10.1145/3701716.371546`, we were unable to find any corresponding paper. We attempted to search via Google, the ACM Digital Library and DOI web, but could not locate any related publication.
>
> ***
>
> ### Q13
>
> **Question:**
> How do the benchmark results compare when evaluating cross-cultural generalization versus within-culture accuracy?
>
> **Response:**
> In fact, our current test set targets **within-culture accuracy**, meaning we evaluate how models perform on each country’s cultural knowledge independently, without interaction between countries’ evaluations. **Cross-cultural generalization** typically refers to training a model on the cultural knowledge of one country and then testing it on another country’s cultural knowledge. However, since the training data of all evaluated LLMs are based on publicly available corpora, we have no access to their internal cultural distributions. As a result, cross-cultural generalization cannot be reliably assessed in our setting.
>
> ***

---

> ### Author Response · Authors · 2025-11-23
> **To Reviewer Lf39 (Part 4)**
>
> ### Q14
>
> **Question:**
> Did you observe any bias patterns in model outputs linked to specific cultural or linguistic groups, and if so, how were they handled?
>
> **Response:**
> Yes. We observe that:
>
> * **Llama-3.2-11B-Vision-Instruct** models tend to perform better for the *same* culture when the prompts use Latin-based languages such as English, Indonesian, and Malay, but perform noticeably worse in non-Latin languages. For example, given the same cultural content, performance in Japanese and Korean is far inferior to that in English.
> * **Qwen3-30B-A3B-Thinking-2507** performs reasonably well on non–low-resource languages, but its performance drops sharply on Mongolian and Tamil, revealing bias in the distribution of its training data.
> * In the Text-Only modality, **GPT-4o** shows much worse performance on the Fashion/Clothing category than on other categories, especially in non-English settings, likely due to insufficient cross-lingual knowledge alignment in this domain.
>
> * At the same time, almost all models exhibit biases across Asian cultures—for instance, they are generally more familiar with Korean and Vietnamese cultures. We provide a more detailed discussion of these patterns in Sec. 4.2.
>
> However, the primary goal of our paper is to explain models’ performance and trends across different cultures, rather than to mitigate or correct these cultural and linguistic biases. In future work, exploring the use of in-context learning to steer the observed biases in model outputs may be a promising direction for addressing this issue.
>
> ***
>
> ### Q15
>
> **Question:**
> Check if model size has any impact of the performance and bias?
>
> **Response:**
> Within a given model family, the performance of models with the same architecture but different sizes **has already been thoroughly reported in their technical reports across a wide range of tasks, with a consistent conclusion that larger variants perform better.** Similarly, the paper *'BLEnD-Vis: Benchmarking Multimodal Cultural Understanding in Vision Language Models'* corroborates the findings of the previous technical report in the cultural domain. **It concludes that within the same model family, given identical architectures and comparable training data, larger models consistently achieve superior performance.** Therefore, we did not conduct further repeated experiments of this kind in our paper.
>
> [1] B. C. Z. Tan, W. Zheng, Z. Liu, N. F. Chen, H. Lee, K. T. W. Choo, and R. K.-W. Lee, "BLEnD-Vis: Benchmarking Multimodal Cultural Understanding in Vision Language Models," arXiv preprint arXiv:2510.11178, 2025.
>
> ***
>
> ### Q16
>
> **Question:**
> Add detailed results on 14 model families as claimed. Currently, it is very unclear.
>
> **Response:**
> Please see the table below. We also add it into **Appendix A5, Table 9.**
>
> | Model | Type | Language Coverage | Modalities |
> | :--- | :--- | :--- | :--- |
> | Qwen2.5-VL-32B-Instruct | Vision–Language (VL) | Multilingual | Image + Text |
> | Llama-3.2-11B-Vision-Instruct | Vision–Language (VL) | Multilingual | Image + Text |
> | Kimi-VL-A3B-Instruct | Vision–Language (VL) | English only | Image + Text |
> | DeepSeek-VL2-Small | Vision–Language (VL) | English only | Image + Text |
> | GLM-4.1V-9B-Thinking | Vision–Language (VL) | Multilingual | Image + Text |
> | InternVL-Chat-V1-5 | Vision–Language (VL) | English only | Image + Text |
> | Qwen2.5-Omni-7B | Omni-modal | Multilingual | Image + Text + Speech |
> | Qwen3-30B-A3B-Thinking-2507 | Text-only LLM | Multilingual | Text |
> | Baichuan-M2-32B | Text-only LLM | Multilingual | Text |
> | GLM-4-9B-Chat | Text-only LLM | Multilingual | Text |
> | Ministral 8B-Instruct | Text-only LLM | Multilingual | Text |
> | GPT-4o | Vision–Language (VL) | Multilingual | Image + Text |
> | Claude-Sonnet-4 | Vision–Language (VL) | Multilingual | Image + Text |
> | Gemini 2.5 Flash | Omni-modal | Multilingual | Image + Text + Speech |

---

> > ### Comment · Reviewer_Lf39 · 2025-11-24
> > **Rebuttal Response to Authors**
> >
> > Dear Authors,
> >
> > Thank you for the detailed rebuttal and the addition of the cultural theme analysis, which I believe improves the paper's depth.
> >
> > However, I retain strong reservations regarding the scope. Calling a benchmark "Asian" with only 8 countries (~15% coverage) remains misleading, and I am not fully convinced by the quality over quantity argument, and I don't think it fully justifies the broad title given the exclusion of Central and Southwest Asia. Will the paper title or claims be adjusted to accurately reflect the current limited geographical coverage?
> >
> > Additionally, in your Q4 response and the paper, you categorize *Rephrase Question (Text)* under the *Multi-modal* Part. Here, I get that the question comes from a multimodal source, but as the evaluation is not being done in a multimodal setup (with text only), I am not convinced that it is the right framing to do.  Please clarify why text rephrasing is considered a distinct modality rather than data augmentation; this appears to misrepresent the multimodal distribution.
> >
> > Regarding Q6, you mention combining MCQ with an explanation metric. I wanted to know how specifically the open-ended explanation is evaluated (e.g., LLM-as-a-Judge, n-gram matching, or human eval).
> >
> > About `10.1145/3701716.3715468`, I'm really sorry. I've checked my notes and found the last `8` missing in the review.
> >
> > Also, the response to Q14 is narrow. Given *the primary goal of our paper is to explain models’ performance and trends across different cultures*, it should be done in more depth so that other works can be built on solving these problems.
> >
> > As several key concerns still remain, I think, I'll keep my score of 4 now.
> >
> > Thank you.

---

> > > ### Author Response · Authors · 2025-11-25
> > > **To Reviewer Lf39 (Round 2, Part 1)**
> > >
> > > We sincerely thank you for your response and valuable feedback. We deeply appreciate this opportunity for discussion and have carefully followed your advice to continuously improve the quality of our paper. Please find our point-by-point responses to your new suggestions below.
> > >
> > > **Q1: However, I retain strong reservations regarding the scope. Calling a benchmark "Asian" with only 8 countries (\~15% coverage) remains misleading, and I am not fully convinced by the quality over quantity argument, and I don't think it fully justifies the broad title given the exclusion of Central and Southwest Asia. Will the paper title or claims be adjusted to accurately reflect the current limited geographical coverage?**
> > >
> > > **A1:** We have carefully considered your suggestion. Given that we are not currently incorporating cultural data from additional countries, we have decided to change the title to **'MMA-SEA: A Multilingual and Multimodal Alignment Framework for Culturally-Grounded Evaluation'**. This title better reflects the scope of countries included in our current study. In future work, we are committed to continuously expanding the range of countries and languages represented in our dataset.
> > >
> > > **Q2: Additionally, in your Q4 response and the paper, you categorize Rephrase Question (Text) under the Multi-modal Part. Here, I get that the question comes from a multimodal source, but as the evaluation is not being done in a multimodal setup (with text only), I am not convinced that it is the right framing to do. Please clarify why text rephrasing is considered a distinct modality rather than data augmentation; this appears to misrepresent the multimodal distribution.**
> > >
> > > **A2:** Our dataset is primarily divided into two components:
> > >
> > > 1.  **The Multimodal Alignment component:** Here, 'multimodal alignment' refers to the inclusion of distinct modalities—such as pure text, VQA, and Spoken QA—where questions across these modalities are semantically equivalent (one-to-one correspondence). **It does not imply that each modality is formed by combining two different modalities; rather, it emphasizes that this subset contains multimodal data where different modalities correspond to one another.**
> > > 2.  **The Text-Only component:** As briefly introduced in the second paragraph of **Section 3.4**, *"This component contains questions that were not suitable for pairing with an image or are inherently text-based."* Additionally, since certain sentences in this component involve code-switching that results in suboptimal TTS generation quality, this subset consists exclusively of pure text data without multimodal counterparts.
> > >
> > > **Q3: Regarding Q6, you mention combining MCQ with an explanation metric. I wanted to know how specifically the open-ended explanation is evaluated (e.g., LLM-as-a-Judge, n-gram matching, or human eval).**
> > >
> > > **A3:** For the open-ended explanations, we state in **Sec. 4.3** that *"We use an LLM-as-Judge approach to verify whether, given a correct answer, the model’s explanation matches the item’s knowledge points."*
> > >
> > > Specifically, we use the Prompt shown in **Appendix A.13, Table 11** to prompt GPT-4o to perform LLM-as-a-judge. The Prompt is
> > >
> > > "You are an expert evaluator. Your task is to determine whether the **Model Answer** correctly and completely incorporates the information from the **Knowledge Point**.
> > >
> > > Knowledge Point:
> > > `{knowledgepoint}`
> > >
> > > Model Answer:
> > > `{modelanswer}`
> > >
> > > If the Model Answer correctly and completely includes the information from the Knowledge Point, **or** if the model’s response aligns with the Knowledge Point’s content and viewpoint, **and** the Model Answer does not contain any factual errors, answer **"yes"**.
> > > Otherwise, answer **"no"**.
> > >
> > > Please only return **"yes"** or **"no"**.
> > > "
> > >
> > > Before adopting the LLM-as-a-judge approach, we first conducted experiments on human–model agreement and model consistency (GPT-4o, Claude Sonnet 4, Gemini 2.5 Flash) to verify its feasibility. Because our evaluation focuses on factual correctness, the consistency among these models reached as high as **96%**. We therefore selected the best-performing GPT-4o as our final evaluation model. For more details, please refer to **Appendix A.13**.
> > >
> > > **Q4: About 10.1145/3701716.3715468, I'm really sorry. I've checked my notes and found the last 8 missing in the review.**
> > >
> > > **A4:** Thank you for the clarification; we will cite it at an appropriate place in the paper.

---

> > > ### Author Response · Authors · 2025-11-25
> > > **To Reviewer Lf39 (Round 2, Part 2)**
> > >
> > > **Q5: **Also, the response to Q14 is narrow. Given the primary goal of our paper is to explain models’ performance and trends across different cultures, it should be done in more depth so that other works can be built on solving these problems.**
> > >
> > > **A5:** Building upon the phenomena observed in our experiments, we conducted a systematic and in-depth analysis of model biases. **Given the intrinsic inextricability of culture and language**, we structured our analysis along two dimensions: **Language Bias** and **Culture Bias**, while also exploring the underlying causes of these phenomena.
> > >
> > > ## Language Bias
> > >
> > > Experimental analysis reveals systematic linguistic bias across models. Llama-3.2 demonstrates superior performance with Latin-script languages (English, Indonesian, Malay) for identical cultural contexts, while exhibiting degraded performance with non-Latin scripts.
> > >
> > > **Table 23** illustrates this phenomenon: the model correctly processes English dialect puns but fails in Chinese contexts, disregarding the critical premise *"the city where Qian Xuesen's ancestral home is located"* and generating severe hallucinations.
> > >
> > > Qwen3 exhibits robust performance on non-low-resource languages but experiences substantial degradation in Mongolian and Tamil (20.4% and 6.4% decreases relative to English, respectively). Notably, errors in non-low-resource languages are predominantly reasoning-based, whereas low-resource language errors manifest as option elaborations disconnected from the query, indicating severe contextual integration deficiencies.
> > >
> > > These phenomena expose **language distribution bias** when processing identical cultural content and underscore **cross-lingual knowledge transfer limitations** in multilingual LLMs. Since next-token prediction, the predominant pre-training objective, primarily incentivizes monolingual token prediction **without explicit cross-lingual semantic alignment**. **Knowledge remains largely compartmentalized within individual languages**, constraining cross-lingual transfer and weakening low-resource language modeling. **Leveraging semantic alignment to transfer high-resource language knowledge for improved low-resource language comprehension constitutes a critical research direction.**
> > >
> > > ## Culture Bias
> > >
> > > Consistent with prior research demonstrating training data imbalances that favor Western cultures, our study identifies systematic cultural bias across eight Asian countries. Models achieve highest accuracy on Korean culture (63.98%), attributable to robust global cultural diffusion and abundant data resources (Jang et al., 2024; Dal Yong, 2018). Vietnamese culture (62.96%) similarly benefits from substantial digital content driven by 79.8% social media penetration (DataReportal et al., 2025). Conversely, China and India exhibit larger performance gaps, correlating with substantial cultural and regional heterogeneity. Mongolia demonstrates the weakest performance, exemplifying data scarcity effects in low-resource settings.
> > >
> > > **From a data distribution perspective**, cultural popularity and global visibility positively correlate with training data quality and quantity (Korea, Vietnam). However, cultural diversity and internal complexity constitute additional critical factors. Despite massive data volumes, the intrinsic complexity of Chinese culture, together with its pronounced regional variation (e.g., northern–southern tangyuan/yuanxiao distinctions), precludes uniform representation. Similarly, India's 22 official languages yield substantial regional variation in festival customs. Mongolia exhibits a "long-tail effect": beyond prototypical low-resource status, the dual usage of Cyrillic and traditional Mongolian scripts exacerbates data sparsity.
> > >
> > > **Regarding model training**, Indonesian, Vietnamese, and Malaysian local languages share Latin orthography with English, **facilitating cross-lingual knowledge transfer**. **Shared subword tokens reduce vocabulary compression requirements, further enhancing representational capacity and downstream performance**. These findings indicate that cultural bias stems from the confluence of uneven data distribution and cultural-linguistic complexity (China, India, Mongolia), rather than data volume alone (Korea, Vietnam). **Future research should prioritize curating high-quality, regionally diverse datasets and developing training strategies supporting cross-script transfer and fine-grained intra-cultural distinctions, particularly for low-resource and long-tail cultures.**
> > >
> > >
> > > (See next part for Table 23. We will update the analysis of these biases in **Appendix A.20. Analysis of cultural and linguistic biases.**)

---

> > > ### Author Response · Authors · 2025-11-25
> > > **To Reviewer Lf39 (Round 2, Part 3)**
> > >
> > > **Table 23: Language and Cultural Bias Cases.** The first and second examples are the English and Chinese versions of the same question; the third and fourth examples are the English and Mongolian versions of the same question.
> > >
> > > | Question | Category | Language | Model Answer |
> > > | :--- | :--- | :--- | :--- |
> > > | **In the city where Qian Xuesen's ancestral home is located, you hear two people having a conversation:** "A: Qin ah zong ah peng! / B: Qin ah zong ah peng!" **What are they doing?** (A. "Sawing wood"; B. "Tug of war"; C. "Fluffing cotton"; D. "Rock-paper-scissors") | Language/Race | **English** | **In the conversation, the two people are saying "Qin ah zong ah peng" which is a Wu Chinese phrase used as a signal to start a game of rock-paper-scissors.** |
> > > | **在钱学森祖籍所在的城市，你听到两个人正在对话：** "甲：琴啊棕啊棚！ / 乙：琴啊棕啊棚！" **请问，他们这是在做什么呢？** (A. 锯木头; B. 拔河; C. 弹棉花; D.石头剪刀布) | Language/Race | **Chinese** | **他们这是在弹棉花。"琴啊棕啊棚"是一句传统的中国谐音谜语，谐音为"弹棉花"。** |
> > > | **What characteristics of Mongolian people's production and daily life do traditional saddles reflect?** (A. Agriculture-based, horse tack primarily used for farm work.; B. Nomadic lifestyle and military requirements are equally emphasized.; C. Urban trade is frequent, and saddlery primarily focuses on aesthetics.; D. Relying on donkey and mule transport, saddle culture had limited influence.) | Transportation | **English** | **Possible Option Letters: B.** Explanation (simplified): Traditional Mongolian saddles mainly reflect two key aspects of Mongolian life: **Nomadic lifestyle** - Mongolians have long lived as nomads on the grasslands and relied heavily on horses for moving, herding animals, and daily travel. Their saddles are strong, stable, and suitable for long-distance riding across rough terrain. **Military needs** - The Mongols were famous warriors. Horses were essential in battle, so the saddles were designed to help riders stay steady and fight while riding, especially for shooting arrows on horseback. **Why the other options are wrong:** Option A: The saddle reflects a mobile, horse-based lifestyle, not mainly agriculture. Option C: It is about practicality and function, not urban trade or decoration. Option D: Mongolians mainly relied on horses, not donkeys or mules. So Option B is the most accurate choice. |
> > > | **Уламжлалт эмээл нь Монгол ард түмний үйлдвэрлэл, өдөр тутмын амьдралын ямар шинж чанаруудыг тусгасан байдаг вэ?** (A.Хөдөө аж ахуй нь голлон, морин тоног төхөөрөмж нь ихэвчлэн талбайн ажилд ашиглагддаг.; B. Нүүдэлчин амьдрал ба цэргийн хэрэгцээ тэнцүү эрхэмлэгддэг.; C. Хотын худалдаа ихтэй, эмээл гоё сайхныг эрхэмлэдэг.; D. Илжиг, луусанд тулгуурласан тээвэр, эмээлний соёлын нөлөө хязгаарлагдмал.) | Transportation | **Mongolian** | **Буцах боломжит сонголтууд: A, B, D.** Тайлбарлал: **A. Хөдөө аж ахуй нь голлон, морин тоног төхөөрөмж нь ихэвчлэн талбайн ажилд ашиглагддаг.** * Уламжлалт эмээл нь Монгол ард түмний үйлдвэрлэлийн үндсэн шинж чанаруудыг тусгасан байдаг. Монгол ард түмний үйлдвэрлэл нь хөдөө аж ахуйг эрхэмлэдэг байсан бөгөөд энэ нь талбайн ажил болон мориныг ашигласан хөдөлмөрөөр илэрдэг. Морины тоног төхөөрөмжийг талбайн ажилд голчлон ашигладаг. Зарим тохиолдолд морийг нүүдэл, өдөр тутмын хөдөлгөөн, аж ахуйн бусад үйл ажиллагаатай холбоотой ажилд ашигладаг. **B. Нүүдэлчин амьдрал ба цэргийн хэрэгцээ тэнцүү эрхэмлэгддэг** * Монгол ард түмний үйлдвэрлэл нь нүүдэлчин амьдралд тулгуурладаг. Нүүдэлчин амьдрал нь тал нутгаар нүүх, мал маллах, байршлаа байнга өөрчлөх зэрэгтэй холбоотой. Үүнтэй зэрэгцэн, Монголчууд цэргийн хэрэгцээг маш өндөрт тавьдаг байсан. Цэргийн хэрэгцээ нь нүүдэлчин амьдралтай салшгүй холбоотой бөгөөд энэ нь Монгол ард түмний үйлдвэрлэлийн үндсэн шинж чанаруудын нэг юм. **D. Илжиг, луусанд тулгуурласан тээвэр, эмээлний соёлын нөлөө хязгаарлагдмал** * Монгол ард түмний үйлдвэрлэлд илжиг, луусанд тулгуурласан тээврийг ашиглах явдал байсан ч энэ нь үндсэн гол хэсэг нь биш юм. Илжиг, луусны тээвэр нь тодорхой нөхцөлд ашиглагддаг ч морьтой харьцуулахад хязгаарлагдмал нөлөөтэй. Энэ нь эмээлний соёл Монголын нүүдэлчин, морин соёлд илүү гүн шингэснийг харуулдаг. Сонголт: A, B, D |

---

> > > > ### Comment · Reviewer_Lf39 · 2025-11-28
> > > > **Rebuttal (Round 2) Response to Authors**
> > > >
> > > > Thanks for the update.
> > > >
> > > >
> > > > The title suggestion sounds appropriate now. Maybe add what "SEA" means in the title too by adding "in Southeast Asia" or so. Also, thanks for the detailed responses in Q3 and Q5. By the way, *96%* consistency (Q3 response) is measured by which method? You should add this in the paper too. A.13 has some data, but not the method used; also, 96% and 98% with that? Majority voting? (in this case, Cohen's kappa) Or, individual? Is it inter-annotator agreement (multiple, like Fleiss's kappa; 4 annotators: 3 human and one AI)? Maybe, consider adding agreement scores (Cohen's kappa) with each of the three annotators too? Also, only 50 samples seems quite inadequate, given 2000 total questions in the dataset (or 27K; I am confused). I am sorry to say, but this paper has such issues everywhere, with some details always missing.
> > > >
> > > > I went through the rebuttal again. And, there are still some problems.
> > > >
> > > > - In Q4, you said you have 500 questions per modality, and Table 1 reports 27,000 total samples. I don't have a clear picture now of what's going on.
> > > > - In Q5, you said, *Yes, Tables 9, 10, 11, and 12 in the appendix report the accuracy for each modality separately.* But they have different things listed there. For example, *Table 9: Overview of selected LLMs Table 10: Number of Incorrect Cases Across Different Models and Languages. Table 11: English prompts for different modals. Table 12: Prompts Used with Closed-Source APIs for Translation, Answer Extraction, and Answer–Reference Consistency Checking.* There are no results.
> > > > - In the appendix, consider adding a table of contents, and the tables and figures also should be close to the sections. We have referred to tables across papers in most questions, which are hard to navigate and understand sometimes.
> > > > - In Q6, the response is Appendix A.9 has only tables (Table 17, 18, 19), but no explanation or analysis on what's going on.
> > > > In Q10, Figure 2 should be improved. There is no logic for making it a line chart in this one. Maybe reporting in a table will be better to compare, given the current situation with a lot of points and lines intersecting each other.
> > > > - In Q11, again, Tables 9-12 do not have such results. Please check before saying.
> > > > - Also, can you provide more explanations for W6 (*In Figure 2, performance values appear nearly identical.*) For example, it seems `CN-zh` and `CN-en` have identical scores across different models in Text Only. Some other combinations show such patterns, too. Why is it? This is what I meant by Q13.
> > > >
> > > > I also agree with several concerning points by `Reviewer fa36` too in his rebuttal response.
> > > >
> > > > So, again, I'm sorry, but I'll keep my overall score. I think it still needs a major rework in terms of presentation, validation, analysis depth, and proposing proper future directions guided by the diagnostic insights. I liked the paper at first, but as I go through it again and review the rebuttal responses, I keep noticing more flaws, and I am starting to feel less confident about it.

---

> > > > > ### Author Response · Authors · 2025-11-28
> > > > > **To Reviewer  Lf39**
> > > > >
> > > > > Thank you for your questions and we want to clarify two points.
> > > > > 1. We believe the paper already describes the dataset distribution. Specifically, we state that both the Text-Only component and the Multimodal component contain 500 base questions in each part. These 500 questions are further instantiated across different modalities and language versions.
> > > > > Using China as an example: the Text-Only component contains 500 questions in Chinese and 500 corresponding questions in English. In the Multimodal component, there are 500 rephrased VQA questions, 500 VQA questions, and 500 Spoken QA questions. Each of these modalities is available in both English and Chinese, yielding (500* 3 * 2) question instances. Following this construction, the total number of questions across all countries is 27,000.
> > > > >
> > > > > 2. When we submitted our first-round rebuttal, the contents of Tables 9, 10, 11, and 12 in the manuscript did indeed correspond exactly to the numerical results underlying Figure 2. However, as the manuscript has gone through multiple rounds of revision, the ordering of the tables has changed. You may verify this by checking the earliest version in the “Revisions” history or the version you originally downloaded. During the rebuttal process, different reviewers requested different changes, which required us to adjust and update the manuscript accordingly. **We therefore kindly ask that such details be checked in a timely manner against the corresponding version of the paper; in this case, the misunderstanding appears to have arisen because the tables were revisited only after several rounds of updates.**

---

### Official Review · Reviewer_y7qL · 2025-10-31

**Soundness:** 3
**Presentation:** 3
**Contribution:** 3
**Rating:** 4
**Confidence:** 3

**Summary:**

MMA-ASIA introduces a tri-modal (text, image, speech), multilingual benchmark for evaluating “cultural awareness” across 8 Asian countries, 10 languages, 27k MCQ items with aligned text/VQA/spoken versions and a five-axis protocol (disparity, cross-lingual, cross-modal, generalization, grounding). The paper claims 79%+ multi-step reasoning, human-curated knowledge points, and an LLM-as-judge module to detect shortcut reasoning. Main findings: accuracy is generally modest (<80% for most SOTA models), English often outperforms low-resource languages, cross-lingual and cross-modal consistency is low, and spoken accents sometimes help.

**Strengths:**

- Careful alignment and scope: same items across three modalities and languages (including English + local language) with a clear five-dimension protocol; helpful comparison table vs prior datasets (Table 1).
- Grounding checks: knowledge points per item and Rationale Unfaithfulness Rate (RUR) expose shortcut-driven wins; the paper shows non-trivial RUR even for strong models (Fig. 3).
- Analyses beyond accuracy: cross-lingual and cross-modal consistency plots (Fig. 4), attention heatmaps and a Vision-ablated Prefix Replay diagnostic to probe visual-token effects.

**Weaknesses:**

W1. Impact & new insight are underdeveloped.
- The headline observations (English vs. low-resource gaps; text > VQA > speech; low cross-lingual/modal consistency) are valuable but mostly reconfirm known trends (resource asymmetry, multimodal transfer difficulty) without a strong new causal or design insight that changes how we build or evaluate M(L)LMs. The paper’s own figures support the descriptive nature (accuracy bars in Fig. 2, consistency radars in Fig. 4), but the analysis often stops short of actionable, general lessons (e.g., ablations that isolate dataset artifacts vs. model architecture effects).

W2. Extensiveness: breadth is good, but depth of evaluation is uneven.
- The benchmark spans 8 countries/10 languages and modalities, yet downstream corroboration is limited to MCQ-style accuracy/consistency; there’s no transfer to real-world tasks (retrieval-augmented QA, interactive grounding) to show external validity. The “generalization” analysis is interesting but narrow (two models, English only).

W3. Comparative baselines and protocol clarity need to be crisper in the main text.
- A lot of the critical experimental protocols (prompts, decoding, seeds, hardware) live in the appendix; given the cross-model comparisons, a concise main-paper protocol box would improve trust in compute-matched fairness (prompts shown in Table 7; settings summarized in A.6).

W4. Grounding validator depends on an LLM judge; robustness discussion is brief.
- RUR uses a single LLM-as-judge with a small agreement study; while pragmatic, the approach could shift with judge choice and language. The paper notes high agreement and chooses a single judge for cost, but more sensitivity analysis (judge swaps, multilingual judging) would bolster confidence.

**Questions:**

- What’s the new actionable takeaway? Beyond reporting gaps (Fig. 2, Fig. 4), can you show design interventions a model developer could adopt (e.g., accent-aware speech preconditioning, cross-modal consistency losses) and quantify improvements on MMA-ASIA?
- Protocol box in main paper. Please surface a one-page box (prompts, seeds, decoding, image resizing, hardware) so cross-model comparisons are evidently compute-matched without digging through the appendix. (Table 7, A.6 are close—bring them up front.)
- Judge robustness. How sensitive is RUR to the choice of judge (Claude vs. GPT-4o vs. Gemini) and to judging in local languages rather than English rationales? Consider a small triangulation table.
- Broaden generalization analysis. The step-decomposition study is run on two models in English; please extend to additional models/languages and report where “knowledge is present but integration fails” vs. “knowledge missing”.

---

> ### Author Response · Authors · 2025-11-23
> **To Reviewer y7qL (Part 1)**
>
> We sincerely thank the Reviewer y7qL for the time and constructive feedback. We appreciate the opportunity to clarify our contributions and have conducted additional experiments to address the concerns raised regarding the depth of our evaluation and generalization analysis. Below, we provide detailed responses to each specific comment.
>
> ### Response to W1 and Q1
>
> **W1. Impact & new insight are underdeveloped.**
> The headline observations (English vs. low-resource gaps; text > VQA > speech; low cross-lingual/modal consistency) are valuable but mostly reconfirm known trends (resource asymmetry, multimodal transfer difficulty) without a strong new causal or design insight that changes how we build or evaluate M(L)LMs. The paper’s own figures support the descriptive nature (accuracy bars in Fig. 2, consistency radars in Fig. 4), but the analysis often stops short of actionable, general lessons (e.g., ablations that isolate dataset artifacts vs. model architecture effects).
>
>  And
>
> **Q1. What’s the new actionable takeaway?**
> Beyond reporting gaps (Fig. 2, Fig. 4), can you show design interventions a model developer could adopt (e.g., accent-aware speech preconditioning, cross-modal consistency losses) and quantify improvements on MMA-ASIA?
>
> **Response:**
> Our evaluation framework encompasses five distinct dimensions. The principal findings of this paper are summarized as follows:
>
> **Section 4.2: Cultural Awareness Disparity**
> * **English Proficiency in Low-Resource Contexts:** English often demonstrates superior performance in many low-resource cultural contexts, a finding consistent with prior literature.
> * **Limitations in Medium-to-High Resource Languages:** However, this advantage does not necessarily extend to medium-to-high-resource languages. We identify the influence of culture-specific terminology as a primary factor, representing a novel insight.
> * **Accents as Cultural Cues:** While accents in speech are often treated as noise in general tasks, within the cultural domain, they may serve as valuable cultural cues that actually enhance model performance (**New Insight**).
> * **Modality Performance Hierarchy:** The performance of Omni-models follows a distinct hierarchy: Text-only > VQA > Speech (**New Insight**).
>
> **Section 4.3: Cultural Awareness Grounding Validation**
> * **Shortcut Learning in MCQs:** Models exhibit "shortcut learning" behaviors when answering culture-related multiple-choice questions. Consequently, their actual cultural perception accuracy is significantly lower than their MCQ performance, with discrepancies ranging from 5% to 20% (**New Insight**).
>
> **Section 4.4: Cultural-Awareness Consistency Across Languages**
> * **Inconsistency Across Languages:** Models demonstrate poor cross-lingual consistency in cultural domains, corroborating findings from previous studies.
>
> **Section 4.5: Cultural-Awareness Consistency Across Modalities**
> * **Inconsistency Across Modalities:** Models exhibit suboptimal cross-modal consistency in cultural contexts. Beyond the scarcity of training data, we identify two novel contributing factors:
>     1.  The pitfall of prompt-guided selective attention.
>     2.  The tendency for image tokens to contribute to reasoning hallucinations.
>
> **Section 4.6: Cultural Knowledge Generalization**
> * **Limited Generalization:** There is a notable deficiency in the generalization capabilities of models regarding cultural knowledge (**New Insight**).
>
> In fact, it is important to note that attributing these findings to specific architectural features is difficult in this context due to the diversity of the evaluated models. Insights derived from one architecture rarely generalize to others.
>
> However, we hope that **the pitfall of prompt-guided selective attention** and **the tendency for image tokens to contribute to reasoning hallucinations** will inspire future researchers to develop better solutions to these issues.

---

> ### Author Response · Authors · 2025-11-23
> **To Reviewer y7qL (Part 2)**
>
> ### Response to W2 and Q4
>
> **W2. Extensiveness: breadth is good, but depth of evaluation is uneven.**
>
> And
>
> **Q4. Broaden generalization analysis.**
>
> **Response:**
> Our evaluation is not limited to MCQ: **we also require models to generate explanations for their chosen options, which is an open-ended generation component**. We then use an LLM-as-judge setup to jointly assess the MCQ answer and the generated explanation in order to compute the model’s cultural awareness accuracy. We have added the exact scores for the MCQ+Explanation evaluation metric in **Appendix A.9**, which makes the model’s cultural awareness correctness more clearly demonstrated. In Section 4.3, we report the accuracy differences with and without incorporating model explanations into the final decision.
>
> Our primary goal is to assess the cultural awareness of existing models. Following prior work, we did not include additional techniques such as RAG, as these methods are numerous and heterogeneous, and a fair comparison would make evaluation prohibitively costly and time-consuming. We sincerely appreciate your suggestion and will consider incorporating such methods in future experiments.
>
> Furthermore, when analyzing cross-modal consistency gaps, we introduce a novel Vision-ablated Prefix Replay (VPR) method to probe these differences. The heatmap-based tracing also offers additional insights into the sources of cross-modal discrepancies.
>
> **Updated Generalization Analysis:**
> For the "generalization" analysis, we additionally include Chinese and Indonesian as analysis languages and expand the set of evaluated models to include Claude Sonnet 4, GLM-4-9B-chat, Mistral-8B-Instruct, and Qwen3-30B-A3B-Thinking. **The new results and analyses are presented below** and have also been incorporated into **Section 4.6 “CULTURAL KNOWLEDGE GENERALIZATION”** of the paper. Please refer to the latest uploaded paper for more details.
>
>
> "We investigate the failure modes of Claude Sonnet 4, GLM-4-9B-chat, Mistral-8B-Instruct, and Qwen3-30B-A3B-Thinking on multi-step reasoning tasks across English, Chinese, and Indonesian. Our experiment utilizes problems where each model initially failed, decomposing them into 3–8 atomic sub-questions with ground-truth answers. An LLM (GPT-4o) judges the correctness of each step. The results in Table 2 reveal distinct failure patterns influenced significantly by language resource availability and model architecture.
>
> 1.  **Knowledge Availability vs. Integration:** Compared to low-resource languages, in high-resource languages (English and Chinese), errors are often due to integration failures rather than a complete lack of knowledge. For instance, in Chinese, Claude and GLM achieve high successful correction rates of 41.1% and 40.5% respectively after decomposition, indicating that these models possess the requisite cultural knowledge but struggle to activate it in a single-step inference. Conversely, in Indonesian, performance is strictly bottlenecked by knowledge gaps. Sub-question failure rates soar across all models, peaking at 94.5% for Mistral and 88.2% for GLM, proving that decomposition cannot compensate for fundamental deficits in training data.
> 2.  **The “Lost in Integration” Phenomenon:** While knowledge gaps explain most failures, a notable portion stems from integration failure—where models answer all sub-questions correctly but fail to synthesize the final answer. This is particularly evident in Qwen’s performance on English tasks. Despite a reasonable sub-question accuracy, Qwen exhibits a disproportionately high integration failure rate of 10.7% (compared to ∼2% for GLM and Mistral). This suggests a “reasoning misalignment” where the model successfully retrieves isolated facts in English but lacks the cross-lingual or cultural reasoning capability to connect them coherently. Interestingly, Claude shows a similar but milder trend in Indonesian (9.2% integration failure), hinting that as models operate in weaker languages, their ability to synthesize logical chains degrades faster than their ability to recall facts.
>
> In summary, the transfer and generalization of cultural knowledge pose a significant challenge for language models, irrespective of their scale. In high-resource languages, closed-source model (Claude) exhibits broad knowledge coverage but limited cultural generalization, whereas open-source models are constrained by knowledge gaps and poor transferability. In low-resource language settings, the cultural generalization of closed-source models is more significantly impacted. In contrast, knowledge gaps account for over 88.5% in open-source models, highlighting their deficiency in low-resource language knowledge."

---

> ### Author Response · Authors · 2025-11-23
> **To Reviewer y7qL (Part 3)**
>
> ### Response to W3 and Q2
>
> **W3. Comparative baselines and protocol clarity need to be crisper in the main text.**
> A lot of the critical experimental protocols (prompts, decoding, seeds, hardware) live in the appendix; given the cross-model comparisons, a concise main-paper protocol box would improve trust in compute-matched fairness (prompts shown in Table 7; settings summarized in A.6).
>
> And
>
> **Q2. Protocol box in main paper.**
> Please surface a one-page box (prompts, seeds, decoding, image resizing, hardware) so cross-model comparisons are evidently compute-matched without digging through the appendix. (Table 7, A.6 are close—bring them up front.)
>
> **Response:**
> We have also considered this issue, but including the experimental procedures and settings in the main paper would cause it to exceed the page limit of the ICLR conference. Therefore, we can only place these details in the appendix for now.
>
> ---
>
> ### Response to W4 and Q3
>
> **W4. Grounding validator depends on an LLM judge; robustness discussion is brief.**
> RUR uses a single LLM-as-judge with a small agreement study; while pragmatic, the approach could shift with judge choice and language. The paper notes high agreement and chooses a single judge for cost, but more sensitivity analysis (judge swaps, multilingual judging) would bolster confidence.
>
> And
>
> **Q3. Judge robustness.**
> How sensitive is RUR to the choice of judge (Claude vs. GPT-4o vs. Gemini) and to judging in local languages rather than English rationales? Consider a small triangulation table.
>
> **Response:**
> Our dataset is a culturally grounded fact-based evaluation benchmark, so the assessment involves **minimal subjective factors**. The main criterion is whether the model’s answer is consistent with the factual content recorded in our data. Before selecting the judge models, we conducted both human–human agreement studies and model–model agreement studies. **The inter-model agreement reached 96%, indicating that its impact on the RUR results is negligible.** We then chose, among GPT, Claude, and Gemini, the models with the highest human–model agreement as LLM-as-Judge to further ensure the reliability of the evaluations. More details can be found in **Appendix A.13, Consistency Analysis and Hyperparameter Settings for LLM-as-Judge**.
>
> For non-English results, we note that GPT, Claude, and Gemini may perform substantially **worse** in non-English languages, especially in low-resource languages. Under these settings, the reliability of LLM-as-Judge becomes questionable. Therefore, we did not compute RUR for non-English results. **We leave a detailed analysis for non-English settings to future work, when stronger multilingual models become available to ensure judgment accuracy or when large-scale human evaluation is feasible.**

---

> > ### Comment · Reviewer_y7qL · 2025-11-25
> > **Comment to author**
> >
> > Thank you for the clarification and detailed response. It resolved many of my earlier concerns. I have a few suggestions for revision:
> >
> > - The MCQ+Explanation evaluation metric currently in Appendix A.9 seems important and insightful; I would recommend moving (at least a concise version of) this analysis into the main paper.
> >
> > - Since you mention that some content is placed in the appendix due to space constraints, I suggest reconsidering the prioritization of sections between the main text and the appendix, so that this valuable experiment receives more prominence.
> >
> > - For the statement that “GPT, Claude, and Gemini may perform substantially worse in non-English languages,” it would strengthen the paper to provide some quantitative evidence or references (e.g., performance gaps on multilingual benchmarks) to support this claim. Is it possible to see some results?

---

> > > ### Author Response · Authors · 2025-11-26
> > > **To Reviewer y7qL (Round 2)**
> > >
> > > We sincerely appreciate your continued guidance and valuable comments in this second round of review. We have humbly adopted your suggestions and have carefully revised the paper to incorporate these insights.
> > >
> > > **Q1:**
> > > The MCQ+Explanation evaluation metric currently in Appendix A.9 seems important and insightful; I would recommend moving (at least a concise version of) this analysis into the main paper. Since you mention that some content is placed in the appendix due to space constraints, I suggest reconsidering the prioritization of sections between the main text and the appendix, so that this valuable experiment receives more prominence.
> > >
> > > **A1:**
> > > Following your suggestions, we have adjusted the paper’s structure and **moved a portion of the table content from Appendix A.9 to Section 4.3**. Additionally, **we have included a paragraph in Section 4.3 that evaluates model performance using the combined MCQ+Explanation metric**. Please refer to our updated paper for details. We have highlighted all revisions in blue to facilitate your review.
> > >
> > > ---
> > >
> > > **Q2:**
> > > For the statement that “GPT, Claude, and Gemini may perform substantially worse in non-English languages,” it would strengthen the paper to provide some quantitative evidence or references (e.g., performance gaps on multilingual benchmarks) to support this claim. Is it possible to see some results?
> > >
> > > **A2:**
> > > **We have added an evaluation of Human-Model consistency across different language settings in Appendix A.13.** We cited recent work by other researchers and included our own experimental results involving Tamil and Malay to support this claim. The specific content is as follows:
> > >
> > > > **Human–model agreement rates under different language settings.** Prior work has extensively evaluated the performance of different LLMs on the same tasks across different languages, revealing significant gaps between English and low-resource language settings (Gupta et al., 2025; Romanou et al., 2024; Xuan et al., 2025). Xuan et al. (2025) utilize the MMLU-ProX benchmark to demonstrate that leading models, including GPT-4o and Gemini, suffer a massive accuracy drop, up to 30%, when transitioning from English to low-resource languages. Complementing this, Romanou et al. (2024) highlight a cultural gap in the INCLUDE benchmark, where a lack of regional knowledge accounts for nearly 40% of model failures in non-English contexts. Furthermore, in the educational domain, Gupta et al. (2025) report that GPT-4o and Gemini exhibit significant performance biases, struggling with complex pedagogical tasks such as feedback generation in languages like Telugu and Farsi compared to their English performance.
> > > >
> > > > We have also conducted experiments on using LLMs as evaluators in non-English scenarios. We compared the accuracy of using LLMs to evaluate outputs in English, Malay, and Tamil, with this evaluation performed on 50 semantically identical questions across different languages. We used human judgment as ground truth to calculate each model’s Human-Model consistency across different languages. As shown in Table 22, model evaluation accuracy in non-English scenarios is significantly lower than in English scenarios, especially in low-resource languages. Therefore, our evaluation experiments for model’s explanation are currently conducted only in English settings to ensure accurate reflection of model capabilities.
> > >
> > > **Table 22: Human–model agreement rates for different models used as LLM-as-judge under different language settings.**
> > >
> > > | Language | GPT-4o | Claude Sonnet 4 | Gemini 2.5 Flash |
> > > | :--- | :--- | :--- | :--- |
> > > | English | 98% | 98% | 96% |
> > > | Malay | 90% | 88% | 92% |
> > > | Tamil | 76% | 80% | 72% |
> > >
> > > ---
> > >
> > > ### **We are deeply grateful for your constructive feedback and the time taken to review our paper again. We hope that our response and the updated manuscript have satisfactorily addressed your concerns, and we would be grateful if you could consider re-evaluating our submission.**

---

### Official Review · Reviewer_fa36 · 2025-11-01

**Soundness:** 2
**Presentation:** 3
**Contribution:** 2
**Rating:** 2
**Confidence:** 5

**Summary:**

This paper introduces MMA-ASIA which is a comprehensive evaluation framework for evaluating cultural awareness of LLMs. The newly proposed benchmark that is a part of this framework consists of 27k QA pairs in 10 languages across 8 Asian countries. The benchmark consists of data across 3 modalities of image, text and audio. The proposed 5-dimensional evaluation protocol consists of cultural-awareness disparities across countries, cross-lingual consistency, cross-modal consistency, cultural knowledge generalization, and grounding validity. The paper investigates multiple LLMs. VLMs and audio models to understand and compare their capabilities regarding cultural awareness in this context.

**Strengths:**

1.	Interesting direction: The proposed multilingual multimodal benchmark explores an interesting direction of analyzing the capacity of modern multimodal models in handling data from Asian languages.
2.	Inclusion of audio: The inclusion of audio in the proposed benchmark is a novel contribution as the existing multilingual QA datasets mostly focus on visual and text data.
3.	Clear writing: The writing of the paper is clear, and the concepts explained and performed analysis is easy to understand for the most part.
4.	Detailed supplementary material: The supplementary material is extensive and detailed, as it includes additional key contributions and other analyses.

**Weaknesses:**

Major:
1. Limited question types (MCQ only): The questions included in the proposed benchmark comprise Multiple Choice Questions only. However previous related work has included other type of questions like Short Question Answers, Long Question Answers (Captions) and True/False Question Answers. MCQ has Implicit bias and randomness that can often lead to issues in analysis. Previous works have tackled this by shifting the order of options in multiple evaluation iterations and converting the MCQs to open-ended questions.
2. Issues in Data Quality: The supplementary material shows a few examples of questions in different languages. However, there are grammatical/structural issues in the provided examples. For example, in the Hindi question example, the last 3 lines of the shown question do not make sense in sentence structure and grammatical context.
3. Limited scope in terms of number of languages: The authors propose a benchmark to represent Asian languages but only include 10 languages which are not enough to represent Asia in general. Benchmarks that are not Asia-specific (CVQA [1] and ALM-Bench [2]) include more Asian languages. Also the proposed benchmark does not include languages like Bengali, Arabic, Punjabi and Urdu which are among the most spoken languages in Asia.
4. Inconsistency in model selection: The authors choose an inconsistent array of models which can hurt the overall performance analysis. The models shown vary in scale (3B v/s 11B v/s 32B...) as well as use of reasoning/non-reasoning models in the same analysis.
5. Lack of model scaling analysis: The authors do not perform a model scaling analysis, i.e., using different sizes of the same model (for example, Qwen2.5-VL 3B v/s 7B vs 32B), which is a very helpful study included in benchmark papers.
6. Language support for various included VLMs: The authors do not include KIMI, DeepSeek and InternVL in any VQA analysis for non-English languages. However, as shown in ALM-Bench [2], they can accept these tokens. The performance might not be as good, however it does give some output.
7. Missing analysis of “cultural themes”: The authors mention the use of 9 cultural themes. However, there is no analysis of said themes or how they impact the performance of models.
8. Audio comprising 100% TTS: A component of audio QA should be live recorded, instead of having 100% TTS.

Minor:
1. The term “RI” used in Table 1 has not been explained/addressed/elaborated.
2. The captions for Fig. 1 and Table 1 are incomplete and short
3. More details are required regarding the selection of cultural categories and languages. Sec. 3.2 only states that they were decided through “collaborative discussions”
4. line 309: what is "itsemphknowledge"

**Questions:**

1.	The authors state that issues encountered in machine translation were addressed by human reverification. What types of issues were faced and how often? A statistical analysis would be helpful. How were the issues addressed?
2.	The authors show results on more VLMs as compared to LLMs. However, every VLM has a base LLM, some of which were excluded. What is the reason for that?
3.	What is the authors’ hypothesis behind the low performance of GPT on the Chinese language compared to other models (especially open-source models) when previous works like CVQA [1] and ALM-Bench [2] show otherwise?
4.	Why are models like GPT which have the capacity to accept audio tokes excluded from the audio QA performance analysis?
5.	The authors provide analysis of performance of models based only on MCQ final answer, however the reasoning output by the models is not evaluated/analyzed. Are there any experiments/evaluations that show the efficacy and quality of generated reasoning?
6.	How did you make sure that the VQA questions require visual input and cannot be answered without it (same for audio QA)?
7.	How were the 56 key words mentioned in Sec 3.3 obtained?
[1] Romero, David, et al. "CVQA: culturally-diverse multilingual visual question answering benchmark." Proceedings of the 38th International Conference on Neural Information Processing Systems. 2024.
[2] Vayani, Ashmal, et al. "All languages matter: Evaluating lmms on culturally diverse 100 languages." Proceedings of the Computer Vision and Pattern Recognition Conference. 2025.

---

> ### Author Response · Authors · 2025-11-23
> **To Reviewer fa36 (Part 1)**
>
> We sincerely thank the reviewer for the constructive feedback and insightful comments. These suggestions have significantly helped us improve the quality and clarity of our manuscript. We have carefully considered all points raised and have addressed them point-by-point below.
>
> **W1. Limited question types (MCQ only):** The questions included in the proposed benchmark comprise Multiple Choice Questions only. However previous related work has included other type of questions like Short Question Answers, Long Question Answers (Captions) and True/False Question Answers. MCQ has Implicit bias and randomness that can often lead to issues in analysis. Previous works have tackled this by shifting the order of options in multiple evaluation iterations and converting the MCQs to open-ended questions.
>
> **A1:** Our dataset evaluation is not limited to the MCQ format. **We also require the model to generate an accompanying explanation, which is an open-ended response, and we explicitly evaluate this free-form reasoning.** This is precisely to address the implicit bias and randomness mentioned above, and it is one of the main focuses of this paper, namely, whether the model truly understands the relevant cultural knowledge. Even if we shuffle the option order or convert the question to a T/F format, the model can still guess the correct answer by chance. **By jointly considering the cultural knowledge point and the model’s explanation, we more accurately capture the model’s true performance. Thus, our evaluation in practice combines MCQ with open-ended question answering.** We have added to **Appendix A.9 (Table 16, 17, 18, 19)** the exact evaluation results obtained by combining the two metrics above for your reference.
>
> **W2. Issues in Data Quality:** The supplementary material shows a few examples of questions in different languages. However, there are grammatical/structural issues in the provided examples. For example, in the Hindi question example, the last 3 lines of the shown question do not make sense in sentence structure and grammatical context.
>
> **A2:** This error was caused by a mistake during the drag-and-drop operation in formatting. The sentence in the original dataset is:
> ”रवि दक्षिण एशिया में उष्णकटिबंधीय जलवायु के लिए अनुकूलित पारंपरिक परिधानों का अध्ययन कर रहे हैं। उन्होंने पश्चिम बंगाल में ग्रामीण पुरुषों के एक समूह को पसीना पोंछने, धूप में सिर ढकने और यहां तक ​​कि अस्थायी मास्क के रूप में हल्के सूती कपड़े का उपयोग करते हुए देखा। वह संभवतः किस पारंपरिक वस्तु का उल्लेख कर रहे हैं?”
> **We have updated it in Figure 9.**
>
> **W3. Limited scope in terms of number of languages:** The authors propose a benchmark to represent Asian languages but only include 10 languages which are not enough to represent Asia in general. Benchmarks that are not Asia-specific (CVQA [1] and ALM-Bench [2]) include more Asian languages. Also the proposed benchmark does not include languages like Bengali, Arabic, Punjabi and Urdu which are among the most spoken languages in Asia.
>
> **A3:** Our dataset differs substantially from prior work such as CVQA and ALM-Bench. We aim to construct a **tri-modal alignment dataset** in which over **79%** of the questions require multi-step reasoning, and **we additionally record the specific knowledge point associated with each question to support our five new evaluation dimensions.** This makes data construction significantly more challenging than in CVQA and ALM-Bench: every stage requires extensive manual effort. Moreover, to ensure data quality, we did not rely on large language models or crowdsourcing (which would allow us to scale the dataset more easily), but instead involved people who are native to the evaluated culture and language.
>
> **Another major contribution of our paper lies in the dataset construction methodology.** We present the dataset creation pipeline in detail, and this procedure is scalable to cultural settings of any country. **The current dataset is a starting point rather than an endpoint**: we proposed 5 important evaluation aspects and evaluated them on selected Asian cultures. In parallel, **we plan to collaborate with more researchers from different countries and continuously expand the cultural and linguistic coverage of our dataset**, following the construction methodology described in the paper, including adding Arabic. For Indian culture, given that India has 22 languages, we currently select Hindi, **the most widely spoken and representative**, and also include Tamil. In the future, we will strive to incorporate more Indian languages.

---

> ### Author Response · Authors · 2025-11-23
> **To Reviewer fa36 (Part 2)**
>
> **W4. Inconsistency in model selection:** The authors choose an inconsistent array of models which can hurt the overall performance analysis. The models shown vary in scale (3B v/s 11B v/s 32B...) as well as use of reasoning/non-reasoning models in the same analysis.
>
> **A4:** We followed the evaluation protocols of prior works such as CVQA, ALM-bench, and BLEnD, and similarly **selected representative models with appropriate sizes within each model family for our evaluation**. For example, within the Qwen2.5-VL series, we chose the 32B model.
>
> Moreover, **different model families do not always provide the same set of model sizes**; for instance, the LLaMA family includes an 11B model, whereas Qwen does not. More importantly, **even when model sizes are comparable, their architectures, training data, and other design choices differ.** If we were to analyze all these factors for each individual model, it would shift the focus away from our main goal. Our evaluation is primarily concerned with comparing how different model families perform on cultural datasets, rather than studying performance differences arising from variations in model size or parameters themselves.
>
> **W5. Lack of model scaling analysis:** The authors do not perform a model scaling analysis, i.e., using different sizes of the same model (for example, Qwen2.5-VL 3B v/s 7B vs 32B), which is a very helpful study included in benchmark papers.
>
> **A5:** We followed a large body of prior evaluation work (e.g., CVQA, ALM-bench, BLEnD, CulturalBench) and decided **not** to compare models that share the same architecture but differ only in parameter size. **First**, within a given model family, the performance of models with the same architecture but different sizes has already been thoroughly reported in their technical reports across a wide range of tasks, with a consistent conclusion that larger variants perform better. **Second**, the impact of model size within a family is not specific to our dataset; it is a general phenomenon that can arise on any benchmark. Therefore, we did not conduct a dedicated evaluation along this axis, as it is not closely aligned with the main goals of our study.
>
> **W6. Language support for various included VLMs:** The authors do not include KIMI, DeepSeek and InternVL in any VQA analysis for non-English languages. However, as shown in ALM-Bench [2], they can accept these tokens. The performance might not be as good, however it does give some output.
>
> **A6:** At present, most large language models can technically accept multilingual input because even if their main training corpus is in English, **it inevitably contains some text from other languages**. However, languages that are not well represented in the primary training data will be stored at the byte level during tokenization. As a result, most models can accept inputs in various languages and output some non-English characters.
>
> In practice, though, the model has not really been trained on these languages. Even if its intermediate representations encode some relevant cultural knowledge, **the output is constrained by its limited ability to express that language**, and thus becomes essentially meaningless. Therefore, for models that officially declare support for English only, we do not report their performance in other languages. This is, first, **to ensure fairness in evaluation**, and second, **to correctly reflect the model’s cultural understanding rather than artifacts caused by its limited language expression**.

---

> ### Author Response · Authors · 2025-11-23
> **To Reviewer fa36 (Part 3)**
>
> **W7. Missing analysis of “cultural themes”:** The authors mention the use of 9 cultural themes. However, there is no analysis of said themes or how they impact the performance of models.
>
> **A7:** We added an analysis of the *cultural themes* in Appendix A7. Specifically, **we report model performance across different cultural themes under each modality (Figures 12, 13, and 14), and further investigate the relationship between cultural themes and language to examine whether certain cultural topics are harder for specific languages.**
>
> Our analysis is as follows:
> > “To investigate how different cultural categories affect model performance, we evaluate models by category under each modality, taking the average accuracy across all languages within a modality as the category’s accuracy. As shown in Figures 12, 13, and 14, under the Text-Only and VQA modalities, models consistently struggle more with the “Fashion/Clothing” and “Transportation” categories compared to others, while performing strongly on “Daily life habits/Culture” and “Education.” In the speech modality, models show slightly weaker performance on “Food/Cuisine” and “History,” but still maintain strong results on “Daily life habits/Culture” and “Education”.
> >
> > These patterns likely reflect both data and task asymmetries across categories. Daily life habits/culture and education rely more on high-frequency, broadly documented facts and relatively coarse-grained reasoning, making them easier for models trained on abundant, well-aligned multilingual data. By contrast, fashion/clothing and transportation encode highly time-sensitive and region-specific concepts (e.g., changing trends, local garment names, route and line names), while food/cuisine and history also depend on rare, culturally bound proper nouns; in the speech modality, models must directly map variable acoustic realizations of these low-frequency terms to the correct concepts, further depressing performance in these categories.
> >
> > We further explore the relationship between cultural types and languages based on Figure 16. When Mongolian is employed as the input language, the model exhibits markedly superior performance in the education category relative to other categories. Conversely, when Hindi, Vietnamese, and Tamil are utilized as input languages, the model demonstrates suboptimal performance in the Fashion/Clothing, Geographical location and climate, and education categories, respectively.
> >
> > We posit that the underlying cause of this phenomenon may be attributed to the fact that for low-resource languages such as Mongolian, casual text from the internet (e.g., social media, forums) is relatively scarce. High-quality corpora for such languages predominantly originate from government documents, digitized textbooks, academic papers, or Wikipedia. This compels the model to ”absorb” a substantial volume of formal, educational texts during the training phase. Consequently, the model exhibits a form of ”overfitting” with exceptionally high performance when processing the education category. In contrast, Hindi internet data may be dominated by news, politics, religion, or literature. Contemporary fashion and clothing-related content on the Indian internet likely exists primarily in English (or Hinglish, a Hindi-English code-mixed variety). If the model is trained exclusively on pure Hindi, it consequently lacks the vocabulary and contextual framework necessary to describe ”fashion.” Similarly, if Vietnamese training corpora lack specific technical descriptions of geography and climate, the model will exhibit domain-specific knowledge gaps. Tamil, being widely used across diverse regions including Sri Lanka, India, and Singapore, encounters substantial educational disparities across these countries, and such regional conflicts may impede the model’s ability to adequately fit educational domain knowledge.”
>
> We have updated the paper accordingly.
>
> **W8. Audio comprising 100% TTS:** A component of audio QA should be live recorded, instead of having 100% TTS.
>
> **A8:** In **Appendix A4** of the paper, we state: “Each generated audio sample was individually verified. When errors occurred, such as inappropriate pauses, missing segments, or mispronunciations, we first adjusted the input text and re-synthesized the audio, as TTS systems are often highly sensitive to textual variations. If repeated corrections still failed, we resorted to manual re-recording. Unlike in other language tasks, our requirement here was not fluency or naturalness, but rather clear articulation of the questions and answer options.” Therefore, **the speech component is not 100% synthesized: around 10% of the difficult-to-synthesize cases were recorded manually.**

---

> ### Author Response · Authors · 2025-11-23
> **To Reviewer fa36 (Part 4)**
>
> **Q1:** The authors state that issues encountered in machine translation were addressed by human reverification. What types of issues were faced and how often? A statistical analysis would be helpful. How were the issues addressed?
>
> **A1:** The main problems fall into the following categories, listed with their proportion of all issues and the corresponding handling methods:
> * **Terminology translation errors** – 23%
> * **Culture-specific terms with no direct translation** – 61%
> * **Literal mistranslation of culture-related vocabulary** – 16%
>
> For terminology translation errors, we correct them using dictionaries and consultation with expert linguists. For terms with no direct equivalent, we adopt **transliteration**. For **literal mistranslations of culture-related vocabulary, we first rephrase the original expression to an equivalent meaning and then translate it**.
>
> **Q2.** The authors show results on more VLMs as compared to LLMs. However, every VLM has a base LLM, some of which were excluded. What is the reason for that?
>
> **A2:** When evaluating the VLMs, we considered two types of questions: (1) VQA questions and (2) their corresponding rephrased text-only versions. If a VLM is trained on top of an LLM, **our evaluation on the text-only questions already reflects its capability to handle pure text.** Moreover, our primary goal in testing these two types of questions on VLMs is to **compare how the same model processes inputs with different modalities but identical semantics.** Conducting additional experiments on the underlying LLM brings limited added value to our analysis.
>
> **Q3.** What is the authors’ hypothesis behind the low performance of GPT on the Chinese language compared to other models (especially open-source models) when previous works like CVQA [1] and ALM-Bench [2] show otherwise?
>
> **A3:** Our dataset contains more than **79% multi-step reasoning**, whereas CVQA and ALM-Bench mainly involve single-step reasoning. When evaluating GPT on VQA-style data, we observed that GPT often first guesses the image content and then reasons on top of that guess, which frequently leads to errors.
>
> For example:
> > *Question:* "What is the mountain in the picture called?"
> > *Options:*
> > A. The world's most unique mountain
> > B. The world's number one mountain
> > C. Central Mountain
> > D. The most revered of the Five Great Mountains
> > *Correct Option:* A
> >
> > *GPT’s Answer:* "A. The world's most unique mountain... D. The most revered of the Five Great Mountains. The mountain in the image is **Mount Huangshan (Yellow Mountain)**... Based on the options provided, none of them directly refer to Mount Huangshan. However, if the question is asking about a mountain associated with reverence and cultural significance, **D. The most revered of the Five Great Mountains** could be a plausible answer... If the image is indeed of Mount Huangshan, none of the options perfectly match its description."
>
> However, when we explicitly tell GPT that the mountain in the image is “Huang Shan” and **without passing** the image to GPT, it is able to select the correct answer. This is consistent with our conclusion in Section 4.4 that “**image tokens contribute to reasoning hallucinations.**” Moreover, as we noted in the paper, **Chinese culture is complex and wide-ranging, which makes the questions substantially more difficult once they are formulated as multi-step reasoning problems.** Taken together, these two factors lead to the phenomenon that GPT performs worse on the Chinese cultural subset of our dataset, while performing well on other datasets like “CVQA”.
>
> **Q4:** Why are models like GPT which have the capacity to accept audio tokens excluded from the audio QA performance analysis?
>
> **A4:** Our evaluation on speech mainly targets **omni models**, i.e., models that can simultaneously accept text, image, and audio inputs. The web version of GPT-4o can take all three modalities as input, but in the API offering, **GPT-4o only accepts text and images**, while another model, **GPT-4o-Audio**, accepts text and audio **but cannot take images**. Since neither of them is a single omni model handling all three modalities and thus cannot provide a direct three-way comparison, we have temporarily excluded these models from our speech evaluation.

---

> ### Author Response · Authors · 2025-11-23
> **To Reviewer fa36 (Part 5)**
>
> **Q5:** The authors provide analysis of performance of models based only on MCQ final answer, however the reasoning output by the models is not evaluated/analyzed. Are there any experiments/evaluations that show the efficacy and quality of generated reasoning?
>
> **A5:** **Section 4.3 CULTURAL AWARENESS GROUNDING VALIDATION** has evaluated the reasoning results. Our dataset’s evaluation strategy is as follows: we first require the model to output the MCQ option, and then **provide an explanation for its choice**. We then compare this explanation against the correctness of the manually recorded knowledge points to determine the authenticity of the model’s cultural awareness. In other words, **only when the model both selects the correct option and gives a correct explanation do we consider it to truly understand the relevant culture.** Moreover, because our evaluation content is factual rather than subjective, we adopt an LLM-as-judge method for this part. Before using the LLM judge, we conducted a human agreement study to ensure the reliability of this approach.
>
> At the same time, in **Section 4.5 CULTURAL-AWARENESS CONSISTENCY ACROSS MODALITIES**, we also **manually analyze the reasoning outputs**. We find that images can contribute to hallucinations in the reasoning outputs. **Additional analyses can be found in A.13 ATTENTION HEATMAPS FOR INCORRECT MODEL PREDICTIONS**. We have also added **additional error case studies, which have been included in Appendix A16**:
>
> > “As illustrated in Figure 20, when queried with “What is the mountain in the picture called?”, the model exhibits hallucinations under both English and Chinese prompts; however, the Chinese reasoning more closely approximates the ground truth. When the VQA query is reformulated into a corresponding text-only question, errors under the English prompt are primarily attributed to cross-lingual alignment failures. Specifically, the model erroneously maps “the world’s number one mountain” to the Chinese phrase “天下第一奇山”, resulting in an incorrect prediction. Conversely, the Chinese prompt elicits the correct answer and explanation. This observation substantiates our hypothesis that image tokens play a more significant role in inducing hallucinations during reasoning. Furthermore, for non-English languages with adequate representation, employing the native language for culture-specific queries consistently enhances performance.
> >
> > Figure 21 depicts the divergence in model responses across Text-Only, VQA, and Speech modalities for the same inquiry. VQA results indicate that the model lacks specific visual grounding for “Chongqing Guoyuan Port,” rendering it unable to recognize the location. Consequently, it relies on generic scene-level inference, which leads to erroneous outputs. In the Speech modality, failures stem from token misinterpretation or transcription errors—for instance, the phonetic misrecognition of “Chongqing Guoyuan” as “Chongqing Guoyan”, which similarly yields an incorrect response. In other words, the lack of relevant image training data and the disturbances introduced by speech input both make the model more prone to errors in the VQA and speech modalities than in the text-only modality.”
>
> **Q6.** How did you make sure that the VQA questions require visual input and cannot be answered without it (same for audio QA)?
>
> **A6.** When manually reviewing the constructed VQA questions, we **remove the associated images and ask annotators to judge whether each question can still be answered**. If a question can be answered without seeing the image, it is marked as invalid; otherwise, it is considered to genuinely require visual information.
>
> **Q7.** How were the 56 key words mentioned in Sec 3.3 obtained?
>
> **A7.** The 56 cultural keywords were obtained through a structured brainstorming process. We first gathered a group of researchers and annotators with diverse cultural backgrounds to independently propose candidate terms. We then merged and de-duplicated these candidates, organized them into major cultural words, and iteratively refined the list to ensure both thematic diversity and broad geographical coverage across different Asian regions (also mentioned in Sec. 3.3). The final 56 keywords are those that remained after this filtering for cultural salience, non-redundancy, and regional balance.
>
> ---
>
> **Minor Errors Fix:**
>
> “RI” in Table 1 is “Response interpretability”; and "itsemphknowledge" is “its \emph{knowledge points}”. We have added more details on the topic selection in **Sec. 3.2 of the manuscript as follows and have updated the paper accordingly**. **We have also provided more detailed captions for Fig. 1 and Table 1.**

---

> ### Comment · Reviewer_fa36 · 2025-11-27
> **Rebuttal Response (Part 1)[Weaknesses]**
>
> Dear Authors, thank you for the detailed rebuttal for my review comments which is always a display of good initiative. The author response partially addresses my concerns. The breakdown is as follows:
>
> **W1.** Evaluation of generated reasoning can be helpful in mitigating implicit biases that come with MCQ type questions. However I recommend generating the reasoning first and asking the model to generate the final output based on the reasoning, not the other way around, as Language models can try to justify their choice instead of benefiting from the reasoning to get a final response.
>
> **W2.** That is understandable. I would recommend adding more qualitative examples from the dataset in the final version of the paper for clarity.
>
> **W3.** I acknowledge the authors' contribution of a trimodal dataset, however I am not convinced in terms of the scope of the dataset being limited in terms of languages. The dataset should not claim to cover the entirety of Asia (title) as it comprises of only 10 languages, and geographic coverage is also limited.
>
> **W4.** Thank you for the clarification. However, the core concern remains unresolved. Using models of inconsistent scales (e.g., Qwen2.5-VL 32B vs. LLaMA 11B) weakens several conclusions drawn in the paper. For example, the claim that “Qwen trails closed-source models by 10%” may simply be an artifact of comparing mismatched model sizes rather than a true performance gap. The authors argue that different model families do not always release the same size variants, but this does not prevent a within-scale comparisons using the sizes that do exist. The cited works (ALM-Bench, CVQA, BLEnD) all include structured groupings, e.g., thinking models vs. standard 7-9B models vs. large 30B+ models, to ensure fairness. A more appropriate comparison structure would be something like:
> 1. Thinking models: GLM-4.1V-9B-Thinking, Qwen3-VL-8B-Thinking, SophiaVL-R1, LlamaV-O1
> 2. Normal scale (≈7-11B): Qwen2.5-VL-7B-Instruct, GLM-4.1V-9B, InternVL-Chat, Phi-4-MM, LLaMA-3.1V-11B
> 3. Large scale (30B-70B): models where families release larger variants
>
> This avoids mixing 7B with 32B or 11B with 32B, and allows the authors to make claims about open-source vs. closed-source performance more responsibly. Ultimately, providing experimental results with scale-consistent groupings would resolve this issue far more convincingly than theoretical justification alone.
>
> **W5.** While it is true that technical reports show larger variants tend to perform better, this does not replace the need for scaling analysis on this specific cultural dataset. Cultural and multilingual tasks often exhibit non-standard scaling behavior, and interesting insights can emerge, such as certain cultural categories improving only marginally with scale, others benefiting disproportionately from larger variants, or some languages showing scaling saturation.
> Prior benchmarks that skip scaling analysis do so because their goals differ; this dataset explicitly concerns cultural understanding, where scale-related patterns may be uniquely informative.
> Even a lightweight comparison (e.g., Qwen2.5-VL 3B → 7B → 32B) would strengthen the paper and verify whether cultural competencies scale similarly to generic VQA tasks. Again, empirical results here would be more convincing than reasoning alone.
>
> **W6.** I understand the reasoning behind restricting multilingual evaluation to models that explicitly declare non-English support. However, ALM-Bench and other studies have demonstrated that models like KIMI, DeepSeek, and InternVL can accept and process multilingual tokens, even if imperfectly. Since the paper aims to evaluate cultural and language-grounded understanding, excluding these models entirely may hide meaningful qualitative behaviors or failure patterns. Even low-quality outputs can reveal whether models exhibit partial cultural grounding, whether certain languages trigger systematic misunderstandings, or whether tokenization-level processing still enables partial cultural reasoning. Including these models, with clear disclaimers about limited language support, would give a more complete picture of the cultural landscape across VLMs. Providing even limited experimental results would be far more informative than omitting these models entirely based on theoretical assumptions.
>
> **W7.** I acknowledge and appreciate the addition of cultural analysis results.
>
> **W8.** Thank you for the clarification. I would encourage the authors to add some context to this to the main paper.

---

> > ### Comment · Reviewer_fa36 · 2025-11-28
> > **Rebuttal Response (Part 2)[Questions]**
> >
> > **Q1.** Thank you for providing the high-level breakdown of translation issues. However, the current description remains insufficiently evidenced. Simply reporting percentages does not demonstrate how these categories were defined, how errors were detected, or how the verification process operated in practice.
> > I strongly recommend including:
> > 1. Clear definitions of each error type (e.g., what qualifies as a “terminology error” vs. “literal mistranslation”).
> > 2. The identification process: how were issues flagged, automatically, manually, or both?
> > 3. Annotation protocol:
> >     - How many human reviewers verified each question?
> >     - What instructions/guidelines were they given?
> >     - Was inter-annotator agreement measured?
> > 4. Qualitative examples (preferably several per category) in either the main paper or supplementary material.
> >
> > Without these, the reported percentages lack transparency, and readers cannot assess annotation quality or reproducibility. Adding concrete examples and clearer methodology would substantially strengthen the credibility of your dataset construction.
> >
> > **Q2.** The explanation provided still leaves an important gap. Since the underlying LLMs are the direct foundation of the evaluated VLMs, reporting their results would quantitatively strengthen the author's claim that language-only performance carries over to text-only VQA rephrasings.Instead of inferring this implicitly, the authors could have simply included the underlying LLM results, which would provide a complete picture of multimodal vs. unimodal behavior and allow more direct attribution of improvements (or regressions) to the vision encoder. Given that many prior benchmarks do report both (to enable separation of visual vs. linguistic contributions), including these results would have been preferable. Actual empirical results are more compelling than conceptual justification.
> >
> > **Q3.** This might be difficult to verify exactly, however I recommend the authors to include some qualitative examples which showcase the hypothesized behavior in the paper
> >
> > **Q4.** This is a reasonable response. I would like to recommend using ASR for converting the audio/speech questions to text, for verification for models that cannot directly take audio tokens.
> >
> > **Q5.** As previously mentioned in **W1**, I would recommend the authors to ask models to generate reasoning first and to use that reasoning to get to the final output, instead of the other way around. As far as the reasoning evaluation is concerned an explicit metric to evaluate the reasoning quality can be helpful, instead of a single score for the entire response. The reasoning generated cannot be scored on a binary scale, as a reasoning can be 30% relevant or 80% relevant, and there needs to be an analysis on relevance of reasoning.
> >
> > **Q6.** There needs to be a detailed procedure highlighted for determination of visual requirement for answering a question. Qualitative assessment of questions is not sufficient for this purpose. What factors are considered by the annotators for deciding which question requires visual input and which doesn't? Some examples would also be appreciated.
> >
> > **Q7.** Similar to **Q6**, there needs to be a detailed procedure to describe the final list of keywords. What words did the experts start from, how were keywords shortlisted, what factors were used to determine the importance of said keywords. An in-depth insight into this can elevate the paper.
> >
> > Overall, I appreciate the effort the authors have put into the rebuttal. However, as stated before, my concerns are partially addressed. If the authors address the remaining issues and answer the questions I have, I might consider increasing my score.

---

### Official Review · Reviewer_BrUn · 2025-11-02

**Soundness:** 3
**Presentation:** 3
**Contribution:** 3
**Rating:** 6
**Confidence:** 2

**Summary:**

This paper introduces MMA-ASIA, a benchmark designed to evaluate cultural awareness across Asian languages and cultural contexts. The benchmark spans 8 Asian countries and 10 languages, and covers three input modalities: text, image, and speech. In addition to benchmarking model performance, the authors also analyze why models behave inconsistently across languages and modalities, using a vision-ablated prefix replay method to reveal modality-specific biases and divergence patterns.

**Strengths:**

1. Multilingual and multicultural evaluation is increasingly essential as foundation models are deployed globally, and South & Southeast Asian languages remain significantly underserved in existing benchmarks.
2. The benchmark thoughtfully includes diverse bias categories, capturing cultural, demographic, and contextual nuances relevant to Asian societies.
3. The experimental evaluation is extensive, covering multiple languages, modalities, and model families, which strengthens the validity of the findings.

**Weaknesses:**

1. The contribution is primarily benchmark-focused, with limited methodological novelty beyond the dataset and evaluation setup.
2. The analysis would benefit from a deeper investigation into failure cases and model behavior across languages and modalities.
3. The writing and organization could be improved for clarity and flow, particularly in framing the motivation and guiding the reader through the core contributions.

**Questions:**

1. Have you evaluated the stability of the LLM outputs across repeated queries? In other words, how sensitive are the results to sampling variability, and what level of uncertainty do the models exhibit across trials?
2. Could you elaborate on the significance of the proposed metrics? In particular, how do they differ from metrics used in prior work, and what specific advantages or insights do they provide in the context of cultural awareness evaluation?

---

> ### Author Response · Authors · 2025-11-23
> **To Reviewer BrUn**
>
> We sincerely thank the Reviewer BrUn for the time and insightful comments. We appreciate the constructive feedback, which has been invaluable in helping us clarify our contributions, refine our evaluation metrics, and improve the overall quality of our manuscript. We have carefully considered all points raised and have addressed them point-by-point below.
>
> **W1:**
> The contribution is primarily benchmark-focused, with limited methodological novelty beyond the dataset and evaluation setup.
>
> **A1:**
> We propose several **new perspectives for evaluating model performance in the cultural domain**. In contrast to prior work that evaluates data from a single modality, we are the first to construct an aligned tri-modal dataset and to introduce five distinct dimensions for evaluating models.
>
> To uncover potential differences between VQA and text-only question types, **we further propose a novel Vision-ablated Prefix Replay (VPR) method to probe the underlying causes**. Therefore, our paper makes both dataset and methodological contributions.
>
> ***
>
> **W2:**
> The analysis would benefit from a deeper investigation into failure cases and model behavior across languages and modalities.
>
> **A2:**
> In addition to the error case analysis in Sec. 4.5, **we have added more cross-lingual and cross-modal failure case analyses in Appendix A.16**. The content is shown below and has already been incorporated into the paper:
>
> > “As illustrated in Figure 20, when queried with “What is the mountain in the picture called?”, the model exhibits hallucinations under both English and Chinese prompts; however, the Chinese reasoning more closely approximates the ground truth. When the VQA query is reformulated into a corresponding text-only question, errors under the English prompt are primarily attributed to cross-lingual alignment failures. Specifically, the model erroneously maps “the world’s number one mountain” to the Chinese phrase “天下第一奇山”, resulting in an incorrect prediction. Conversely, the Chinese prompt elicits the correct answer and explanation. This observation substantiates our hypothesis that image tokens play a more significant role in inducing hallucinations during reasoning. Furthermore, for non-English languages with adequate representation, employing the native language for culture-specific queries consistently enhances performance.
> >
> > Figure 21 depicts the divergence in model responses across Text-Only, VQA, and Speech modalities for the same inquiry. VQA results indicate that the model lacks specific visual grounding for “Chongqing Guoyuan Port,” rendering it unable to recognize the location. Consequently, it relies on generic scene-level inference, which leads to erroneous outputs. In the Speech modality, failures stem from token misinterpretation or transcription errors—for instance, the phonetic misrecognition of “Chongqing Guoyuan” as “Chongqing Guoyan”, which similarly yields an incorrect response. In other words, the lack of relevant image training data and the disturbances introduced by speech input both make the model more prone to errors in the VQA and speech modalities than in the text-only modality.”
>
> ***

---

> ### Author Response · Authors · 2025-11-23
> **To Reviewer BrUn (Part 2)**
>
> **W3:**
> The writing and organization could be improved for clarity and flow, particularly in framing the motivation and guiding the reader through the core contributions.
>
> **A3:**
> Based on your suggestions, we have **revised the Abstract and Introduction** as follows, and have updated these changes in the latest version of the paper.
>
> **Revised Abstract:**
> > Large language models (LLMs) are now used worldwide, yet their multimodal understanding and reasoning often degrade outside Western, high-resource settings. A critical, yet underexplored challenge is assessing whether these models possess genuine "cultural awareness" that is consistent across different modalities (text, vision, speech) and languages. We propose MMA-ASIA, a comprehensive framework to evaluate LLMs’ cultural awareness with a focus on Asian contexts. MMA-ASIA centers on a human-curated, multilingual, and multimodally aligned multiple-choice benchmark covering 8 Asian countries and 10 languages, comprising 27,000 questions; over 79% require multi-step reasoning grounded in cultural context, moving beyond simple memorization. Crucially, this is the first dataset aligned at the input level across three modalities: text, image (visual question answering), and speech. This enables direct assessment of cross-modal knowledge transfer. Building on this benchmark, we propose a five-dimensional evaluation protocol that measures -- (i) cultural-awareness disparities across countries, (ii) cross-lingual consistency, (iii) cross-modal consistency, (iv) cultural knowledge generalization, and (v) grounding validity. To ensure rigorous assessment, a Cultural Awareness Grounding Validation Module detects “shortcut learning” by checking whether the requisite cultural knowledge supports correct answers. Finally, through comparative model analysis, attention tracing, and an innovative Vision-ablated Prefix Replay (VPR) method, we probe why models diverge across languages and modalities, offering actionable insights for building culturally reliable multimodal LLMs.
>
> **Revised Introduction (First paragraph):**
> > Large language and vision–language models are being increasingly deployed across various cultures and languages. Yet, their behavior remains uneven; performance is strongest in high-resource, Western contexts and degrades in non-Western settings, particularly across Asia (Chiu et al., 2025; Romero et al., 2024; Vayani et al., 2025; Wang et al., 2024; Myung et al., 2025; Ng et al., 2025). As multimodal, multilingual models proliferate (Chen et al., 2024a; Bai et al., 2023; Jiang et al., 2023; OpenAI et al., 2024; Touvron et al., 2023; Romanou et al., 2024), a fundamental question arises: Does a model possess a coherent underlying understanding of a culture, or does it merely exhibit fragmented knowledge depending on the input modality and language? While recent works have probed cultural knowledge in text or images separately, they fail to capture the consistency of cultural awareness—defined here as the model’s ability to provide stable, reasoned answers to semantically equivalent inputs regardless of whether the prompt is textual, visual, or spoken.
>
> ***
>
> **Q1:**
> Have you evaluated the stability of the LLM outputs across repeated queries? In other words, how sensitive are the results to sampling variability, and what level of uncertainty do the models exhibit across trials?
>
> **A1:**
> In **A.6 PROMPT TEMPLATES AND EXPERIMENTAL SETTINGS**, we state that *“We decode with greedy search (no sampling; `do_sample = false`, `num_beams = 1`), so temperature / top-p / top-k are not used”* and *“For all closed-source models, we set `temperature = 0` to minimize randomness and improve reproducibility,”* in order to ensure output stability and experimental reproducibility. In other words, **since we have eliminated randomness from the experiments, we did not conduct multiple runs and report averaged results.** More experiment details can also be found in A.6.
>
> ***

---

> ### Author Response · Authors · 2025-11-23
> **To Reviewer BrUn (Part 3)**
>
> **Q2:**
> Could you elaborate on the significance of the proposed metrics? In particular, how do they differ from metrics used in prior work, and what specific advantages or insights do they provide in the context of cultural awareness evaluation?
>
> **A2:**
> **Cultural Awareness Evaluation Framework**
> Our goal is to go beyond single-score accuracy and provide a multi-dimensional view of cultural awareness that separates (i) cultural bias, (ii) language transfer, (iii) modality transfer, (iv) shortcut vs. grounded reasoning, and (v) genuine cultural generalization. Our evaluation metrics bridge the gaps in existing culture benchmarks, such as CLIcK, BLEnD, which lack assessments for cross-modal consistency, cross-lingual consistency, cultural knowledge generalization, and the authenticity of cultural perception.
> Below we explain how each metric differs from prior work and what additional insights it provides:
> 1.  **Cultural Awareness Disparity**
>     * *Prior Work:* Previous studies on cultural disparity primarily focused on single modalities (e.g., text-only or VQA), comparing model performance variations across test data from different countries.
>     * *Our Approach:* Our Cultural Awareness Disparity evaluates perception differences across **three distinct dimensions**:
>         * **Fixed Modality:** We select the best-performing language associated with a specific country to represent that country's cultural awareness, allowing us to compare perception differences between different national cultures.
>         * **Fixed National Culture:** Focusing on a single country (internal comparison), we compare how different languages impact cultural awareness.
>         * **Fixed Language:** We compare differences in the model's cultural awareness across different modalities.
>     * *Insight:* This is an innovative, three-dimensional metric rather than a single-dimension comparison. It considers three key external factors (modality, language, and culture) to more comprehensively reveal the causes of disparity.
> 2.  **Cultural Awareness Grounding Validation**
>     * *Prior Work:* Previous papers typically evaluated cultural awareness using independent multiple-choice questions (MCQ), open-ended responses, or True/False questions.
>     * *Our Approach:* Our paper innovatively proposes **combining MCQs with model explanations (open-ended generation)**. We consider the model to genuinely understand the cultural knowledge only when it both selects the correct option and generates an explanation that matches the specific knowledge point.
>     * *Insight:* This metric reveals the authenticity of the model’s perception in the cultural domain. It provides a more precise measure of the model's understanding of specific cultures and minimizes spurious accuracy caused by shortcut behaviors. This is a completely novel evaluation metric.
> 3.  **Cultural-Awareness Consistency Across Languages**
>     * *Prior Work:* Previous papers reported model perception performance for different cultures under different language conditions independently.
>     * *Our Approach:* In contrast, our metric measures the cross-lingual consistency of the model's performance for the same culture across different languages, regardless of whether the answer is correct.
>     * *Insight:* This indicator reflects the model's capability for cross-lingual knowledge transfer.
> 4.  **Cultural-Awareness Consistency Across Modalities**
>     * *Prior Work:* Previous papers reported model performance within specific modalities separately, with no connection established between them.
>     * *Our Approach:* Our metric measures the cross-modal consistency of the model's performance for the same culture across different modalities, regardless of answer correctness.
>     * *Insight:* This indicator reflects the model's capability for cross-modal knowledge transfer and represents a completely new measurement.
> 5.  **Cultural Knowledge Generalization**
>     * *Prior Work:* Previous papers only reported model accuracy in the cultural domain but never considered cultural knowledge generalization—specifically, whether the model can correctly apply knowledge after acquiring it.
>     * *Our Approach:* Our metric innovatively measures whether the model, after mastering multiple related cultural knowledge points, can proficiently apply them.
>     * *Insight:* This is crucial in the cultural domain, where knowledge points are numerous and interconnected, and it is key to judging whether model gains come from stronger memorization (via more parameters) or from **genuine improvements in generalization**.
>
> **Summary**
> Our five evaluation metrics reflect:
> * The disparities in the model's cultural knowledge across countries, modalities, and languages.
> * The model's genuine degree of cultural understanding (grounding).
> * The model's cross-lingual cultural knowledge transfer capability.
> * The model's cross-modal cultural knowledge transfer capability.
> * The model's cultural knowledge generalization capability.

---

### Note · Authors · 2025-12-01

I have read and agree with the venue's withdrawal policy on behalf of myself and my co-authors.